

 SciPost Phys. Lect. Notes 102 (2025)

# High-dimensional random landscapes:
# From typical to large deviations

**Valentina Ros**

Université Paris-Saclay, CNRS, LPTMS, 91405, Orsay, France

valentina.ros@cnrs.fr

*Part of the 2024-07: Theory of Large Deviations and Applications collection*
*Session 123 of the Les Houches School, July 2024*
*published in the Les Houches Summer School Lecture Notes series*

## Abstract

We discuss tools and concepts that emerge when studying high-dimensional random landscapes, i.e., random functions on high-dimensional spaces. As an illustrative example, we consider an inference problem in two forms: low-rank matrix estimation (case 1) and low-rank tensor estimation (case 2). We show how to map the inference problem onto the optimization problem of a high-dimensional landscape, which exhibits distinct geometrical properties in the two cases. We discuss methods for characterizing typical realizations of these landscapes and their optimization through local dynamics. We conclude by highlighting connections between the landscape problem and large deviation theory.

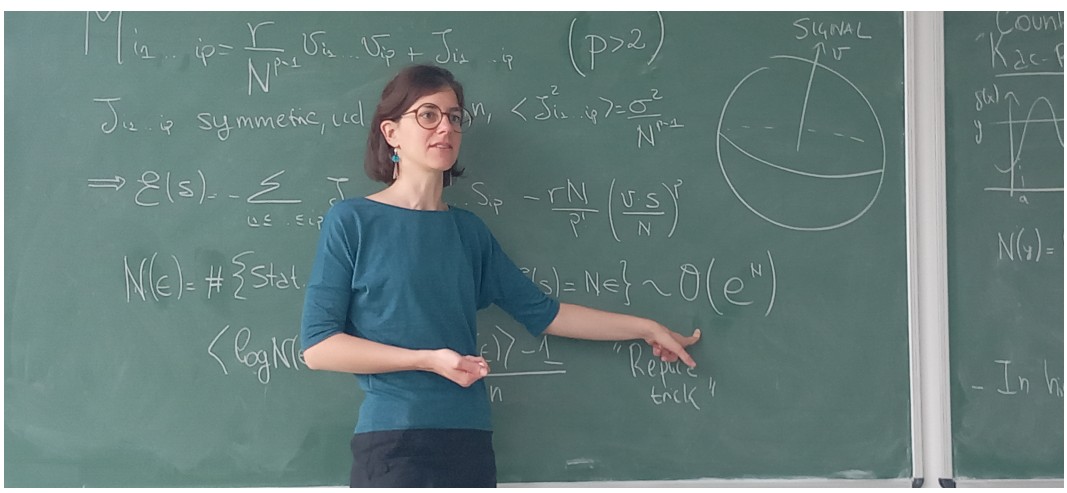

# 1 Random landscapes in high dimension, and these notes

High-dimensional random landscapes are random functions $\mathcal{E}(\mathbf{s})$ of many variables $N \gg 1$, $\mathbf{s} = (s_1, \ldots, s_N)$. They emerge naturally when studying complex systems, which are typically made of a large number of components $i = 1, \ldots, N \gg 1$, that can be agents in an economy, neurons in biological or artificial neural networks, species in ecosystems, particles or spins in materials and so on. Each component can be in different configurations $s_i$, and thus the configuration of the entire system is described by vectors $\mathbf{s}$ belonging to high-dimensional configuration spaces. In complex systems, the components interact with each other, with couplings that are in general heterogeneous and fluctuating from pair to pair (or group to group) of components, both in magnitude and sign. Such interactions are often complicated to write down or to infer from measurements, an occurrence that makes it meaningful to model them as random variables. The complex systems in general evolve by making local moves in configuration space, updating their configuration in the direction that decreases (or increases) the value of some underlying functional $\mathcal{E}(\mathbf{s})$: an energy [1], a fitness [2], or a loss [3] or cost function [4]. These are functions of the configuration $\mathbf{s}$ which encode the random interactions between the constituents: the dynamical evolution can thus be thought of as a (stochastic) optimization problem of a high-dimensional random landscape $\mathcal{E}(\mathbf{s})$.

## 1.1 The high-dimensional landscape program

What can we expect from these optimization processes in high dimensions? Some insight into this question can be obtained by analyzing the structure of the landscape $\mathcal{E}(\mathbf{s})$. For ease of discussion, we henceforth assume that the $s_i$ are continuous variables, and we refer to $\mathcal{E}(\mathbf{s})$ as an "energy landscape" that the system aims at minimizing through its dynamics. Let us summarize a few relevant questions that can be asked in this context:

(1) *The optimizers.* What are the properties of the global minimizer(s) of the energy landscape, the ground state configuration(s) $\mathbf{s}_{\text{GS}}$ defined as $\mathbf{s}_{\text{GS}} = \text{argmin}\,\mathcal{E}(\mathbf{s})$? What is the ground state energy, i.e., the value of the landscape at the minimizer? What is the position of $\mathbf{s}_{\text{GS}}$ in configuration space?

(2) *The landscape topology and geometry.* What is the structure of landscape on top of the global minimum? Are there plenty of *metastable states*, i.e., of local minima of the energy function, that attract the optimization dynamics and tend to trap it for large times? What is their distribution in energy and in configuration space, and how far are them from the ground state or any other special point in configuration space? How does the landscape change when tuning relevant parameters of the problem?

(3) *The optimization dynamics.* What are the typical timescales required to optimize the landscape with local algorithms, and how do they scale with $N$, the dimensionality of the system? Which metastable states trap the dynamics at various timescales, and how does the system escape from such trapping local minima and explore the energy landscape?

The "high-dimensional landscape program" aims at addressing these questions, that we summarized in Figure 1. In this context, we say that the landscape $\mathcal{E}(\mathbf{s})$ is *rugged* whenever its local minima (which we refer to also as metastable states) proliferate: their number grows *exponentially fast with $N$*; that is, it exhibits the same scaling with $N$ as the volume of the configuration space accessible to the system. When this is not the case and the number of local minima does not grow as fast, we say that the landscape is *not rugged*. Because local minima attract and trap the optimization dynamics, when the landscape is rugged the dynamical search for the global minimum is expected to be slow. We say that optimization is *hard*

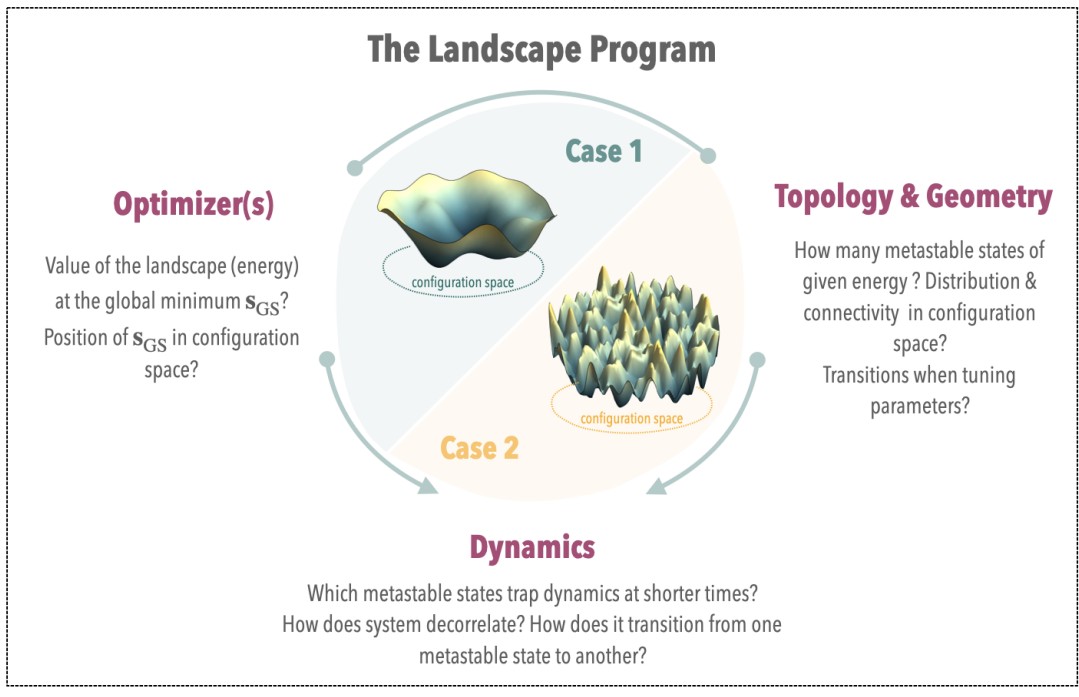

Figure 1: Sketch of the main questions to address in the study of high-dimensional random landscapes. The case 1 discussed in these notes corresponds to a non-rugged landscape, while case 2 corresponds to a rugged, very non-convex landscape.

when it requires timescales that grow exponentially fast with $N$. When the landscape is optimized over timescales that may grow with the system's size $N$, but only polynomially, we say that optimization is *easy*. Glasses are prototypical examples of systems with rugged energy landscapes, in which optimization (that is, equilibration at low/zero temperature) is a hard task. Mean-field models of glasses are associated to energy landscapes with a complicated geometry, with plenty of local minima acting as metastable states for the dynamics, that slow it down and hinder equilibration [5,6]. These numerous local minima have different energies and locations in configuration space, and one can compute their entropy as a function of these properties. In other words, one can do a statistical physics of metastable states, as much as one does a statistical physics of configurations in equilibrium. While the latter is obtained computing averages with respect to the Boltzmann measure, when studying the distribution of metastable states one has to come up with alternative measures tailored to their specific properties. Several techniques to do this have been developed in the context of the theory of glasses, spanning from early foundational works to more recent advancements (some of which are summarized in the review [7]). Recently, the characterization of high-dimensional random landscapes has re-gained prominence, also driven by the emergence of ruggedness and glassy phenomenology in a wide range of fields, including machine learning, quantum control, theoretical biology, and inference (see [8] for a collection of review papers).

## 1.2 These notes

In these notes, we focus on one such problem as a representative example, in order to introduce the key questions and techniques associated with the high-dimensional landscape program. The problem we focus on arises in the context of high-dimensional inference, and it is a problem of denoising. We will explore two variations of it: one corresponding to *low-rank matrix estimation* (case 1) and the other to *low-rank tensor estimation* (case 2). While concep-

tually similar, the associated landscapes differ significantly, as sketched in Fig. 1. In case 1, the energy landscape is not rugged, making the optimization problem relatively simple. In contrast, case 2 features a rugged energy landscape with numerous local minima, resulting in a much harder optimization problem. Case 1 is discussed in Section 2. Although simpler, this example is instrumental for introducing the problem, the key questions and the strategy which are also relevant for case 2, see Sec. 2.1. Sec. 2.2 introduces some notions on one of the main technical tools of these notes: Random Matrix Theory (RMT). RMT is key to analyze the landscape of case 1 in detail, in Sec. 2.3. The Appendices include exercises that complement the main text and provide a guided derivations of some of the statements reported in Section 2. Section 3 is dedicated to case 2. In Sec. 3.1 we introduce the entropy of metastable states (or complexity), and discuss how to characterize typical realizations of these rugged energy landscape, as opposed to rare ones. Sec. 3.2 introduces the tools for computing the distribution of metastable states, such as the Kac-Rice (KR) counting formulas and the Replica Method (RM). In Sec. 3.3 we summarize the main outcomes of the landscape analysis of case 2.

Before turning to the discussion of these problems, two additional remarks are in order. (1) In these notes, we consider the denoising problems as illustrative examples to motivate our discussion of the landscape program. To this end, we focus on a specific inference framework, maximum likelihood, since it naturally maps the denoising problem onto a landscape optimization problem. We caution, however, that this is not the only possible approach to address inference tasks, and in some cases, it may not be the optimal one, as we briefly mention in Secs. 2.3.1 and 3.3.1. For a broader overview of how these denoising problems are addressed within the framework of statistical inference, we refer the interested reader to the review articles [4, 9]. (2) When addressing these problems, our primary focus is on characterizing the structure of the landscape and the features of the dynamics that occur *typically* —i.e., those that occur for *most realizations* of the random landscape, when $N \gg 1$. In other words, we aim to describe the properties of random landscapes and stochastic dynamical trajectories that are realized with probability $\mathbb{P} \to 1$ as $N \to \infty$. When discussing the properties of metastable states, we use the term "typical" to refer to the behavior exhibited by the *majority* of metastable states: these states vastly outnumber those with different properties, as their number grows exponentially relative to the latter. As we shall see, characterizing typical properties in high-dimensional systems requires some effort and ad hoc techniques (such as the Replica Method) to avoid being influenced by *rare events*. However, certain interesting open problems in the landscape program are in fact connected to rare events, which are such that $\mathbb{P} \to 0$ as $N \to \infty$. These rare events are particularly significant in understanding the optimization dynamics of rugged landscapes in high (but not infinite) dimension over *long timescales*. These are dominated by *highly atypical processes*, such as activated events. Characterizing such dynamics in high-dimensional settings remains an open theoretical challenge. This is where the study of random high-dimensional landscapes overlaps strongly with the focus of this Les Houches summer school, Large Deviation Theory (LDT). We discuss some aspects of this in the final Section 4 of these notes.

## 2 Case 1: Quadratic high-dimensional random landscapes

### 2.1 Why: An example from high-dimensional inference

#### 2.1.1 An "easy" inference problem: Noisy matrices

Let us begin by introducing the inference problem. In denoising problems, the goal is to infer a *signal* that has been corrupted by *noise*, see Fig. 2 for an illustration. We focus in these notes on cases in which the signal is high-dimensional (e.g., an $N$-dimensional vector with $N \gg 1$),

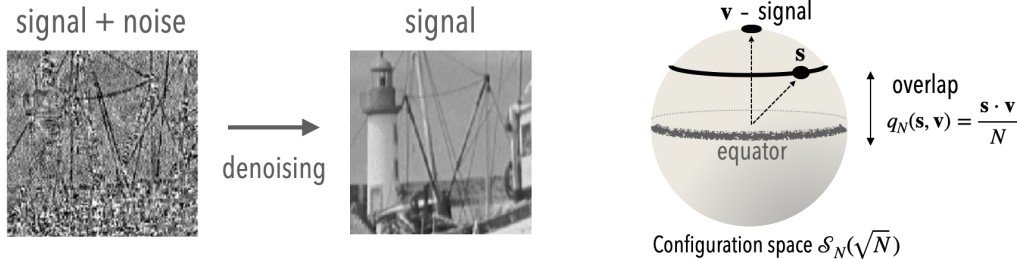

Figure 2: *Left.* An example of denoising problem: an instance of an image (the signal) corrupted by noise is presented, and the goal is to recover the signal. Figure adapted from [11]. *Right.* Sketch of the configuration space associated to the denoising problems discussed in these notes, the hypersphere $\mathcal{S}_N(\sqrt{N})$ embedded in $\mathbb{R}^N$. The signal is a high-dimensional vector $\mathbf{v}$; the portion of configuration space that is orthogonal to the signal is referred to as the *equator* in these notes.

and we aim at understanding under which conditions information about such signal can be recovered *typically* – that is, for most realizations of the noise. This problem naturally fits within the framework of statistical physics [10].

As an illustration, we consider the well-known "spiked matrix" or "low-rank matrix estimation" problem, that was first introduced in [12] and in its various incarnations is still subject of active research. The goal of this problem is to denoise low-rank matrices perturbed by noise. We focus specifically on the case of "rank-one perturbed GOE matrices" or "spiked GOE matrices": these are $N \times N$ matrices $\mathbf{M}$ that admit the decomposition

$$\mathbf{M} = \frac{r}{N}\mathbf{v}\mathbf{v}^T + \mathbf{J}, \qquad \mathbf{v} \in \mathcal{S}_N(\sqrt{N}), \qquad \mathbf{J} \in \mathrm{GOE}(\sigma^2), \tag{1}$$

where the first term is a rank-one matrix, often called the *spike*, corresponding to the signal, while the second term is a matrix with random entries, corresponding to the noise. Let us discuss these two terms separately.

- The signal. The signal is an $N$-dimensional vector $\mathbf{v} = (v_1, \ldots, v_N)$ which satisfies the spherical condition $||\mathbf{v}||^2 = \sum_{i=1}^N v_i^2 = N$, meaning that it belongs to the hypersphere $\mathcal{S}_N(\sqrt{N}) = \{\mathbf{s} : ||\mathbf{s}||^2 = N\}$, that is the configuration space of this problem (see Fig. 2). This vector is fixed, but unknown. It identifies one special direction in the space $\mathcal{S}_N(\sqrt{N})$, and the goal is to figure out which one: of course, reaching this goal is challenging due to the high-dimensionality $N \gg 1$ of the space.

- The noise. The noise is represented by the matrix $\mathbf{J}$, which has random entries with a statistics independent of the signal $\mathbf{v}$. We assume that this matrix is symmetric, $J_{ij} = J_{ji}$, and that the distribution of its entries is Gaussian. Moreover, we consider the particularly simple case in which those entries that are not related by symmetry are uncorrelated and centered, meaning that

$$\mathbb{E}[J_{ij}] = 0, \qquad \mathbb{E}[J_{ij}J_{kl}] = \frac{\sigma^2}{N}\delta_{ik}\delta_{jl}\left(1 + \delta_{ij}\right), \qquad i \leq j, k \leq l. \tag{2}$$

The probability to observe one instance of the matrix $\mathbf{J}$ is thus given by

$$\mathbb{P}_N(\mathbf{J})d\mathbf{J} = \frac{1}{2^{\frac{N}{2}}}\left(\frac{N}{2\pi\sigma^2}\right)^{\frac{N(N+1)}{4}} e^{-\frac{N}{4\sigma^2}\mathrm{Tr}\mathbf{J}^2}\prod_{i \leq j}dJ_{ij}, \qquad \mathrm{Tr}\mathbf{J}^2 = \sum_{i,j}J_{ij}^2. \tag{3}$$

Random matrices with this distribution of the entries belong to the Gaussian Orthogonal Ensemble (GOE) with variance $\sigma^2$, denoted with $\text{GOE}(\sigma^2)$. This terminology refers to the fact that this ensemble is invariant with respect to orthogonal transformations, i.e., to rotations of the orthonormal basis with respect to which the matrix is expressed. Indeed, let $\mathbf{O}$ be a $N \times N$ orthogonal matrix, satisfying $\mathbf{O}\mathbf{O}^T = \mathbf{I}$ where $\mathbf{I}$ is the identity matrix. This matrix defines a change of basis, and the matrix $\mathbf{J}$ expressed in the new basis reads $\mathbf{J}_R = \mathbf{O}\mathbf{J}\mathbf{O}^T$. Rotational invariance of the ensemble corresponds to the statement that $\mathbf{J}$ and $\mathbf{J}_R$ are statistically equivalent, i.e., they have the same probability of occurrence

$$\mathbb{P}_N(\mathbf{J})d\mathbf{J} = \mathbb{P}_N(\mathbf{J}_R)d\mathbf{J}_R. \tag{4}$$

This equality follows from the fact that the trace is cyclic and that $d\mathbf{J}_R = |\det \mathcal{J}(\mathbf{O})| d\mathbf{J} \equiv d\mathbf{J}$, where $\mathcal{J}(\mathbf{O})$ is the Jacobian[1] of the transformation $\mathbf{J} \to \mathbf{O}\mathbf{J}\mathbf{O}^T$. This invariance allows us to draw some consequences on the statistical properties of the eigenvectors of matrices belonging to the GOE ensemble. Indeed, the eigenvectors $\mathbf{u}_R$ of $\mathbf{J}_R$ are related to those of $\mathbf{J}$ by the same orthogonal transformation, $\mathbf{u}_R = \mathbf{O}\mathbf{u}$; given that the two matrices have the same probability of occurrence, their eigenvectors are also *statistically equivalent*. This holds true for any possible rotation $\mathbf{O}$ of the basis in which the eigenvectors are expressed. As a consequence, the statistics of the eigenvectors components is also rotationally invariant. In fact, one finds that the distribution of a single eigenvector, averaged over the GOE ensemble, is equivalent to that of random vectors sampled uniformly from the sphere $\mathcal{S}_N(1) = \left\{\mathbf{u} : \|\mathbf{u}\|^2 = 1\right\}$: It is *isotropic*, with no particular direction singled out.[2]

It is convenient to introduce the *signal-to-noise ratio* $r/\sigma$, which measures the relative strength of the signal with respect to the typical size of fluctuations of the noise. Moreover, for ease of discussion we refer to the subspace of $\mathcal{S}_N(\sqrt{N})$ that is orthogonal to the (unknown) vector $\mathbf{v}$ as *the equator*. We are now in the position to state more precisely the inference problem: we have access to an instance of the noisy matrix $\mathbf{M}$, and we assume knowledge of the signal to noise ratio $r/\sigma$, as well as of the form of the distribution of the noise (3). The goal is to determine whether this information allows us to infer the position of the unknown vector $\mathbf{v}$ on the hypersphere $\mathcal{S}_N(\sqrt{N})$. Tools of statistical physics allow us to answer this question for typical instances of the noisy matrix, i.e., to characterize what happens with high probability with respect to the noise. To see how, we now break down this question into sub-parts, focusing on one specific statistical approach (maximum likelihood) that allows us to map this inference problem into the optimization problem of a high-dimensional landscape.

> **[B1] Maximum Likelihood.** In statistics, maximum likelihood gives a prescription to estimate unknown parameters (in our case, the signal $\mathbf{v}$) based on observed data (in our case, the matrix $\mathbf{M}$). The starting point is the Bayes formula, which dictates the following equality between probabilities:
>
> $$P(\mathbf{v} = \mathbf{s}|\mathbf{M}) = P(\mathbf{M}|\mathbf{v} = \mathbf{s})\frac{P(\mathbf{s})}{P(\mathbf{M})}. \tag{5}$$
>
> In a statistical interpretation, the *posterior* $P(\mathbf{v} = \mathbf{s}|\mathbf{M})$ is the probability that the signal takes

---

[1] The Jacobian has components $\mathcal{J}(\mathbf{O})_{ij,kl} = \partial[J_R]_{ij}/\partial J_{kl} = O_{ik}O_{lj}^T$. Using that $\mathbf{O}\mathbf{O}^T = \mathbf{I}$, one can easily show that $[\mathcal{J}\mathcal{J}^T]_{ij,nm} = \delta_{in}\delta_{jm}$, implying that $|\det\mathcal{J}| = 1$.

[2] Notice that the joint distribution of the complete set of eigenvectors must include the constraint that they are orthogonal to one another, and this orthogonality condition couples the eigenvectors. The orthogonal matrix made by the eigenvectors can however be viewed as being drawn at random from the set of all orthogonal matrices: Every orthonormal basis is equally likely, with no preferred direction. This is phrased mathematically by saying that the eigenvectors are distributed according to the Haar measure on the orthogonal group $O(N)$ of $N \times N$ orthogonal matrices.

a given value $\mathbf{s}$ given that the matrix $\mathbf{M}$ is observed; the function $P(\mathbf{M}|\mathbf{v}=\mathbf{s})$ gives instead the *likelihood* that $\mathbf{M}$ is observed under the assumption that $\mathbf{v} = \mathbf{s}$, while the *prior $P(\mathbf{s})$* encodes any available information on the unknown parameter $\mathbf{v}$. The maximum likelihood estimator is obtained optimizing the likelihood, subject to any constraint imposed by the prior. In the spiked matrix problem, our prior information on the signal corresponds to its normalization, meaning that we can assume $||\mathbf{s}||^2 = N$. Moreover, from the statistics of the noise it follows that

$$P(\mathbf{M}|\mathbf{v}=\mathbf{s}) = \frac{1}{2^{\frac{N}{2}}} \left( \frac{N}{2\pi\sigma^2} \right)^{\frac{N(N+1)}{4}} e^{-\frac{N}{4\sigma^2}\text{Tr}\left(\mathbf{M}-\frac{r}{N}\mathbf{s}\mathbf{s}^T\right)^2}, \qquad (6)$$

see Eq. (3). One can check that the maximizer of this function under the constraint $||\mathbf{s}||^2 = N$ is precisely given by (7).

### 2.1.2 From denoising to high-dimensional landscapes

To address the denoising problem, let us introduce an *estimator* of the signal $\mathbf{v}$, that is, a guess for $\mathbf{v}$ built out of the measured matrix $\mathbf{M}$. For this problem, it is particularly meaningful to consider the *maximum likelihood* estimator [13], defined as

$$\mathbf{s}_{\text{MLE}} := \underset{\mathbf{s}\in\mathcal{S}_N(\sqrt{N})}{\text{argmax}}\ \mathbf{s}^T\mathbf{M}\mathbf{s} = \underset{\mathbf{s}\in\mathcal{S}_N(\sqrt{N})}{\text{argmax}} \sum_{i,j} M_{ij}s_is_j. \qquad (7)$$

As the name suggests, this vector is the one maximizing the likelihood function associated to the inference problem, see the Box [B1]. This estimator is also the vector which maximizes the quadratic form associated to the matrix $\mathbf{M}$. It therefore coincides with the global minimum, i.e., the ground state of the energy landscape:

$$\mathcal{E}_r(\mathbf{s}) = -\frac{1}{2}\sum_{i,j}M_{ij}s_is_j = -\frac{1}{2}\sum_{i,j}J_{ij}s_is_j - \frac{rN}{2}\left(\frac{\mathbf{s}\cdot\mathbf{v}}{N}\right)^2, \qquad \mathbf{s}\in\mathcal{S}_N(\sqrt{N}), \qquad (8)$$

meaning that

$$\mathbf{s}_{\text{MLE}} \equiv \mathbf{s}_{\text{GS}}. \qquad (9)$$

Due to the fact that the noise matrix $\mathbf{J}$ is distributed, (8) defines a high-dimensional random landscape: thus, finding the maximum likelihood estimator amounts to solving the optimization problem for the random landscape $\mathcal{E}_r(\mathbf{s})$. The more the estimator (9) lies in the vicinity of the signal $\mathbf{v}$, the more it is informative of the latter. We define, as a measure of proximity in configuration space $\mathcal{S}_N(\sqrt{N})$, the *overlap* function:

$$q_N(\mathbf{s},\mathbf{s}') := \frac{\mathbf{s}\cdot\mathbf{s}'}{N}. \qquad (10)$$

In the inference setting, one is particularly interested in $q_N(\mathbf{s}_{\text{MLE}},\mathbf{v}) \equiv q_N(\mathbf{s}_{\text{GS}},\mathbf{v})$. What we defined as the equator corresponds to configuration $\mathbf{s}$ such that $q_N(\mathbf{s},\mathbf{v}) = 0$: for any fixed $\mathbf{v}$, this region has a surface that for large $N$ scales exactly as the total surface of the hypersphere, meaning that the overwhelming majority of configurations in $\mathcal{S}_N(\sqrt{N})$ are at the equator when $N \to \infty$. In other words, for any fixed $\mathbf{v}$, if a vector $\mathbf{s}$ is extracted randomly with uniform measure on $\mathcal{S}_N(\sqrt{N})$, then *typically* $q_N(\mathbf{s},\mathbf{v}) \to 0$ when $N \to \infty$, see also the Box [B2].

To gain some intuition on the optimization problem, let us consider some limiting cases.

- $r \to 0$. In this limit, the energy landscape (8) is independent of the signal $\mathbf{v}$. It is a Gaussian field with *isotropic statistics*, satisfying:

$$\mathbb{E}[\mathcal{E}_0(\mathbf{s})] = 0, \qquad \mathbb{E}[\mathcal{E}_0(\mathbf{s})\mathcal{E}_0(\mathbf{s}')] = \frac{N\sigma^2}{2}\left(\frac{\mathbf{s}\cdot\mathbf{s}'}{N}\right)^2 = \frac{N\sigma^2}{2}q_N^2(\mathbf{s},\mathbf{s}'). \qquad (11)$$

The landscape $\mathcal{E}_0(\mathbf{s})$ corresponds to a well-known model in the statistical physics of disordered systems, the "pure spherical $p$-spin model" with $p = 2$. It is a variation of the Sherrington-Kirkpatrick model in which the spins are not hard ($s_i = \pm 1$) but soft variables ($s_i \in \mathbb{R}$). This model has been introduced in [14] and since then it has been studied extensively, as it provides a solvable mean-field model for dynamical phenomena such as coarsening [15, 16]. From (11) one sees that the average of the random landscape is independent of the configuration $\mathbf{s}$, and the covariance depends on $\mathbf{s}, \mathbf{s}'$ only through their overlap $q_N(\mathbf{s}, \mathbf{s}')$: in other words, the statistics of the random field is isotropic, rotationally invariant. It is natural to expect that in this limit, given the independence of the landscape on $\mathbf{v}$, the ground state typically exhibits no correlations with $\mathbf{v}$ and it behaves as a "random vector" with respect to it. Therefore, one expects that typically $q_N(\mathbf{s}_{\mathrm{GS}}, \mathbf{v}) \overset{N \to \infty}{\longrightarrow} 0$ when $r = 0$, as we show explicitly in the Box [B2]: the ground state is at the equator, uninformative of the signal. In fact, the isotropy of the landscape statistics implies that typically $q_N(\mathbf{s}_{\mathrm{GS}}, \mathbf{v}) \sim N^{-1/2}$, which is the typical overlap between two vectors extracted randomly with a uniform measure on the hypersphere $\mathcal{S}_N(\sqrt{N})$.

- $\underline{\sigma \to 0.}$ In this limit, the landscape is a deterministic convex function with minima attained exactly at $\mathbf{s}_{\mathrm{GS}} = \pm \mathbf{v}$: the maximum likelihood estimator is fully informative of the signal.

For generic values of $r/\sigma$, the two terms in (8) are in competition: the term proportional to $r$ favors energetically configurations in the vicinity of the signal, while the fluctuating term favors configurations that are independent of the signal and, typically, orthogonal to it on the hypersphere. Moreover, while the term depending on the signal is convex, the randomness tends to generate a landscape with a more complicated structure. In the large-dimensional limit $N \to \infty$, this competition will give rise to sharp transitions when tuning the signal-to-noise ratio $r/\sigma$.

---

**[B2] High-dimensional geometry: typical overlaps and isotropy.** When $r = 0$, the random energy landscape (8) is independent of $\mathbf{v}$: the ground state configuration $\mathbf{s}_{\mathrm{GS}}$ is a random vector with respect to $\mathbf{v}$, and the average (and typical) value of the overlap $q_N(\mathbf{s}_{\mathrm{GS}}, \mathbf{v})$ can be computed assuming that $\mathbf{s}_{\mathrm{GS}}$ is a vector extracted randomly from $\mathcal{S}_N(\sqrt{N})$, with uniform measure. Let us then compute the average of the squared overlap over random vectors:

$$\mathbb{E}_{\mathbf{s}}\left[\left(\frac{\mathbf{s} \cdot \mathbf{v}}{N}\right)^2\right] := \frac{1}{|\mathcal{S}_N(\sqrt{N})|} \int_{\mathcal{S}_N(\sqrt{N})} d\mathbf{s} \left(\frac{\mathbf{s} \cdot \mathbf{v}}{N}\right)^2. \tag{12}$$

Since the overlap is rotationally invariant and the manifold we are integrating over as well, one can rotate the reference frame and choose a new basis $\hat{\mathbf{e}}_i$ in $\mathbb{R}^N$ in such a way that $\mathbf{v}$ coincides with one of the vectors, $\mathbf{v} = \sqrt{N}\hat{\mathbf{e}}_1 = \sqrt{N}(1, 0, \ldots, 0)$. Then

$$\mathbb{E}_{\mathbf{s}}\left[\left(\frac{\mathbf{s} \cdot \mathbf{v}}{N}\right)^2\right] = \frac{1}{N}\mathbb{E}_{\mathbf{s}}\left[s_1^2\right]. \tag{13}$$

By rotational invariance, we could have chosen $\mathbf{v}$ to be an arbitrary basis vectors, meaning that statistically all the components of $\mathbf{s}$ are equivalent and $\mathbb{E}_{\mathbf{s}}\left[s_1^2\right] = \mathbb{E}_{\mathbf{s}}\left[s_2^2\right] = \cdots \mathbb{E}_{\mathbf{s}}\left[s_N^2\right] := \mathbb{E}_{\mathbf{s}}\left[s^2\right]$. The normalization then imposes

$$\sum_{i=1}^{N}\mathbb{E}_{\mathbf{s}}\left[s_i^2\right] = N\mathbb{E}_{\mathbf{s}}\left[s^2\right] = N \longrightarrow \mathbb{E}_{\mathbf{s}}\left[s^2\right] = 1, \tag{14}$$

from which it follows that

$$\mathbb{E}_{\mathbf{s}}\left[\left(\frac{\mathbf{s} \cdot \mathbf{v}}{N}\right)^2\right] = \frac{1}{N}\mathbb{E}_{\mathbf{s}}\left[s^2\right] = \frac{1}{N} \overset{N \to \infty}{\longrightarrow} 0. \tag{15}$$

---

In other words, two vectors extracted randomly on the high-dimensional hypersphere are typically orthogonal. From (15), one also sees an important consequence of *isotropy*: given a basis $\hat{\mathbf{e}}_i$, any vector that is uncorrelated to the basis is such that all its components $s_i = \mathbf{s}\cdot\hat{\mathbf{e}}_i$ are statistically equivalent, and of the same order of magnitude. As we shall see in these notes (see e.g. Box [B4]), this is connected to notions such as *delocalization* and *freeness*. We also make use of this fact in Exercise 2, given in Appendix A.2.

### 2.1.3 Questions and strategy

Let us now refine the definition of the problem, by rephrasing the general questions outlined in Sec. 1 from the perspective of inference. In the low-rank matrix estimation problem, the signal plays the role of a special point in the configuration space $\mathcal{S}_N(\sqrt{N})$, and thus questions about "the position" of the ground state and of the metastable states in configuration space naturally refer to their position with respect to the signal, measured by the overlap $q_N(\mathbf{s},\mathbf{v})$. Moreover, this problem has a relevant parameter, the signal-to-noise ratio $r/\sigma$, and one can inspect how the landscape changes when tuning it. We focus on the following three questions.

Q1. **Signal recovery with maximum likelihood (*i.e., equilibrium*).** This is a question about the optimizer(s) of the landscape. For which values of $r/\sigma$ does $\mathbf{s}_{\text{MLE}} = \mathbf{s}_{\text{GS}}$ provide information about the position of the unknown signal $\mathbf{v}$? In other words, for which signal-to-noise ratio values does the following hold:

$$q_\infty(\mathbf{s}_{\text{GS}},\mathbf{v}) := \lim_{N\to\infty} q_N(\mathbf{s}_{\text{GS}},\mathbf{v}) > 0, \tag{16}$$

implying that the inference problem can, in principle, be solved by optimizing the landscape, as the optimizer carries some information about the unknown signal? This question can be formulated in terms of equilibrium statistical mechanics, by computing properties of the Boltzmann measure

$$p_\beta(\mathbf{s}) = \frac{e^{-\beta\mathcal{E}_r(\mathbf{s})}}{\mathcal{Z}_\beta}, \qquad \mathcal{Z}_\beta = \int_{\mathcal{S}_N} d\mathbf{s}\, e^{-\beta\mathcal{E}_r(\mathbf{s})}, \tag{17}$$

in the limit $\beta \to \infty$, when the measure collapses to the ground state(s) configuration(s). In particular, the overlap (16) measures the equilibrium magnetization (at $\beta \to \infty$) along the direction identified by $\mathbf{v}$, which plays the role of a generalized magnetic field.

Q2. **Landscape topology and geometry (*i.e., metastability*).** This is a question about the landscape's structure. Is the energy landscape $\mathcal{E}_r(\mathbf{s})$ rugged, meaning that there are exponentially-many local minima? How suboptimal are they in terms of likelihood, i.e., what is their energy distribution? Do they provide information about the signal, i.e., what is their distribution on the hypersphere $\mathcal{S}_N$, and specifically, what is their overlap with $\mathbf{v}$?

Q3. **Algorithmic optimization with gradient descent (*i.e., dynamics*).** This is a question about optimization dynamics. Is the optimization problem (with local optimization algorithms) a hard problem, meaning that the search for $\mathbf{s}_{\text{GS}}$ requires exponentially large timescales (in $N$) to be successful? In particular, we focus here on stochastic optimization dynamics of the form

$$\frac{d\mathbf{s}(t)}{dt} = -\boldsymbol{\nabla}_\perp \mathcal{E}_r(\mathbf{s}) + \sqrt{\frac{2}{\beta}}\boldsymbol{\eta}_\perp(t), \quad \mathbb{E}\big[\boldsymbol{\eta}_\perp(t)\big] = 0, \quad \mathbb{E}\big[\boldsymbol{\eta}_\perp(t)\boldsymbol{\eta}_\perp(t')\big] = I\,\delta(t-t'), \tag{18}$$

where $\boldsymbol{I}$ denotes the identity matrix. The first term in the right-hand side of the dynamical equation corresponds to *gradient descent*: the configuration evolves following the direction of steepest descent of the energy landscape (here, $\boldsymbol{\nabla}_\perp$ denotes the gradient of the function restricted to the hypersphere $\mathcal{S}_N(\sqrt{N})$, see Box [B3] for a precise definition). The second term corresponds to the stochasticity, and it is given by Gaussian white noise which perturbs randomly the configuration while keeping it on the hypersphere. The strength of the stochasticity depends on $\beta$: systems that equilibrate (at sufficiently large times) under this dynamics visit configurations with a frequency set by the Boltzmann measure (17) with the same $\beta$ that controls the strength of the stochasticity. In particular, equilibrating at $\beta \to \infty$ corresponds to converging to the ground state(s), i.e., to solving the inference problem.

These questions address different aspects of the problem. Q1 asks when the inference problem is *theoretically solvable*. In other words, under what conditions does optimizing the landscape result in an optimizer that provides information about the signal? To answer this, one studies the problem in theory by assuming knowledge of $\mathbf{v}$, calculating the properties of the optimizer $\mathbf{s}_{\mathrm{GS}}$, and identifying the conditions under which (16) is typically satisfied. Q2 and Q3, on the other hand, concern whether finding the ground state is *practically achievable* using algorithms that rely on local information on the landscape, such as gradient descent. These two questions are closely connected: it is reasonable to expect that optimization will be difficult when the landscape is rugged, as local optimization algorithms are likely to get stuck in local minima, preventing the convergence to the ground state(s).

**The strategy: Statistical physics of stationary points.** In fact, all the above questions concern some special configurations $\mathbf{s}^*$, that are the *stationary points* (local minima, maxima, saddles) of the landscape, satisfying $\boldsymbol{\nabla}_\perp \mathcal{E}_r(\mathbf{s}^*) = 0$. Our strategy to address these questions is then to study the *typical distribution* of such stationary points in terms of their properties, such as:

(i) their energy density $\epsilon_N(\mathbf{s}^*) := \frac{\mathcal{E}_r(\mathbf{s}^*)}{N}$;

(ii) their linear stability, i.e., whether they are minima, saddles or maxima. This information is encoded in the (Riemannian) Hessian matrices $\boldsymbol{\nabla}_\perp^2 \mathcal{E}_r(\mathbf{s}^*)$ of the landscape evaluated at the stationary point, see Box [B3] for their precise definition. The eigenvalues of the Hessian give the curvature of the landscape in the vicinity of the stationary point. We define the index of a stationary point as

$$\kappa_N(\mathbf{s}^*) := \# \left\{ \text{negative eigenvalues of } \boldsymbol{\nabla}_\perp^2 \mathcal{E}_r(\mathbf{s}^*) \right\}.$$

In this definition, local minima have index $\kappa_N = 0$, maxima have index $N - 1$, and all the values of $\kappa$ in between correspond to saddles of the landscape;

(iii) their geometry, i.e., their position on the hypersphere with respect to the signal, measured by the overlap $q_N(\mathbf{s}^*, \mathbf{v})$.

The statistical physics framework enables us to determine the distribution of stationary points for the most probable realizations of the landscape (this is what we mean with *typical distribution*) in the limit $N \to \infty$. We shall see how this information allows us to address the questions above. To get there, we need to introduce one of the main ingredients for computing this distribution for random quadratic landscapes: Random Matrix Theory (RMT).

## 2.2 How: Random matrix theory

### 2.2.1 From landscapes back to random matrices

Consider a fixed realization of $\mathbf{M}$, which generates a fixed realization of the landscape $\mathcal{E}_r(\mathbf{s})$. To find the stationary points on the hypersphere $\mathcal{S}_N(\sqrt{N})$, it is convenient to consider those of the function:

$$\mathcal{E}_r(\mathbf{s}; \lambda) = -\frac{1}{2} \sum_{i,j} M_{ij} s_i s_j + \frac{\lambda}{2} \left( \sum_i s_i^2 - N \right), \quad \mathbf{s} \in \mathbb{R}^N. \tag{19}$$

This function is defined on $\mathbb{R}^N$, and the second term is introduced to implement the spherical constraint $\sum_{i=1}^N s_i^2 = N$, with a Lagrange multiplier $\lambda$ that has to be optimized over as well. When evaluated at configurations on the hypersphere, the two functions (19) and (8) coincide. Stationary points of (19) are pairs $(\mathbf{s}^*, \lambda^*)$ satisfying (we use the symmetry of $\mathbf{M}$):

$$\frac{\partial \mathcal{E}_r(\mathbf{s}; \lambda)}{\partial s_i} = -\sum_j M_{ij} s_j + \lambda s_i \Big|_{\mathbf{s}^*, \lambda^*} = 0,$$

$$\frac{\partial \mathcal{E}_r(\mathbf{s}; \lambda)}{\partial \lambda} = \frac{1}{2} \left( \sum_i s_i^2 - N \Big|_{\mathbf{s}^*, \lambda^*} \right) = 0. \tag{20}$$

Multiplying the first equation by $s_i$, summing over $i$ and using the second equation, we obtain:

$$\sum_{ij} M_{ij} s_i^* s_j^* = \lambda^* N \quad \Longrightarrow \quad \lambda^* = \frac{\mathbf{s}^* \cdot \mathbf{M} \cdot \mathbf{s}^*}{N}, \tag{21}$$

which fixes the value of the Lagrange multiplier $\lambda^* = \lambda(\mathbf{s}^*)$ as a function of the configuration $\mathbf{s}^*$. The first equation in (20) is linear, and it is an eigenvalue equation for $\mathbf{M}$, $\mathbf{M}\mathbf{s}^* = \lambda^* \mathbf{s}^*$. It immediately follows that if $\mathbf{u}^\alpha$ with $\alpha = 1, \dots, N$ are unit-norm eigenvectors of $\mathbf{M}$ with eigenvalues $\lambda^\alpha$, then $\mathbf{s}_\alpha^\pm = \pm\sqrt{N}\mathbf{u}^\alpha$ are stationary points of both $\mathcal{E}_r(\mathbf{s}; \lambda)$ and $\mathcal{E}_r(\mathbf{s})$ (notice the symmetry in the sign due to the quadratic nature of the function). Therefore, the total number of stationary points of this quadratic landscape is constrained to be equal to $2N$ for any realization of the random landscape, due to the linearity of the underlying equations. What are the properties of these stationary points?

(i) Energy density. Inspecting Eq. (21), one sees that the Lagrange multiplier is proportional to the quadratic form that defines the energy landscape, $\lambda^* = -2\mathcal{E}_r(\mathbf{s}^*)/N$. Therefore, the stationary points $\mathbf{s}_\alpha^\pm$ associated to the eigenvector with eigenvalue $\lambda^\alpha$ have energy density fixed by their eigenvalue,

$$\epsilon_N(\mathbf{s}_\alpha^\pm) = -\frac{\lambda^\alpha}{2}. \tag{22}$$

(ii) Linear stability. The Hessian associated to (19) is an $N \times N$ matrix with components:

$$[\boldsymbol{\nabla}^2 \mathcal{E}_r(\mathbf{s}; \lambda)]_{ij} = \frac{\partial^2 \mathcal{E}_r(\mathbf{s}; \lambda)}{\partial s_i \partial s_j} = -M_{ij} + \lambda \delta_{ij}. \tag{23}$$

For general $\lambda$, the eigenvalues of this matrix are $\lambda - \lambda^\beta$ for $\beta = 1, \dots, N$. When evaluated at a stationary point $\mathbf{s}_\alpha^\pm$, the matrix has eigenvalues $\lambda^\alpha - \lambda^\beta$. The Hessian $\boldsymbol{\nabla}^2 \mathcal{E}_r(\mathbf{s}; \lambda)$ at each stationary point has therefore at least one zero mode, which is generated by the spherical constraint: the remaining $N-1$ eigenvalues coincide with the eigenvalues of the Riemannian Hessian $\boldsymbol{\nabla}_\perp^2 \mathcal{E}_r(\mathbf{s})$, see Box [B3]. Assuming that the eigenvalues are

labeled in such a way that $\lambda^1 \leq \lambda^2 \cdots \leq \lambda^N$, then the index is the number of eigenvalues $\lambda^\beta$ that are larger than $\lambda^\alpha$, meaning that:

$$\kappa_N(\mathbf{s}_\alpha^\pm) = N - \alpha. \tag{24}$$

The ground states correspond to $\alpha = N$, $\mathbf{s}_{\text{GS}} = \mathbf{s}_N^\pm = \pm\sqrt{N}\mathbf{u}^N$: they are the stationary points associated to the lowest energy density (22), and they have index zero, consistently with the fact that they are minima.

(iii) **Geometry.** This is encoded in the properties of the eigenvectors, in particular in the squared overlaps

$$q_N^2(\mathbf{s}_\alpha^\pm, \mathbf{v}) = \left(\frac{\mathbf{u}^\alpha \cdot \mathbf{v}}{\sqrt{N}}\right)^2. \tag{25}$$

This simple analysis at fixed randomness (that is, realization of the landscape) already shows us that this quadratic landscape is not rugged in the sense defined in Sec. 1, since the total number of its stationary points grows with the size $N$ only linearly and not exponentially; moreover, their properties are fully encoded in the spectral properties of the matrix $\mathbf{M}$. To characterize these properties statistically, therefore, one has to study the statistics of the eigenvalues and eigenvectors of these type of matrices. RMT comes in handy for this.

---

**[B3] Riemannian Hessian and Lagrange multipliers.**

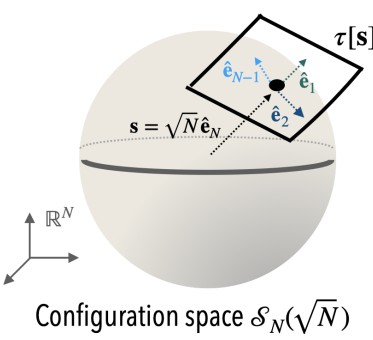

Configuration space $\mathcal{S}_N(\sqrt{N})$

Figure 3: The tangent plane $\tau[\mathbf{s}]$ to the sphere at $\mathbf{s}$ is spanned by $\hat{\mathbf{e}}_\alpha$ with $\alpha = 1, \ldots, N-1$. The vector $\hat{\mathbf{e}}_N$ completes the basis of $\mathbb{R}^N$.

Let us define the Riemannian gradient and Hessian $\nabla_\perp \mathcal{E}_r(\mathbf{s})$, $\nabla_\perp^2 \mathcal{E}_r(\mathbf{s})$ of the function $\mathcal{E}_r(\mathbf{s})$ defined on the hypersphere $\mathcal{S}_N(\sqrt{N})$. The unconstrained gradient (that of the function defined on the whole space $\mathbb{R}^N$) is an $N$-dimensional vector with components $[\nabla \mathcal{E}_r(\mathbf{s})]_i = \partial \mathcal{E}_r(\mathbf{s})/\partial s_i$ with respect to the chosen basis, while the unconstrained Hessian is an $N \times N$ matrix with components $[\nabla^2 \mathcal{E}_r(\mathbf{s})]_{ij} = \partial^2 \mathcal{E}_r(\mathbf{s})/\partial s_i \partial s_j$. The Riemannian gradient on the hypersphere is an $(N-1)$-dimensional vector obtained subtracting from $\nabla \mathcal{E}_r(\mathbf{s})$ the radial component, parallel to $\mathbf{s}$, since the function restricted to $\mathcal{S}_N(\sqrt{N})$ does not vary in that direction. In other words, $\nabla_\perp \mathcal{E}_r(\mathbf{s})$ is the projection of $\nabla \mathcal{E}_r(\mathbf{s})$ on the tangent plane $\tau[\mathbf{s}]$ to the hypersphere at the point $\mathbf{s}$. The latter is the $(N-1)$-dimensional plane spanned by orthonormal vectors $\hat{\mathbf{e}}_\alpha(\mathbf{s})$ with $\alpha = 1, \ldots, N-1$ such that $\hat{\mathbf{e}}_\alpha(\mathbf{s}) \perp \mathbf{s}$. A convenient way to implement this subtraction is to impose the spherical constraint using a Lagrange multiplier (see Eq. (19) in the main text). One considers the extension of the function $\mathcal{E}_r(\mathbf{s})$ to $\mathbf{s} \in \mathbb{R}^N$, and defines $\mathcal{E}_r(\mathbf{s}; \lambda) := \mathcal{E}_r(\mathbf{s}) + \frac{\lambda}{2}\left(\sum_i s_i^2 - N\right)$. The spherical constraint is imposed optimizing this function with respect to the Lagrange multiplier $\lambda$. The latter accounts for the subtraction of the radial component to the gradient. In fact, one has $\nabla \mathcal{E}_r(\mathbf{s}; \lambda) = \nabla \mathcal{E}_r(\mathbf{s}) + \lambda \mathbf{s}$, which implies (setting it to zero and implementing the spherical

constraint) that

$$\lambda = -\frac{\nabla \mathcal{E}_r(\mathbf{s}) \cdot \mathbf{s}}{N}. \tag{26}$$

Therefore,

$$\nabla \mathcal{E}_r(\mathbf{s}; \lambda) = \nabla \mathcal{E}_r(\mathbf{s}) - \left( \frac{\nabla \mathcal{E}_r(\mathbf{s}) \cdot \mathbf{s}}{N} \right) \mathbf{s}. \tag{27}$$

By completing the basis $\hat{\mathbf{e}}_\alpha(\mathbf{s})$ of $\tau[\mathbf{s}]$ with the radial vector $\hat{\mathbf{e}}_N(\mathbf{s}) = \mathbf{s}/\sqrt{N}$ to get a basis of $\mathbb{R}^N$, we see that in this new basis

$$\nabla \mathcal{E}_r(\mathbf{s}; \lambda) = \begin{pmatrix} \nabla_\perp \mathcal{E}_r(\mathbf{s}) \\ 0 \end{pmatrix}. \tag{28}$$

In this basis, the last component of the vector corresponds to the radial direction $\hat{\mathbf{e}}_N(\mathbf{s}) = \mathbf{s}/\sqrt{N}$: the fact that it vanishes follows from (27), which in turn follows from imposing that $\mathbf{s}$ lies on the hypersphere. The projection of $\nabla \mathcal{E}_r(\mathbf{s}; \lambda)$ on $\tau[\mathbf{s}]$ is obtained neglecting that vanishing component. In a similar way, the $N \times N$ Hessian reads:

$$\nabla^2 \mathcal{E}_r(\mathbf{s}; \lambda) = \nabla^2 \mathcal{E}_r(\mathbf{s}) + \lambda \mathbf{I} = \nabla^2 \mathcal{E}_r(\mathbf{s}) - \left( \frac{\nabla \mathcal{E}_r(\mathbf{s}) \cdot \mathbf{s}}{N} \right) \mathbf{I}. \tag{29}$$

The Riemannian Hessian $\nabla_\perp^2 \mathcal{E}_r(\mathbf{s})$ is obtained projecting (29) onto $\tau[\mathbf{s}]$: if the matrix is expressed in the extended basis $\hat{\mathbf{e}}_\alpha(\mathbf{s})$, the projection amounts to neglect the last row and column. Notice that, because $\mathcal{E}_r(\mathbf{s})$ is homogeneous, the diagonal shift is $\nabla \mathcal{E}_r(\mathbf{s}) \cdot \mathbf{s} = \mathbf{s} \cdot \nabla^2 \mathcal{E}_r(\mathbf{s}) \cdot \mathbf{s}$, and thus in the extended basis $\hat{\mathbf{e}}_\alpha(\mathbf{s})$ the term due to the Lagrange multiplier is subtracting the diagonal element $\hat{\mathbf{e}}_N(\mathbf{s}) \cdot \nabla^2 \mathcal{E}_r(\mathbf{s}) \cdot \hat{\mathbf{e}}_N(\mathbf{s})$; moreover, since $\nabla^2 \mathcal{E}_r(\mathbf{s}) \cdot \mathbf{s} = \nabla \mathcal{E}_r(\mathbf{s})$, at a stationary point the last line and column of the Hessian in the extended basis $\hat{\mathbf{e}}_\alpha(\mathbf{s})$ are also zero. This implies that $\nabla^2 \mathcal{E}_r(\mathbf{s}; \lambda)$ has one eigenvalue that is zero, which is neglected when projecting to the tangent plane to get $\nabla_\perp^2 \mathcal{E}_r(\mathbf{s})$.

### 2.2.2 Spiked GOE matrices: A detour into basic facts

In this Section, we recap some notions of RMT which turn out to be relevant for the inference problem. Though motivated by this inference problem, this Section is self-contained and can be read independently. The derivation of some statements is given in the form of guided exercises, which can be found in the Appendices.

Our focus are the spectral properties of spiked GOE random matrices of the form

$$\mathbf{M} = \mathbf{J} + \mathbf{R} = \mathbf{J} + r \, \mathbf{w}\mathbf{w}^T, \qquad \mathbf{J} \in \text{GOE}(\sigma^2), \qquad ||\mathbf{w}||^2 = 1, \qquad r > 0, \tag{30}$$

where:

- $\mathbf{J}$ is a matrix extracted from the Gaussian Orthogonal Ensemble (GOE): it is symmetric, and its entries $J_{i \leq j}$ are independent Gaussian variables with moments given in (2). Realization of matrices extracted from this ensemble thus occur with the probability (3). The variance of the entries is scaled with $N$ in such a way that the eigenvalues of the matrix are typically of $O(1)$, meaning that they lie on an interval in the real axis whose width remains bounded when $N \to \infty$. The GOE ensemble is an example of invariant ensemble: the distribution of these matrices is invariant with respect to orthogonal transformations, as we discussed around Eq. (4). Moreover, GOE matrices are also an example of Wigner matrices, the latter being random matrices which are real, symmetric and with independent, identically distributed entries (not necessarily Gaussian) [17].

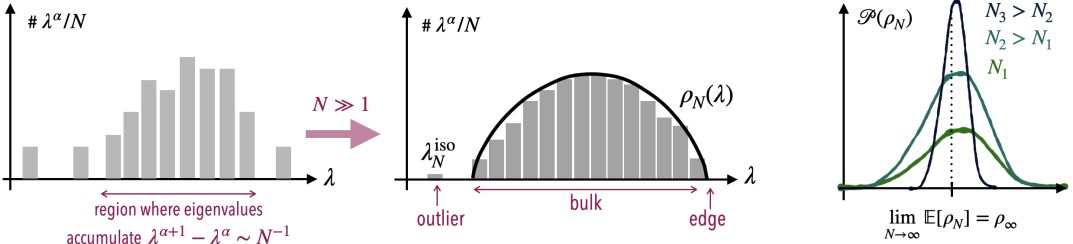

Figure 4: *Left.* Sketch of the distribution of eigenvalues $\lambda^\alpha$ for increasing size of the matrix $N$: as $N$ increases, the eigenvalues accumulate and their distribution converges to a continuous limit described by a density $\rho_N$. Eigenvalues that do not belong to the support of the density are called isolated or outliers. *Right.* Sketch of the distribution of a self-averaging random variable: as $N$ grows, the width of the distribution shrinks around the average value, and the distribution collapses to it in the limit $N \to \infty$.

- $\mathbf{R} = r\mathbf{w}\mathbf{w}^T$ is a deterministic rank-one matrix with one eigenvalue equal to $r$ and $N-1$ null eigenvalues. Being almost vanishing, this term represents a *perturbation* to the GOE matrix $\mathbf{J}$: this is why matrices of the form (30) are also referred to as rank-one perturbed GOE matrices. To connect to the denoising problem introduced in the previous section, it suffices to identify $\mathbf{v} = \sqrt{N}\mathbf{w}$.

We denote with $\lambda^\alpha, \mathbf{u}^\alpha$ the eigenvalues and eigenvectors of the matrix $\mathbf{M}$, for $\alpha = 1, \ldots, N$. We assume the ordering $\lambda^1 \leq \cdots \leq \lambda^N$, and $||\mathbf{u}^\alpha|| = 1$. Moreover, we set $q^\alpha = \mathbf{u}^\alpha \cdot \mathbf{w}$. We will pay particular attention to the statistics of the maximal eigenvalue and eigenvector of this matrix, and to their scaling with the size $N$; for notational purposes, we set

$$\lambda_N^{\max} := \lambda^N, \qquad \mathbf{u}_N^{\max} := \mathbf{u}^N, \qquad q_N^{\max} := \mathbf{u}^N \cdot \mathbf{w}, \qquad \xi_N^{\max} := [q_N^{\max}]^2 = \left(\mathbf{u}^N \cdot \mathbf{w}\right)^2, \quad (31)$$

where the subscript is now used to indicate the size of the matrix $\mathbf{M}$. Throughout this Section, the expectation $\mathbb{E}[\cdot]$ denotes the average with respect to the distribution of the matrix $\mathbf{M}$. Even though we focus specifically on the spectral properties of matrices of the form (30), some of the results reported in this Section exhibit a certain degree of universality, meaning they can be generalized to cases in which $\mathbf{J}$ is extracted from ensembles different with respect to the GOE (either invariant or Wigner), or to cases in which $\mathbf{R}$ is a perturbation of rank higher than one (not scaling with $N$, or growing sufficiently slow with $N$). In these notes, we do not state results under the most general conditions: readers interested in such generalizations are referred to the specific references cited when each result is presented. Let us begin with introducing some key concepts and terminology.

**Eigenvalue distribution.** In RMT, one is in general interested in describing properties of the spectrum of the matrices $\mathbf{M}$ in the limit of large matrix size, $N \to \infty$. A central quantity encoding information on the spectrum is the *eigenvalue distribution*, which for finite $N$ and for given symmetric $\mathbf{M}$ with eigenvalues $\lambda^\alpha$ can be written in the form

$$\nu_N(\lambda) = \frac{1}{N} \sum_{\alpha=1}^N \delta(\lambda - \lambda^\alpha), \quad \lambda \in \mathbb{R}. \qquad (32)$$

As the name suggests, this is a distribution: it allows to compute averages over the eigenvalues of the matrix, such as

$$\frac{1}{N}\text{Tr}[f(\mathbf{M})] = \frac{1}{N} \sum_{\alpha=1}^N f(\lambda^\alpha) = \int_{\mathbb{R}} d\nu_N(\lambda) f(\lambda). \qquad (33)$$

**Eigenvalue density and isolated eigenvalues.** For fixed matrix $\mathbf{M}$ and finite $N$, the eigenvalue distribution is a collection of discrete delta peaks on the real axis. The general scenario as the size $N$ increases is depicted in Fig. 4 (*Left*). There is a region on the real axis where eigenvalues accumulate: the gaps between them vanish when $N \to \infty$, in general as $\lambda^\alpha - \lambda^{\alpha-1} \sim N^{-1}$, and their number in an interval $\Delta\lambda = O(1)$ becomes of order $N$. In this region, the distribution of eigenvalues when $N \gg 1$ is well-described by a continuous *density function* $\rho_N(\lambda)$. In addition to this, there might be regions where instead there are eigenvalues that do not accumulate but remain *isolated*. In general, the eigenvalue distribution for large $N$ can therefore be decomposed as

$$d\,\nu_N(\lambda) \stackrel{N \gg 1}{\approx} \rho_N(\lambda)d\lambda + \frac{1}{N}\sum_i \delta(\lambda - \lambda_N^{\text{iso,i}})d\lambda\,, \tag{34}$$

where $\lambda_N^{\text{iso,i}}$ for $i = 1, 2, \ldots$ denote the isolated eigenvalues, also referred to as *outliers*. For the matrices (30), the outlier, whenever it exists, it is "generated" by the rank-one perturbation, as we discuss in more detail below.

**Concentration when $N \to \infty$.** When $\mathbf{M}$ is random, quantities like $\rho_N(\lambda)$ and $\lambda_N^{\text{iso,i}}$ are *distributed*. However, for random matrix ensembles in general these quantities *concentrate* around their *typical value* when $N \to \infty$ [18]. This means that their distributions shrink and asymptotically collapse to a deterministic value, which is the typical (and average) value of these quantities when $N \to \infty$, see Fig. 4 (*Right*) for a sketch. In formulas, one has

$$\lim_{N \to \infty} \rho_N(\lambda) = \rho_\infty(\lambda) = \lim_{N \to \infty} \mathbb{E}[\rho_N(\lambda)]\,, \tag{35}$$

where we stress that in this equation $\rho_N(\lambda)$ is a random function with a distribution, while the asymptotic limit $\rho_\infty(\lambda)$ is a deterministic function. Since the large-$N$ limit of the random variable coincides with the limit of its average, in the language of statistical physics this quantity is said to be *self-averaging* (the terminology refers to the fact that one single, sufficiently large instance of the matrix is representative of the average behavior). A similar concentration property holds also for isolated eigenvalues of the matrix $\mathbf{M}$, whenever they exist: one can show that

$$\lim_{N \to \infty} \lambda_N^{\text{iso,i}} = \lambda_\infty^{\text{iso,i}}\,, \tag{36}$$

where the quantity $\lambda_\infty^{\text{iso,i}}$ is a deterministic value.

**Typical values, fluctuations, large deviations.** While when $N \to \infty$ the eigenvalue density and the isolated eigenvalues converge to a deterministic limit (their typical value), for finite $N$ fluctuations are present and these quantities are distributed. In the remainder of this Section we first discuss results on the typical values, and subsequently characterize the size of the fluctuations of these quantities when $N \gg 1$ is large but finite, showing how they decay when $N \to \infty$. Finally, we discuss large deviations, which control the probability to see $O(1)$ deviations from the typical values when $N \gg 1$. Of course, by definition of typical value, this probability decays to zero when $N \to \infty$, in general exponentially fast. Large deviations thus characterize *rare events*.

■ **Typical values: The eigenvalue density.** The eigenvalue density can be obtained from the *Stieltjes transform* of the matrix $\mathbf{M}$:

$$\mathfrak{g}_N(z) := \frac{1}{N}\text{Tr}[\mathbf{G_M}(z)] = \frac{1}{N}\sum_{\alpha=1}^{N}\frac{1}{z-\lambda^\alpha} = \int_{\mathbb{R}}\frac{d\,\nu_N(\lambda)}{z-\lambda}\,, \quad \mathbf{G_M}(z) := (z\mathbf{I} - \mathbf{M})^{-1}, \quad z \in \mathbb{C}^-, \quad (37)$$

where $\mathbf{G_M}(z) = (z\mathbf{I} - \mathbf{M})^{-1}$ is the *resolvent* of the matrix $\mathbf{M}$, and $\mathbb{C}^-$ denotes the lower-half complex plane, $z = \lambda - i\eta$ with $\eta > 0$. The function (37) is singular (it has poles) when $z$ approaches the eigenvalues of the matrix, which lie on the real axis: to avoid these singularities, we had defined it on the lower-half complex plane. The goal is to analytically continue this function to the real axis and to characterize its singular behavior, since such behavior carries information on the spectrum of $\mathbf{M}$.

Similarly to the eigenvalue density, the random function (37) is self-averaging, with limiting value

$$\lim_{N\to\infty} \mathfrak{g}_N(z) = \mathfrak{g}_\infty(z) = \lim_{N\to\infty} \mathbb{E}[\mathfrak{g}_N(z)]. \tag{38}$$

In this limit, in the region on the real axis where the eigenvalues accumulate, the poles of $\mathfrak{g}_N$ also accumulate into a branch cut. The discontinuity of $\mathfrak{g}_\infty(z)$ across the branch cut is related to the eigenvalue density as

$$\rho_\infty(\lambda) = \lim_{\eta\to 0} \frac{1}{\pi} \mathrm{Im}\, \mathfrak{g}_\infty(\lambda - i\eta), \tag{39}$$

as it follows from the Sokhotski-Plemelj theorem. This is the *Stieltjes inversion formula*: It implies that the asymptotic density can be extracted from the large-$N$ limit of the Stieltjes transform.

For the rank-one perturbed GOE matrices (30), the function $\mathfrak{g}_\infty(z)$ can be computed with several techniques, including the *Replica Method*, as done in the early work [19]. The replica formalism allows one to compute the behavior of the averages $\mathbb{E}[\mathfrak{g}_N(z)]$ to leading order in $N$. This calculation is the content of the Exercise 1 given in Appendix A.1. Let us summarize here the main outcomes of that calculation.

(1) The first result is that the finite-rank term $\mathbf{R} = r\,\mathbf{w}\mathbf{w}^T$ does not affect the limiting function $\mathfrak{g}_\infty(z)$ nor the limiting density $\rho_\infty(\lambda)$, which thus coincides with the density for $r = 0$, that is that of the GOE matrix $\mathbf{J}$. Therefore, $\mathbf{R}$ is indeed a perturbation that does not impact the distribution of eigenvalues to leading order in $N$, which is given solely by the eigenvalue density, see Eq. (34). The same result is obtained for perturbations $\mathbf{R}$ of the GOE matrix of higher rank $k = O(N^0)$.

(2) When $\mathbf{J}$ is GOE, its Stieltjes transform satisfies a self-consistent equation:

$$\mathfrak{g}_\infty(z) = \frac{1}{z - \sigma^2 \mathfrak{g}_\infty(z)}, \quad z \in \mathbb{C}^-. \tag{40}$$

Within the replica formalism, this equation is obtained as a result of a *saddle-point calculation*.

(3) The self-consistent equation is solved by the function

$$\mathfrak{g}_\infty(z) \to \mathfrak{g}_{\mathrm{sc},\sigma}(z) = \frac{z - z\sqrt{1 - \frac{4\sigma^2}{z^2}}}{2\sigma^2}, \tag{41}$$

where the sign in front of the square root is chosen to guarantee that $\mathfrak{g}_{\mathrm{sc},\sigma}(z) = 0$ when $|z| \to \infty$. The continuation of this function to the real axis, $z \to \lambda \in \mathbb{R}$, gives

$$\mathfrak{g}_{\mathrm{sc},\sigma}(\lambda) = \frac{\lambda - \mathrm{sign}(\lambda)\sqrt{\lambda^2 - 4\sigma^2}}{2\sigma^2}, \quad \lambda \notin [-2\sigma, 2\sigma]. \tag{42}$$

(4) By the Stieltjes inversion formula, one gets from (41) the eigenvalue density

$$\rho_\infty(\lambda) \to \rho_{\mathrm{sc},\sigma}(\lambda) = \frac{1}{2\pi\sigma^2}\sqrt{4\sigma^2 - \lambda^2}\,\mathbb{1}_{\lambda\in[-2\sigma,2\sigma]}. \tag{43}$$

This density is known as the *Wigner semicircle law*. The limiting density holds for a large class of matrices of the Wigner type (which, we recall, are symmetric with independent identically distributed entries, not necessarily Gaussian) under certain conditions on the distribution of the entries [20, 21]. Also, it emerges in connection to the spectrum of the adjacency matrix of random graphs [22], or to the Burger equation [23].

■ **Typical values: The isolated eigenvalue(s).** The eigenvalue density of the matrices (30) has a compact support in the interval $[-2\sigma, 2\sigma]$, see (43). Isolated eigenvalues lie outside the support of the eigenvalue density (hence the name "outliers"). As it follows from (37) combined with (34), their contribution to the eigenvalues distribution is of order $1/N$, and it is thus sub-leading with respect to that of the density. When isolated eigenvalues exist, their typical values (36) can also be extracted from the Stieltjes transform $\mathfrak{g}_N(z)$, by computing its large-$N$ expansion to order $1/N$. This calculation is the content of the Exercise 2 given in Appendix A.2. We summarize here the main outcomes.

(1) In absence of the perturbation, $r = 0$, there are no isolated eigenvalues. The minimal and maximal eigenvalues are exactly at the edge of the support of the semicircle law, i.e., almost surely

$$\lim_{N \to \infty} \lambda^1 = -2\sigma, \qquad \lim_{N \to \infty} \lambda^N = 2\sigma. \tag{44}$$

(2) For $r > 0$, a sharp transition occurs in the limit $N \to \infty$ at a critical value $r_c(\sigma) = \sigma$: for $r > r_c$, the matrices **M** have a single isolated eigenvalue, that is the largest one $\lambda_N^{\max}$ [14, 19, 24]. More precisely, almost surely,

$$\lambda_\infty^{\max} = \lim_{N \to \infty} \lambda_N^{\max} = \begin{cases} 2\sigma, & \text{if } r \leq r_c = \sigma, \\ \lambda_\infty^{\text{iso}} = \frac{\sigma^2}{r} + r, & \text{if } r > r_c = \sigma. \end{cases} \tag{45}$$

Thus, for all $r \leq r_c$ the spectrum is the same as for $r = 0$, and the largest eigenvalue sticks to the boundary of the support of the semicircle law. For $r > r_c$, the largest eigenvalue is detached from the eigenvalue density and it is isolated. An analogous statement holds for $r < 0$: in that case, the isolated eigenvalue is the minimal one.

(3) When the largest eigenvalue is isolated, the corresponding eigenvector $\mathbf{u}^N$ acquires a macroscopic, i.e. $O(N^0)$, projection on the vector $\mathbf{w}$ defining the rank-one perturbation. Almost surely when $r > 0$:

$$\xi_\infty^{\max} = [q_\infty^{\max}]^2 = \lim_{N \to \infty} \left( \mathbf{u}_N^{\max} \cdot \mathbf{w} \right)^2 = \begin{cases} 0, & \text{if } r \leq r_c = \sigma, \\ 1 - \frac{\sigma^2}{r^2}, & \text{if } r > r_c = \sigma. \end{cases} \tag{46}$$

For all the other eigenvectors $\mathbf{u}^\alpha$ associated to eigenvalues that are not isolated, instead, almost surely $q^\alpha = \mathbf{u}^\alpha \cdot \mathbf{w} = 0$ when $N \to \infty$. The transition (46) corresponds to the breaking of isotropy and it can be seen as a *localization transition*, as we discuss in Box [B4].

(4) These results hold more generically when the random matrix $J$ is extracted from an invariant ensemble (not necessarily the GOE), with an eigenvalue density $\rho_\infty(\lambda)$ compactly supported in an interval $[a, b]$. The transition in this more general case occurs at $r_c = 1/\mathfrak{g}_\infty(b)$ and almost surely (for $r > 0$):

$$\lambda_\infty^{\max} = \lim_{N \to \infty} \lambda_N^{\max} = \begin{cases} b, & \text{if } r \leq r_c = 1/\mathfrak{g}_\infty(b), \\ \lambda_\infty^{\text{iso}} = \mathfrak{g}_\infty^{-1}\left(\frac{1}{r}\right), & \text{if } r > r_c = 1/\mathfrak{g}_\infty(b), \end{cases} \tag{47}$$

and

$$\xi_\infty^{\max} = [q_\infty^{\max}]^2 = \lim_{N\to\infty}(\mathbf{u}^N \cdot \mathbf{w})^2 = \begin{cases} 0, & \text{if } r \leq r_c = 1/\mathfrak{g}_\infty(b), \\ \frac{-1}{r^2 \mathfrak{g}'_\infty(\lambda_\infty^{\text{iso}})}, & \text{if } r > r_c = 1/\mathfrak{g}_\infty(b), \end{cases} \tag{48}$$

where $\mathfrak{g}_\infty^{-1}$ is the inverse of the limiting function (38), and $\mathfrak{g}'_\infty$ its derivative [24,25]. In the GOE case, $\mathfrak{g}_{\text{sc},\sigma}^{-1}(y) = \sigma^2 y + 1/y$, and it is simple to check that from these formulas (45), (46) are recovered.

These results can be generalized to finite-rank perturbations $\mathbf{R}$ of higher rank $k = O(N^0)$: such perturbations can generate $k$ isolated eigenvalues, each one appearing with a transition [24]. As mentioned above, in the derivation of these results the matrix $\mathbf{J}$ can belong to invariant ensembles that are not the GOE [25]; as it appears clear also from solving Exercise 2, it is important that such matrix has a statistics that is independent of that of the finite-rank perturbation $\mathbf{R}$. In the language of free probability, one would say that the two matrices "are free": for the connection between these results and free probability, see [25, 26].

■ **Finite $N$ fluctuations: Small deviations.** The results discussed so far describe the spectral properties of matrices when $N \to \infty$; in this limit, quantities are self-averaging and concentrate around their typical value. For $N$ large but finite, $\rho_N(\lambda)$ and $\lambda_N^{\text{iso}}$ fluctuate from realization to realization of the disorder, and one can ask what becomes of the transition (45) at finite $N$. We summarize here some results available in the literature on this point.

(1) When $N$ is large but finite, the transition at $r = r_c(\sigma)$ becomes a *crossover*: there is a *critical window* controlled by the scaling variables $\omega = N^{1/3}(r - r_c)$, and one can distinguish three different regimes (we assume here $r > 0$):

$$\begin{aligned} \omega \ll -1 &\Longrightarrow (r_c - r) \gg N^{-\frac{1}{3}}, & \text{subcritical}, \\ \omega \sim O(1) &\Longrightarrow r - r_c \sim N^{-\frac{1}{3}}, & \text{critical}, \\ \omega \gg 1 &\Longrightarrow r - r_c \gg N^{-\frac{1}{3}}, & \text{supercritical}. \end{aligned} \tag{49}$$

The scaling of the critical window with $N^{1/3}$ has been first determined in [27] for complex covariance matrices (the Wishart ensemble), and generalized afterwards to other type of random matrices [24, 28].

(2) *Subcritical regime.* When $r \ll r_c$, the fluctuations of the largest eigenvalue $\lambda_N^{\max}$ are described by the *Tracy-Widom* distribution:

$$\lambda_N^{\max} \sim 2\sigma + N^{-\frac{2}{3}}\sigma\zeta_{\text{TW}}, \tag{50}$$

where $\zeta_{\text{TW}}$ is a random variable distributed according to the Tracy-Widom distribution with parameter $\beta = 1$ [29], which we denote with $P_{\text{TW}}$. This equation means that

$$\lim_{N\to\infty} P\left(x = \frac{N^{\frac{2}{3}}(\lambda_N^{\max} - 2\sigma)}{\sigma}\right) = P_{\text{TW}}(x). \tag{51}$$

As for the Wigner semicircle law, also for the Tracy-Widom distribution one can speak about universality, as the latter appears in a huge variety of contexts. Some of these contexts are discussed in [30, 31], see also [32–34] for more recent examples. Because of the emergence of this distribution in the context of growth models

described by the Kardar-Parisi-Zhang (KPZ) equation [35], one speaks about the *KPZ (Kardar-Parisi-Zhang) universality class*. The scaling (50) also shows that the gap between the largest eigenvalues at the edge of the support of the semicircle law is $O(N^{-2/3})$, i.e., $\lambda^N - \lambda^{N-1} \sim O(N^{-2/3})$. This is the same type of scaling that one has in the unperturbed limit $r = 0$, for GOE matrices [36].

(3) *Supercritical regime.* When $r \gg r_c$, the maximal eigenvalue is isolated, $\lambda_N^{\max} = \lambda_N^{\mathrm{iso}}$, and its fluctuations are Gaussian,

$$\lambda_N^{\mathrm{iso}} \sim \left( \frac{\sigma^2}{r} + r \right) + N^{-\frac{1}{2}} \sqrt{2\sigma^2 \left( 1 - \frac{\sigma^2}{r^2} \right)} \, \zeta_{\mathrm{Gauss}} \,, \tag{52}$$

where $\zeta_{\mathrm{Gauss}}$ is a Gaussian random variable with zero mean and unit variance,

$$\lim_{N \to \infty} P \left( x = \frac{N^{\frac{1}{2}} (\lambda_N^{\mathrm{iso}} - \lambda_\infty^{\mathrm{iso}})}{\sqrt{2\sigma^2 \left( 1 - \frac{\sigma^2}{r^2} \right)}} \right) = P_{\mathrm{Gauss}}(x) \,. \tag{53}$$

(4) *Critical regime.* In this case,

$$\lambda_N^{\max} \sim 2\sigma + N^{-\frac{2}{3}} \zeta_\omega \,, \tag{54}$$

where $\zeta_\omega$ is a random variable whose distribution $P_\omega$ depends on the scaling variable $\omega = N^{\frac{1}{3}} (r - r_c)$, and satisfies the matching conditions [28]

$$P_\omega \xrightarrow{\omega \to \infty} P_{\mathrm{Gauss}} \,, \qquad P_\omega \xrightarrow{\omega \to -\infty} P_{\mathrm{TW}} \,. \tag{55}$$

These results clearly illustrate that the transition occurring at $r = r_c$ is not only a transition in the typical value of the maximal (for $r < 0$, minimal) eigenvalue of the rank-one perturbed GOE matrix: it is also a transition in the *scaling* and *nature* of the fluctuations of largest eigenvalue at finite but large $N$. For rank-1 GOE perturbed matrices, the transition of the typical value was first determined in the seminal work [19]. The transition in the scaling of the fluctuations was instead characterized in [27] for Wishart matrices, and it is now referred to generically as the *BBP transition*. A summary of the BBP scenario for rank-one perturbed GOE matrices is given in Figure 5.

■ **Finite $N$ fluctuations: Large deviations.** We conclude this Section devoted to RMT by briefly discussing some results on the large deviations of the eigenvalue density $\rho_N(\lambda)$ and of the maximal eigenvalue $\lambda_N^{\max}$. The results on small deviations show that the maximal eigenvalue has fluctuations that are of $O(N^{-\alpha})$, where $\alpha = 1/2$ in the supercritical regime and $\alpha = 2/3$ in the subcritical regime: at finite but large $N$, one sees deviations from the typical, asymptotic value $\lambda_\infty^{\max}$ that are of that order, and vanish when $N \to \infty$. LDT describes the probability that the largest eigenvalue takes a value $\lambda_N^{\max} = x$ that is of $O(1)$ different from the typical one. For the unperturbed GOE case ($r = 0$), these large deviations have been characterized extensively, see [31] for a review. Depending on whether one is considering the left or right tail of the large deviation function, the *speed* (i.e., how fast the large deviation probability decays with $N$) is different: if $x < 2\sigma$ (left tail), the probability decays exponentially in $N^2$, while for $x > 2\sigma$ (right tail) the probability decays exponentially in $N$. The reason for this discrepancy is that to have $\lambda_N^{\max} = x < 2\sigma$, one needs to *push* below $x$ a finite fraction of the eigenvalues (those that typically lie in the interval $[x, 2\sigma]$ within the support of the semicircle law): the probability for this to occur is extremely small, as this is a deviation from the typical value of an $O(N)$ number of correlated random variables. In fact, in order to have

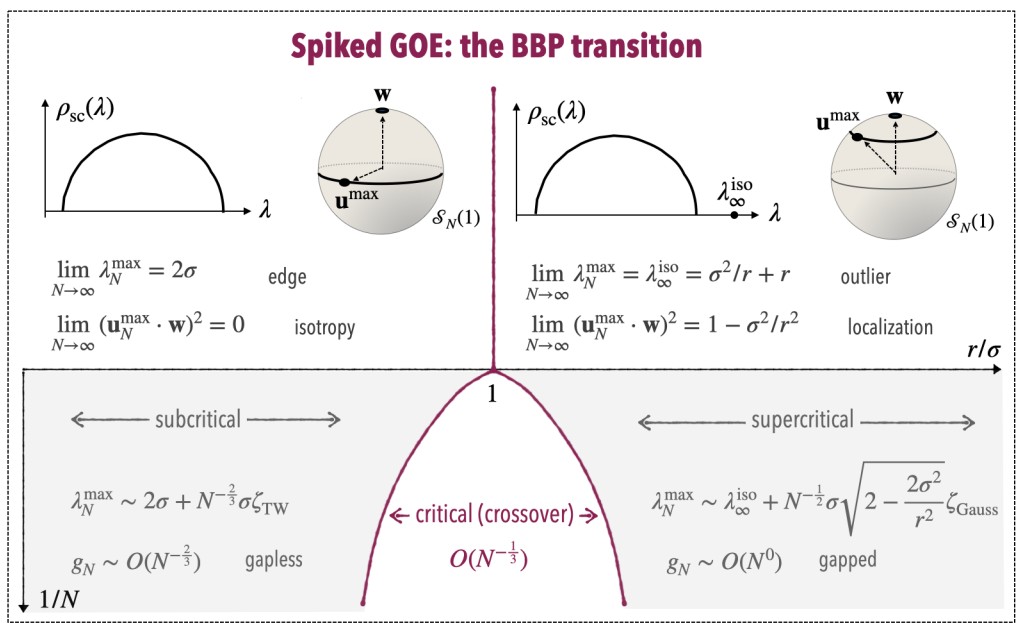

Figure 5: Summary of the main features of the BBP transition for $\mathrm{GOE}(\sigma^2)$ matrices of size $N$ perturbed with a rank-one perturbation of strength $r > 0$ along the direction $\mathbf{w}$. In this figure, $\lambda_N^{\max}$ denotes the maximal eigenvalue of the spiked matrix, $\mathbf{u}_N^{\max}$ the corresponding eigenvector, and $g_N = \lambda^N - \lambda^{N-1}$ the gap between the two largest eigenvalues of the matrix (we use the notations $\lambda_N^{\max}$ and $\lambda^N$ interchangeably for the largest eigenvalue).

$\lambda_N^{\max} = x < 2\sigma$, one needs to enforce a deviation of the full density of eigenvalues with respect to its typical value $\rho_{\mathrm{sc},\sigma}(\lambda)$. Consistently, the speed $N^2$ is the same one controlling the large-$N$ behavior of the distribution of $\rho_N(\lambda)$, which takes the large deviation form:

$$\mathcal{P}_N[\rho]\mathcal{D}\rho = e^{-N^2 \mathcal{S}[\rho]+O(N)}, \tag{56}$$

see [37, 38] for the explicit form of the functional $\mathcal{S}[\rho]$. Of course, the functional $\mathcal{S}[\rho]$ is minimized at the typical value $\rho \equiv \rho_{\mathrm{sc},\sigma}$, where it is equal to zero.

On the other hand, to have $\lambda_N^{\max} = x > 2\sigma$ one only needs to *pull* one single eigenvalue, the maximal one, away from its typical value: these instances happen with a larger probability, which is only exponentially small in $N$. For the subsequent discussion, we are interested in characterizing also the large deviations of the eigenvector component $q_N^{\max}$. For rank-one perturbed GOE matrices, the starting point for determining this probability is the joint distribution of eigenvalues $\lambda^\alpha$ and squared eigenvectors projections $\xi^\alpha := [q^\alpha]^2 = (\mathbf{u}^\alpha \cdot \mathbf{w})^2$, which reads:

$$\mathcal{P}(\{\lambda^\alpha, \xi^\alpha\}_\alpha) = \frac{e^{-N\sum_\alpha f(\lambda^\alpha,\xi^\alpha)}}{A_N} \prod_{\alpha=1}^{N-1} \theta(\lambda^{\alpha+1} - \lambda^\alpha) \prod_{\alpha<\beta} |\lambda^\beta - \lambda^\alpha| \, \delta\left(\sum_\alpha \xi^\alpha - 1\right) \prod_\alpha \frac{1}{\sqrt{\xi^\alpha}}, \tag{57}$$

where $f(\lambda^\alpha, \xi^\alpha) = ([\lambda^\alpha]^2 - 2r\lambda^\alpha\xi^\alpha)/4\sigma^2$, and $A_N$ is a normalization. The $\theta(x)$ appearing in (57) is a step function enforcing the ordering between the eigenvalues that we are assuming,

$$\theta(x) = \begin{cases} 1, & \text{if } x \geq 0, \\ 0, & \text{if } x < 0. \end{cases} \tag{58}$$

The distribution (57) for $r = 0$ is a classic result in RMT [17]. In this limit, $\mathbf{M}$ reduces to the GOE matrix $\mathbf{J}$ and $\mathbf{w}$ is an arbitrary vector of unit norm, independent of $\mathbf{J}$. Eq. (57) describes then the joint distribution of the eigenvalues, and of the eigenvectors' projections onto this arbitrary vector. The form of the joint distribution for $r = 0$ is obtained from (3) performing a change of variables, from the original matrix $\mathbf{J}$ to the pair of matrices $(\mathbf{O}, \mathbf{\Lambda})$, where $\mathbf{\Lambda}$ is the diagonalized version of the matrix $\mathbf{J}$ and $\mathbf{O}$ the corresponding diagonalizing operator, $\mathbf{J} = \mathbf{O}^T \mathbf{\Lambda} \mathbf{O}$. The terms at the exponent in (57) trivially arise from expressing the trace in terms of the new variables, $\mathrm{Tr}\mathbf{J}^2 \to \mathrm{Tr}\mathbf{\Lambda}^2 = \sum_\alpha [\lambda^\alpha]^2$. The remaining terms arise instead from the Jacobian of the change of variables, see [17] for a detailed derivation. When $r > 0$, the distribution of $\mathbf{M} = \mathbf{J} + r\mathbf{w}\mathbf{w}^T$ reads

$$P_N(\mathbf{M})d\mathbf{M} = \frac{1}{\tilde{A}_{N,r}} e^{-\frac{N}{4\sigma^2}\mathrm{Tr}\mathbf{M}^2 + r\frac{N}{2\sigma^2}\mathbf{w}^T\mathbf{M}\mathbf{w}} d\mathbf{M}, \tag{59}$$

where $\tilde{A}_{N,r}$ is a normalization factor. Eq. (57) is recovered performing the same change of variables and following the same steps as for $r = 0$. The $r$-dependent shift at the exponent in (57) arises from the second term in the exponent of (59). Notice that in the GOE $r = 0$ limit, the part of the distribution corresponding to the eigenvalues and that corresponding to the eigenvectors decouple. In particular, the eigenvectors' projections $q^\alpha = \sqrt{\xi^\alpha}$ on the arbitrary vector $\mathbf{w}$ are uniformly distributed up to a normalization condition,

$$\tilde{\mathcal{P}}_N\left(\{q^\alpha\}_\alpha\right) \propto \delta\left(\sum_\alpha [q^\alpha]^2 - 1\right), \qquad q^\alpha = \sqrt{\xi^\alpha}. \tag{60}$$

The components of the vector $\mathbf{w}$ in the basis $\mathbf{u}^\alpha$ are distributed like the components of a vector extracted randomly from the hypersphere. This mirrors the statement that the eigenvectors of GOE matrices have the statistics of an orthonormal basis sampled uniformly, as discussed around Eq. (4) and in Box [B2]. For $r > 0$, instead, there is a term coupling the eigenvalues $\lambda^\alpha$ with the squared eigenvector projections $\xi^\alpha$.

The joint distribution of the largest eigenvalue $\lambda_N^{\max}$ and squared eigenvector projection $\xi_N^{\max}$ can be obtained from (57) by integrating over all variables $\lambda^\alpha, \xi^\alpha$ for $\alpha = 1, \ldots, N-1$. It takes the large deviation form:

$$\mathbb{P}\left(\lambda^N = x, \xi^N = u\right) = e^{-N\mathcal{F}(x,u) + o(N)}, \tag{61}$$

where the explicit expression of the large deviation function $\mathcal{F}$ is given in [39]. Interestingly, the large deviation function has different regimes depending on the choice of $x$ and $u$, and what governs the changes between these regimes is a BBP transition of the second-largest eigenvalue.

---

**[B4] Breaking isotropy: Localization, freezing and condensation.** When $r = 0$, $\mathbf{M}$ reduces to a GOE matrix $\mathbf{J}$ and its eigenvectors $\mathbf{u}^\alpha$ have the statistics of orthogonal vectors uniformly distributed on the hypersphere. In particular, arbitrary vectors $\mathbf{w}$ that are independent of $\mathbf{J}$ have components in the basis $\mathbf{u}^\alpha$ that are statistically equivalent for each element of the basis, with no special direction (i.e., no particular $\mathbf{u}^\alpha$) that is significantly more aligned to $\mathbf{w}$ than the others. Typically, $\xi^\alpha = (\mathbf{w} \cdot \mathbf{u}^\alpha)^2 \sim 1/N$, as we have seen in (15) and (60). Borrowing a terminology from quantum mechanics, we can say that the vector $\mathbf{w}$ is *delocalized* in the basis $\mathbf{u}^\alpha$ (in general, in quantum mechanical problems $\mathbf{w}$ would be a wave function, and $\mathbf{u}^\alpha$ the basis of eigenstates of some local operator). The same situation remains true for $r \in [0, r_c]$. At $r = r_c$, the transition (46) can be interpreted as a *localization transition*: $\mathbf{w}$ becomes aligned towards one element of the basis $\mathbf{u}^\alpha$, and its overlap with all other basis vectors is negligible in comparison. The vector $\mathbf{w}$ is thus localized in the basis

$\mathbf{u}^\alpha$, and isotropy is clearly broken (the components are no longer equivalent to each others). Localization in a given basis can be quantified by the Inverse Participation Ratio (IPR):

$$\text{IPR}(\mathbf{w}) = \frac{\sum_{\alpha=1}^{N}(\mathbf{w}\cdot\mathbf{u}^\alpha)^4}{\sum_{\alpha=1}^{N}(\mathbf{w}\cdot\mathbf{u}^\alpha)^2} = \sum_{\alpha=1}^{N}(\mathbf{w}\cdot\mathbf{u}^\alpha)^4, \tag{62}$$

where we have used the fact that $\sum_{\alpha=1}^{N}(\mathbf{w}\cdot\mathbf{u}^\alpha)^2 = 1$ by normalization. When $N$ is large, the IPR vanishes in the delocalized phase, while it remains of $O(1)$ in the localized one. Indeed, we find

$$\text{IPR}(\mathbf{w}) \sim \begin{cases} \sum_{\alpha=1}^{N}\left(\frac{1}{N}\right)^2 \sim \frac{1}{N} \xrightarrow{N\to\infty} 0, & \text{for } r \le r_c = \sigma, \\ \sum_{\alpha=1}^{N-1}\left(\frac{1}{N}\right)^2 + \left(1-\frac{\sigma^2}{r^2}\right) \sim \frac{1}{N} + 1 - \frac{\sigma^2}{r^2} \xrightarrow{N\to\infty} O(1), & \text{for } r > r_c = \sigma. \end{cases} \tag{63}$$

This also illustrates that while in the delocalized phase the sum defining the IPR is contributed roughly equally by all $N$ summands, in the localized phase the sum is entirely dominated by the maximal among the summand, and all the other terms give a negligible contribution despite being many. The first phenomenology shows up also when discussing concepts such as *quantum chaos*, *free probability* and *eigenstate thermalization* [40]. The second phenomenology is instead connected to the notion of *freezing* and *condensation*, see also Exercise 3 in Appendix A.3.

## 2.3 What: Ground state, metastability, dynamics

Armed with the RMT results of Sec. 2.2.2, we are now in the position to revisit and address the questions of Sec. 2.1.3.

### 2.3.1 Q1. A continuous recovery transition

Recovery of the unknown signal $\mathbf{v}$ is possible via the maximum likelihood estimator whenever the condition (16) is fulfilled. In the language of Sec. 2.2.2, given that $\mathbf{v} = \sqrt{N}\mathbf{w}$ and $\mathbf{s}_{\text{GS}}^\pm = \pm\sqrt{N}\mathbf{u}^N$, this condition is met whenever the typical value of the squared projections $\xi^N = (\mathbf{w}\cdot\mathbf{u}^N)^2$ remains positive when $N \to \infty$: this happens precisely beyond the BBP transitions at $r = r_c(\sigma)$. Therefore, for $N \to \infty$ a sharp transition occurs at $r = r_c(\sigma) = \sigma$, which separates an *impossible* from a *possible* phase. For $r \le r_c(\sigma)$, recovering the signal with maximum likelihood is impossible: even if one is able to optimize the energy landscape and to compute the ground states, one could not extract any useful information from them, given that in this regime $\mathbf{s}_{\text{GS}}^\pm$ are not aligned in the direction of $\mathbf{v}$. On the other hand, when $r > r_c(\sigma)$, the ground states are informative of the signal, and recovering information on the latter is possible via maximum likelihood. The critical value of signal-to-noise ratio $(r/\sigma)_c = 1$ is the *recovery threshold*, and the transition is of second order: the order parameter $q_\infty^2(\mathbf{s}_{\text{GS}}^\pm, \mathbf{v})$ grows continuously from zero to a positive value, without any jump, see Fig. 6 (*Left*).

Let us make a couple of comments on this inference transition. (1) As we discussed in Sec. 2.1.3, the recovery transition can also be found from the equilibrium formalism by studying the $\beta \to \infty$ limit of the Boltzmann measure (17). The equilibrium properties of the system can be studied at arbitrary values of the inverse temperature $\beta$ [14, 41, 42], see Exercise 3 in Appendix A.3. Such study reveals the occurrence of a phase transition at a finite $\beta_c(r)$, characterized by the phenomenon of *condensation* (akin to the Bose-Einstein condensation). The low temperature phase $\beta > \beta_c(r)$ is characterized by a macroscopic value $\sim O(N)$ of the squared overlap between a typical configuration $\mathbf{s}$ sampled with the Boltzmann measure (17), and the extremal eigenvector $\mathbf{u}^N$. In other words, $\lim_{N\to\infty} N^{-1}\langle(\mathbf{s}\cdot\mathbf{u}^N)^2\rangle_\beta = O(1)$, where

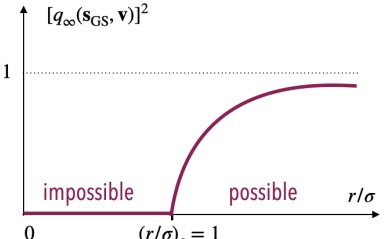
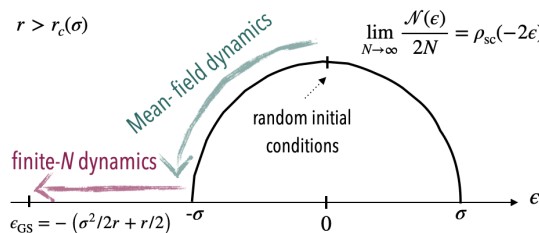

Figure 6: *Left.* Squared overlap of the ground state with the signal as a function of the signal-to-noise ratio, for quadratic landscapes $p = 2$. The recovery transition is continuous. *Right.* Sketch of the behavior of the dynamics initialized randomly, for $p = 2$: mean-field, "short-time" dynamics describes the descent from $\epsilon = 0$ to $\epsilon = -\sigma$, in the region of the landscape dominated by saddles of extensive index. The "long-time" dynamics beyond mean-field describes the exploration of the bottom of the energy landscape, up to convergence to a ground state.

$\langle \cdot \rangle_\beta$ denotes the average with respect to the Boltzmann measure (17). This means that there is a whole range of temperatures for which typical configurations extracted at equilibrium are very similar to (i.e., they have a large overlap with) one of the two ground states of the system, $\mathbf{s}_{\mathrm{GS}}^{\pm} = \pm\sqrt{N}\mathbf{u}^N$. In this low-temperature phase, the Boltzmann measure is thus partitioned into two "states" related by inversion symmetry, similarly to what happens in standard ferromagnets. The nature of this low-temperature phase however depends on the value of $r$: for $r > r_c(\sigma)$, $\mathbf{s}_{\mathrm{GS}}^{\pm}$ are strongly aligned (or anti-aligned) to the signal $\mathbf{v}$, and so are typical configurations at equilibrium: one could lift the degeneracy between the two states by adding a magnetic field in the direction of the signal $\mathbf{v}$, similarly to standard ferromagnets. For $r < r_c(\sigma)$, $\mathbf{s}_{\mathrm{GS}}^{\pm}$ are not (anti-)aligned with $\mathbf{v}$, but are random vectors whose orientation depends on the realization of the landscape: to select either of the two states, one should apply a sample-dependent magnetic field. This is similar to what happens in spin glasses. However, this system in equilibrium is not a spin glass, since in spin glasses the number of states contributing to the Boltzmann measure is larger than two [5]: it is dubbed a *ferromagnet in disguise* [41]. (2) The threshold $(r/\sigma)_c = 1$ is associated to one specific procedure to estimate the signal, that is maximum likelihood. One may wonder whether this threshold is *optimal*, or whether different estimators allow to recover information on the signal for values of $(r/\sigma) < 1$. This is a question to address within the Bayesian formalism, and the answer depends on the assumptions made on the statistical distribution of the signal (i.e., in the language of Box [B1], on the prior). For the low-rank matrix approximation problem that we are studying, one can show that the threshold given by maximum likelihood is optimal, for instance, when the signal vector $\mathbf{v}$ is taken with a spherical prior, meaning that $\mathbf{v}$ is extracted randomly with uniform measure in $\mathcal{S}_N(\sqrt{N})$; in this case, $(r/\sigma) = 1$ coincides with the *detection threshold*, below which no estimator is able to distinguish between the spiked matrix and a GOE matrix. On the other hand, maximum likelihood is generally not optimal when the signal prior encodes additional structure, for example sparsity [43].

### 2.3.2 Q2. A landscape made of saddles

We now consider the structure of the energy landscape on top of the ground states. As shown in Sec. 2.2.1, the stationary points of $\mathcal{E}_r(\mathbf{s})$ are in two-to-one relationship with the eigenvectors of the matrix $\mathbf{M}$. The landscape is *not rugged*, since the total number of stationary points is $2N$ and thus it does not grow exponentially with $N$. We now characterize their properties.

(i) *Energy density.* To study the energy distribution, we define for finite $N$

$$\mathcal{N}_N(\epsilon) = \text{number stationary points } \mathbf{s}^* \text{ such that } \epsilon_N(\mathbf{s}^*) = \epsilon. \qquad (64)$$

Given that the energy density of stationary points is related to the eigenvalues of the spiked matrix, from the RMT results of Sec. 2.2.2 on the eigenvalue density we can conclude that the random variable $\mathcal{N}_N(\epsilon)$ is self-averaging when $N \to \infty$, and

$$\lim_{N \to \infty} \frac{\mathcal{N}_N(\epsilon)}{2N} = \lim_{N \to \infty} \mathbb{E}\left[\frac{\mathcal{N}_N(\epsilon)}{2N}\right] = \rho_{\text{sc},\sigma}(-2\epsilon), \qquad (65)$$

where $\rho_{\text{sc},\sigma}$ is the semicircle law. The isolated eigenvalue, if it exists, does not matter for the asymptotic quantity (65), as it gives a subleading contribution.

(ii) *Linear stability.* The expression for the index (24) shows that the only stationary points that are stable minima are the ground states: all other stationary points have at least one negative Hessian eigenvalue. When $N \gg 1$, the variable $\alpha$ labeling the eigenvalues can be thought of as a continuous variable and most of the eigenvalues have an $\alpha = O(N)$: most stationary points are therefore saddles of *extensive index* $\kappa = O(N)$; in particular, those are the saddles whose energy distribution corresponds to the *bulk* of the density (65). Saddles with *intensive index* $\kappa = O(1)$ correspond to the stationary points whose energy is at the *edge* of the density (65).

(iii) *Geometry.* In the BBP transition, only the maximal (extremal) eigenvalue acquires (when $r > r_c$) an overlap with the direction identified by the rank-one perturbation: all other eigenvector satisfy $\xi^\alpha \overset{N \to \infty}{\longrightarrow} 0$, which implies that the overlap $q_N(\mathbf{s}_\alpha^\pm, \mathbf{v}) \overset{N \to \infty}{\longrightarrow} 0$. This means that all the saddles typically lie at the equator. Notice that, at the scale $1/N$, these overlaps $\xi^\alpha$ have an energy-dependent pattern that can be characterized explicitly by computing the scaled quantities $N q_N^2(\mathbf{s}_\alpha^\pm, \mathbf{v})$, which turn out to be self-averaging [44].

### 2.3.3  Q3. An "easy" but slow landscape optimization

We now consider the dynamics given by the Langevin equation (18), where we recall that $\nabla_\perp \mathcal{E}_r(\mathbf{s})$ denotes the gradient of the energy landscape $\mathcal{E}_r(\mathbf{s})$ restricted to the hypersphere $\mathcal{S}_N(\sqrt{N})$. To implement this spherical constraint, similarly to what we have done for the static analysis of the landscape, we exploit a time-dependent Lagrange multiplier:

$$\frac{ds_i(t)}{dt} = -\sum_{j=1}^N M_{ij} s_j(t) - \lambda(t) s_i(t) + \sqrt{\frac{2}{\beta}} \eta_i(t), \quad \mathbf{s}(t=0) = \mathbf{s}_0, \qquad (66)$$

where at any time $\lambda(t)$ is chosen to enforce $\mathbf{s}(t) \cdot \mathbf{s}(t) = N$. We focus on the situation in which the initial condition $\mathbf{s}_0$ of the dynamics is extracted randomly with a uniform measure on the hypersphere, or equivalently is extracted from an equilibrium Boltzmann measure (17) at $\beta = 0$. In the limit of vanishing noise $\beta \to \infty$, Langevin dynamics is expected to converge to equilibrium at a sufficiently large timescale $\tau_{\text{eq}}(N)$: when $\beta \to \infty$, equilibrium corresponds to the ground states of the energy landscape. Therefore, the equilibration timescale $\tau_{\text{eq}}(N)$ is the one associated to the landscape's optimization. Being the total number of stationary points only polynomial and not exponential in $N$ and due to the lack of trapping metastable states, it is natural to expect that optimization of the quadratic energy landscape is *not hard*, i.e., reaching the ground-state will not require exponentially large timescales, $\tau_{\text{eq}}(N) \nsim O(e^N)$. However, $\tau_{\text{eq}}(N)$ may grow with $N$, even if not exponentially fast. In fact, in this model $\tau_{\text{eq}}(N) \overset{N \to \infty}{\longrightarrow} \infty$: because of this, one has to be careful when studying the dynamics in the

the large dimensional limit ($N \to \infty$) and in the large time limit ($t \to \infty$), since the two limits do not commute. If the dynamics is studied in *mean-field*, i.e., taking the limit $N \to \infty$ first, one is describing the time evolution of the system at *short timescales*, shorter than the equilibration one (which diverges in this limit). To study the dynamics at *large timescales* up to equilibration, one has to keep the size of the system $N$ finite (albeit possibly very large) and characterize time evolution at *times scaling with $N$*. This is in general a challenge, as it requires to go beyond the mean-field limit. Langevin dynamic for quadratic Gaussian landscapes is one of the rare examples for which the dynamics can be characterized both at short and at large timescales, and it exhibits interesting features in both cases. We begin by quantifying what we mean by short and large timescales in this model, and then report results on the dynamics in these two regimes.

**Quadratic landscape: Eigenvalue decomposition, and the dynamical crossover.** Consider the Langevin equation (66) for $\beta \to \infty$ and for a fixed realization of the matrix **M**. We perform a rotation and express the configuration vector $\mathbf{s}(t)$ in the eigenbasis of **M**: the resulting rotated vector has components $s_\alpha(t) = \mathbf{s}(t) \cdot \mathbf{u}^\alpha$, and its time evolution is given by

$$\frac{ds_\alpha(t)}{dt} = -[\lambda^\alpha - \lambda(t)]s_\alpha(t). \tag{67}$$

This is a non-linear equation due to the Lagrange multiplier that couples all the components (or modes) by enforcing the global normalization condition $\sum_{\alpha=1}^{N} s_\alpha^2(t) = N$. To monitor the convergence to the ground state configuration(s), it is convenient to introduce the excess energy density

$$\Delta\epsilon_N(t) := \frac{\mathcal{E}_r(\mathbf{s}(t))}{N} - \epsilon_{\text{GS}}, \tag{68}$$

where $\epsilon_{\text{GS}} = -\lambda_N^{\max}/2$ is the typical value of the ground state energy density, whose value depends on $r$. Using (67), the following expression can be derived [45] for the excess energy under the assumption that the initial condition $\mathbf{s}(t=0) = \mathbf{s}_0$ is random (meaning that $s_\alpha(t=0) = 1$ for all $\alpha = 1, \ldots, N$):

$$\Delta\epsilon_N(t) = \frac{1}{2}\frac{\sum_{\alpha \neq N}(\lambda^N - \lambda^\alpha)e^{-2(\lambda^N - \lambda^\alpha)t}}{1 + \sum_{\alpha \neq N} e^{-2(\lambda^N - \lambda^\alpha)t}} \overset{t \gg 1}{\approx} g_N e^{-2g_N t}, \quad g_N := \lambda^N - \lambda^{N-1}. \tag{69}$$

The excess energy vanishes for systems that have reached equilibrium. From (69), one sees that the largest timescale associated to its decay is the inverse of the smallest among the gaps $\lambda^N - \lambda^\alpha$, that is the gap $g_N$ between the two largest eigenvalues: $g_N^{-1}$ gives therefore a bound to the equilibration timescale. This timescale is also the one at which the system probes the discreteness of the spectrum, i.e., the dynamics becomes sensitive to the finiteness of $N$. We define it as the *crossover timescale*

$$\tau_{\text{cross}}(N) \sim \frac{1}{g_N} = \frac{1}{\lambda^N - \lambda^{N-1}}, \tag{70}$$

since it marks a crossover between mean-field and non mean-field dynamics. When $t \ll \tau_{\text{cross}}(N)$ the dynamics is insensitive to the fact that $N$ is finite, and in fact it behaves as if $N \to \infty$. For $t \gg \tau_{\text{cross}}(N)$, instead, deviations from $N \to \infty$ become appreciable. This crossover timescale distinguishes between the short timescales and the large timescales regime of the dynamics.

This crossover timescale depends clearly on the statistics of extremal eigenvalues, which in turn depends on the value of $r$ according to (49). In the subcritical regime, $r$ is small enough so that there is no isolated eigenvalue, and the maximal eigenvalue has Tracy-Widom fluctuations

as in the $r = 0$ case. The two largest eigenvalues are very close to each others for large $N$, and they have fluctuations of order $O(N^{-2/3})$ around their typical value $2\sigma$. The gap $g_N$ between them is of the same order of magnitude in $N$. On the other hand, in the supercritical regime $r$ is large enough so that the largest eigenvalue is isolated while the second eigenvalue sticks to the boundary of the semicircle $2\sigma$. In this case, the gap $g_N$ is $O(N^0)$: the system is gapped. One can in fact show that in this regime, the timescale distinguishing between mean-field and non mean-field dynamics is logarithmic in $N$ [46, 47]. Finally, the statistics of the gap in the critical regime is, as far as we know, still an open problem in RMT. In summary,

$$\tau_{\text{cross}}(N) \sim \frac{1}{g_N} \sim \begin{cases} O(N^{\frac{2}{3}}), & \text{subcritical regime (Tracy-Widom),} \\ ?, & \text{critical regime,} \\ O(N^0) = O(\log N), & \text{supercritical regime (gapped system).} \end{cases} \tag{71}$$

We begin by discussing results on the dynamics at timescales that are much shorter than the crossover one, where the mean-field formalism applies.

■ **Short timescales: Dynamical Mean-Field Theory (DMFT).** The time evolution of the system at short timescales is described studying the Langevin dynamics in the mean-field limit, i.e., when $N \to \infty$. In models with all-to-all interactions like (66), where each component $s_i$ of the configuration vector interacts with any other component $s_j$, this limit of the dynamics can often be studied exactly. A crucial feature is that when $N \to \infty$ also the dynamics shows concentration: some properties of the dynamical trajectories become *self-averaging*, and reach asymptotic values that are deterministic and coincide with the limiting value obtained averaging the same quantities over the realizations of the noise $\boldsymbol{\eta}(t)$ and the realizations of the landscape $\mathcal{E}_r(\mathbf{s})$. Such self-averaging quantities are in general *global*, i.e., they involve sums over all components of the configuration vector $\mathbf{s}(t)$. Examples are the energy density or the correlation function, and the above statement of self-averagingness corresponds to:

$$\lim_{N \to \infty} \left( \epsilon_N(t) = \frac{\mathcal{E}_r(\mathbf{s}(t))}{N} \right) = \lim_{N \to \infty} \mathbb{E}\left[ \epsilon_N(t) \right] := \epsilon_\infty(t),$$

$$\lim_{N \to \infty} \left( c_N(t, t') = \frac{1}{N} \sum_{i=1}^{N} s_i(t) s_i(t') \right) = \lim_{N \to \infty} \mathbb{E}\left[ c_N(t, t') \right] := c_\infty(t, t'), \tag{72}$$

where now $\mathbb{E}[\cdot]$ denotes the average over both the noise and the random landscape. For the model (66), in the limit $N \to \infty$ one can derive *closed* dynamical equations for a limited number of these self-averaging quantities, that are one-point functions in time (like the energy density) and two-point functions in time (like the correlation function). These quantities play the role of mean-field *order parameters* of the dynamics – in the same way as the magnetization is a mean-field order parameter for Ising systems. The equations relating the order parameters are called in the literature *Dynamical Mean Field Theory (DMFT)* equations. For disordered systems models they have been studied since the foundational works [48, 49], and for the models we are discussing in these notes they have also been derived rigorously in [50]. Nowadays this formalism is used to study the dynamics of high-dimensional systems with randomness in a variety of contexts including inference and learning, see [51–53] for some recent examples and [54] for a review.

For the model we are looking at, not surprisingly one finds that in the mean-field limit the dynamics does not depend on $r$: exactly as for the eigenvalue distribution, the rank-one perturbation gives sub-leading contributions that are negligible when $N \to \infty$. The solution to the DMFT equations for $r = 0$ has been studied[3] in detail in [45]. We briefly summarize

---

[3]Due to the fact that $r$ does not affect the equations in the mean-field limit, the phenomenology of [45] holds true also for $r > 0$, if one focuses on the short timescale regime.

some of the results obtained studying the equations in the noiseless limit ($\beta \to \infty$).

- *Algebraic decay.* The excess energy density (69) converges slowly (algebraically) to a stationary value, that equals to $-\sigma$. The latter is however the ground state energy density only when $r \le r_c$; recalling that the excess energy is defined with respect to the ground state energy density, we therefore have:

$$\epsilon_\infty(t) = \lim_{N\to\infty} \mathbb{E}[\Delta\epsilon_N(t)] = v_{\mathrm{MF}}(t) \stackrel{t\gg 1}{\sim} \begin{cases} \frac{3\sigma}{8t}, & r \le r_c(\sigma), \\ \frac{3\sigma}{8t} - \sigma - \epsilon_{\mathrm{GS}}, & r > r_c(\sigma). \end{cases} \tag{73}$$

The power-law decay of the energy density is an indicator of slow dynamics in the system.

- *Out-of-equilibrium and aging.* In this mean-field regime, the dynamics of the system is *always out-of-equilibrium*, and it exhibits typical features of glassy dynamics. This is particularly evident from the behavior of the correlation function $c_\infty(t, t')$, which (i) is not time-translational invariant: $c_\infty(t, t') \ne c_\infty(t - t')$, as it would hold instead for systems that are in equilibrium; (ii) it does not satisfy the fluctuation dissipation relation constraining correlation and response functions for systems in equilibrium [55]; (iii) it shows a clear separation of timescales in $\tau = t - t'$, with an "equilibrium-like" behavior for small time separation $\tau$, and an out-of-equilibrium behavior for large time separation $\tau$; (iv) the out-of-equilibrium part of the dynamics is characterized by *aging*: the dynamics becomes slower and slower as the system becomes older, i.e., as it ages [56]; (v) the system realizes the scenario of weak ergodicity breaking. We refer to the review [57] for a detailed description of these crucial concepts in glassy dynamics.

- *Landscape interpretation.* When the initial condition of the dynamics $\mathbf{s}_0$ is chosen randomly, its energy density is typically $\epsilon = 0$, since the exponential majority of configurations on the hypersphere have such energy density (i.e., the entropy peaks at $\epsilon = 0$). It is clear from (73) that within the timescales described by the DMFT equations, the system descends in the energy landscape and it explores regions where the energy is extensively higher (in $N$) than the one of the ground state, i.e. the energy density is higher. In these regions, the landscape is dominated by saddles of extensive index, whose density is described by the semicircle law. One can picture the time evolution as a descent along one branch of the semicircle, which asymptotically reaches the boundary at $-\sigma$, see Fig. 6 (*Right*). The knowledge we have gathered on the structure of the energy landscape and on the distribution of the saddles allows us to understand intuitively why the mean-field dynamics is slow, and exhibits the progressive slowing down of aging, even though there are no local minima trapping the dynamics: as the system descends in the energy landscape, indeed, it encounters saddles that have an index that is progressively lower and lower; if the system is attracted by these saddles, the lower is the energy density the harder is to escape from these unstable attractor by finding some direction in configuration space where the landscape curvature is negative, since there are fewer. Therefore, as time proceeds, the system explores portion of the energy landscapes that are lower in energy, from which it is harder to escape, and thus the dynamics slows down. Aging dynamics has therefore a natural interpretation in terms of the geometrical properties of the energy landscape.

■ **Large timescales: Dynamics at finite $N$.** For systems for which $\tau_{\mathrm{eq}}(N)$ grows with $N$, the mean-field dynamical equations of course do not describe equilibration: since they are derived taking the limit $N \to \infty$, the equilibrium regime of the dynamics ($t, t' \sim \tau_{\mathrm{eq}}$) lies outside the regimes of timescales captured by the mean-field formalism. To characterize how the system

equilibrates, and in general to characterize the dynamics beyond the crossover timescales (71), one needs to go beyond mean-field and study the dynamical equations at $N$ finite, and times that grow with $N$. That is, one has to invert the order in which the large-$N$ limit and the large-time limit are taken. As we mentioned, this is in general very challenging, in particular for systems with randomness: for $N$ finite fluctuations matter, quantities are not self-averaging, averages may not coincide with typical values, and one should deal with distributions. The problem (66) for $r = 0$ represents one of the few examples for which also this regime of the dynamics can be studied in great detail, using RMT. At variance with mean-field, the finite-$N$ dynamics depends on $r$, and on which side of the BBP crossover one is in. We summarize some of the key features of this dynamics, distinguishing between the different regimes.

- *Subcritical regime (r small).* In this regime, the system behaves as for $r = 0$, a limit that has been studied in detail in [58]. The crossover timescale is of $O(N^{2/3})$, and one can show that

$$\mathbb{E}[\Delta\epsilon_N(t)] \sim \begin{cases} \nu_{\text{MF}}(t), & t \ll \tau_{\text{cross}}(N) \sim N^{\frac{2}{3}}, \\ N^{-\frac{2}{3}}\nu_{\text{NMF}}(tN^{-\frac{2}{3}}), & t \gg \tau_{\text{cross}}(N) \sim N^{\frac{2}{3}}. \end{cases} \quad (74)$$

For times larger than $\tau_{\text{cross}}(N)$, the system explores the bottom of the energy landscape, that corresponds to intensive energies above the ground states: this corresponds to the edge of the semicircle law. This regime of the dynamics is sensitive to the discreteness of the spectrum and to the gap $g_N$ between extreme values, see (69). In particular, to characterize the behavior of the average excess energy (74), knowledge of the typical value of the gap is not enough: in fact, the quantity $\Delta\epsilon_N(t)$ is not self-averaging for $N$ finite, and its average is dominated by instances in which the gap is atypically small [59]. Therefore, to determine $\nu_{\text{NMF}}$ in (74) one needs to have access to the full distribution of $g_N$, to compute

$$\mathbb{E}[\Delta\epsilon_N(t)] \overset{t \gg 1}{\approx} \int_0^\infty dg_N P(g_N) g_N e^{-2g_N t}, \quad (75)$$

according to (69). This distribution is that of GOE matrices, and it is known [60]; in particular, one can show that

$$\mathcal{P}(N^{\frac{2}{3}}g_N) \sim \begin{cases} bN^{\frac{2}{3}}g_N, & N^{\frac{2}{3}}g_N \to 0 \quad \text{(small gaps)}, \\ e^{-\frac{2}{3}\left(N^{\frac{2}{3}}g_N\right)^{\frac{3}{2}}}, & N^{\frac{2}{3}}g_N \to \infty \quad \text{(large gaps)}, \end{cases} \quad (76)$$

from which it follows

$$\nu_{\text{NMF}}(x) \sim \begin{cases} \frac{3\sigma}{8x}, & x \to 0 \quad \text{(small times)}, \\ \frac{a\sigma}{x^3}, & x \to \infty \quad \text{(large times)}, \end{cases} \quad (77)$$

which matches with the mean-field result when $x = N^{-\frac{2}{3}}t \to 0$, and shows a different but still algebraic decay for large times beyond the crossover one. In (77), $a$ is a constant. Remarkably, even the scaling function $\nu_{\text{NMF}}(x)$ in (74) is known for this problem.

- *Supercritical regime (r large).* In this case, the system is gapped: the energy density of the ground state is $O(1)$ smaller than that of the lower-energy saddles/other configurations. Once the system reaches that low-energy region at sufficiently large times, it relaxes exponentially fast to the ground state. One has:

$$\mathbb{E}[\Delta\epsilon_N(t)] \sim \begin{cases} \nu_{\text{MF}}(t), & t \ll \tau_{\text{cross}}(N) \sim \log N, \\ \frac{C_r}{t^{\frac{3}{2}}}e^{-2t\left|\frac{\sigma^2}{r}+r-2\sigma\right|}, & t \gg \tau_{\text{cross}}(N) \sim \log N. \end{cases} \quad (78)$$

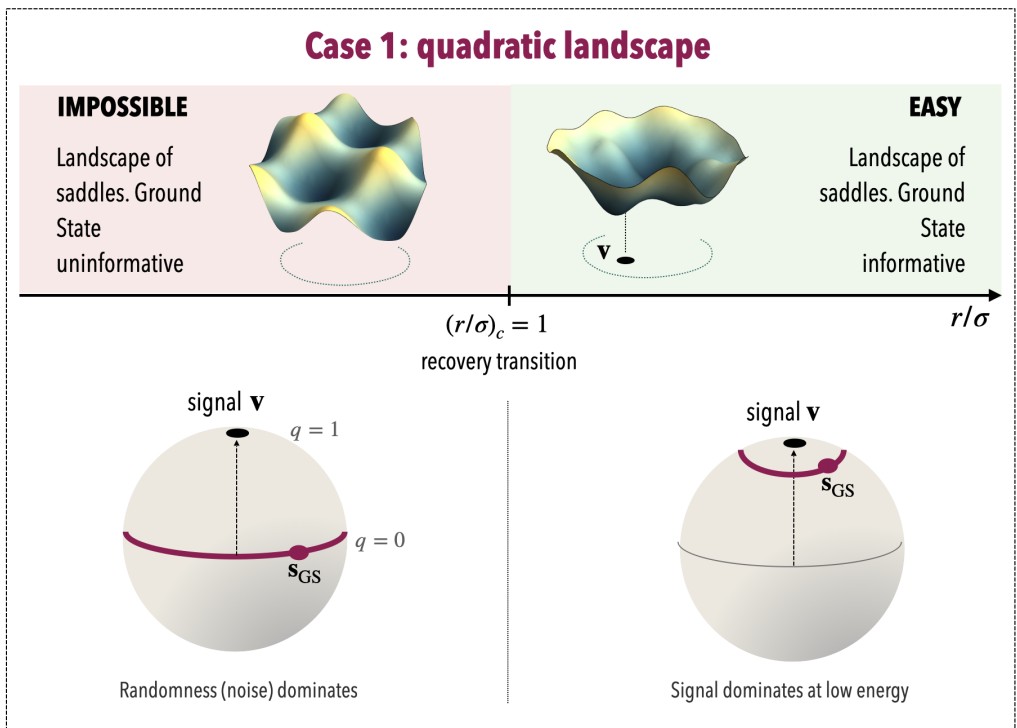

Figure 7: Summary of the phenomenology emerging from the landscape analysis in case 1.

In the long-time regime, the exponential decay is controlled by the gap between the two largest eigenvalues, while the $t^{-3/2}$ correction arises from the contribution of all other eigenvalues in the sum (69), see [61] for its derivation.

- *Critical regime ($r \approx r_c$).* In this case, the average excess energy should be computed from (75), but the explicit form of the gap distribution in the critical regime (as far as we know) is unknown. Recent numerical studies [61] suggest that $\mathcal{P}(g_N)$ behaves as a power law for $g_N$ small, with an exponent $a(\omega) \approx a_0 + a_1 \omega$ that depends linearly on the scaling variable $\omega = N^{1/3}(r - r_c)$. Based on this observation, the authors of [61] derive that

$$\mathbb{E}[\Delta\epsilon_N(t)] \sim \begin{cases} v_{\mathrm{MF}}(t) \sim e^{-2t|\frac{\sigma^2}{r}+r-2\sigma|}, & t \ll \tau_{\mathrm{cross}}(N) \sim N^{\frac{1}{3}}, \\ N^{\frac{1}{3}} f_\omega(tN^{-\frac{1}{3}}), & t \gg \tau_{\mathrm{cross}}(N) \sim N^{\frac{1}{3}}, \end{cases} \quad (79)$$

where $f_\omega(x)$ is an (analytically unknown) scaling function, that they conjecture to behave as $f_\omega(x) \sim 1/x^{2+a(\omega)}$.

This ends our discussion on optimization through Langevin dynamics (or gradient descent), and of its connections with the landscape's structure. We remark that this corresponds to a specific choice of the optimization algorithm, just as maximum likelihood is a specific choice of the estimator of the signal. Different algorithms may be considered, which can perform better than Langevin at signal estimation. The readers interested in algorithms commonly used in the context of statistical inference and in their application to the low-rank matrix estimation problem are referred to [62, 63]. To conclude this Section, in Figure 7 we give a pictorial summary of the phenomenology emerging from the landscape analysis in case 1.

# 3 Case 2: Higher-order high-dimensional random landscape

## 3.1 Why: An example from high-dimensional inference, again

### 3.1.1 A "hard" inference problem: Noisy tensors

Let us now discuss a richer instance of the denoising problem, which was introduced in [64] and which is currently the subject of a very significant stream of works. We consider exactly the same setting as in Section 2; however, instead of focusing on matrices, we focus now on the problem of denoising rank-one perturbed, or spiked, Gaussian *tensors*. Consider tensors of the form:

$$\mathbf{M} = \frac{r}{N^{p-1}}\mathbf{v}^{\otimes p} + \mathbf{J}, \qquad p \in \mathbb{N}, \quad p \geq 3. \tag{80}$$

This object has entries labeled by $p$ indices, equal to

$$M_{i_1 i_2 \cdots i_p} = \frac{r}{N^{p-1}} v_{i_1} v_{i_2} \cdots v_{i_p} + J_{i_1 i_2 \cdots i_p}, \quad i = 1, \ldots, N. \tag{81}$$

As before, the signal $\mathbf{v}$ is a vector belonging to the hypersphere $\mathcal{S}_N(\sqrt{N})$, and the tensor $\mathbf{J}$ is symmetric with respect to permutations of the indices, and its entries are random Gaussian variables satisfying

$$\mathbb{E}[J_{i_1 i_2 \cdots i_p}] = 0, \qquad \mathbb{E}[J^2_{i_1 i_2 \cdots i_p}] = \frac{\sigma^2}{N^{p-1}} \prod_{j=1}^{N} c_j(i_1, \ldots, i_p)!, \qquad i_1 \leq i_2 \leq \cdots \leq i_p, \tag{82}$$

where $c_j(i_1, \ldots, i_p)$ denotes the number of times the index $j$ appears in the index set $(i_1, \ldots, i_p)$. This combinatorial factor rescales the variance of the entries of the tensor in which some indices are repeated, similarly to the matrix case (2), where the diagonal entries have a variance that is twice that of the off-diagonal ones.[4]

With an analogous reasoning as in Sec. 2.1.2, one can show that the maximum likelihood estimator coincides with the ground state of the energy landscape

$$\mathcal{E}_{r,p}(\mathbf{s}) = -\frac{1}{p!}\sum_{i_1,i_2,\ldots i_p} M_{i_1 i_2 \cdots i_p} s_{i_1} s_{i_2} \cdots s_{i_p} = -\frac{1}{p!}\sum_{i_1,i_2,\ldots i_p} J_{i_1 i_2 \cdots i_p} s_{i_1} s_{i_2} \cdots s_{i_p} - \frac{rN}{p!}\left(\frac{\mathbf{s} \cdot \mathbf{v}}{N}\right)^p, \tag{83}$$

which is again a high-dimensional random energy landscape. As for the matrix case, the part of the landscape depending on the signal $\mathbf{v}$ is deterministic, convex and minimized by the signal itself. On the other hand, the part of the landscape that does not depend on the signal is random with isotropic statistics, such that

$$\mathbb{E}[\mathcal{E}_{0,p}(\mathbf{s})] = 0, \qquad \mathbb{E}[\mathcal{E}_{0,p}(\mathbf{s})\mathcal{E}_{0,p}(\mathbf{s}')] = \frac{N\sigma^2}{p!} q_N^p(\mathbf{s}, \mathbf{s}'). \tag{84}$$

The problem is formally completely analogous to that discussed in Section 2, which corresponds to $p = 2$. However, the landscape $\mathcal{E}_{0,p}(\mathbf{s})$ for $p \geq 3$ has a completely different structure with respect to the quadratic one: it is rugged. This landscape, known as the "pure spherical $p$-spin model" with $p > 2$, has been introduced in [65] and since then it has been studied extensively in the theory of disordered systems, as a mean-field model of structural glasses [66,67]. An introductory review of the main theoretical results related to this model is given in [68].

---

[4]To see how this factor arises, it is useful to define the entries of the symmetric tensor as the linear combination $J_{i_1 i_2 \cdots i_p} = [p!]^{-1} \sum_{\pi} X_{\pi(i_1)\pi(i_2)\cdots\pi(i_p)}$, where the sum runs over all the $p!$ permutations $\pi$ of the indices $(i_1 i_2 \cdots i_p)$, and $\mathbf{X}$ is an asymmetric tensor whose entries $X_{i_1 i_2 \cdots i_p}$ are independent Gaussian variables with zero mean and variance $\sigma^2 p!/N^{p-1}$. The components $J_{i_1 i_2 \cdots i_p}$ are then obtained as a sum of $p!/\prod_{j=1}^{N} c_j(i_1, \ldots, i_p)!$ independent random variables. Their variance reproduces (82). Notice that these combinatorial factors are irrelevant when $N$ is large, since the number of tensor entries with (at least two) repeated indices is suppressed by (at least) a factor of $N$ compared to the number of entries with all distinct indices.

### 3.1.2 A landscape with a positive complexity

For the low-rank tensor estimation problem, we aim at addressing the same questions as for the matrix case, by studying the distribution of the stationary points of the landscape. As before, we implement the spherical constraint with a Lagrange multiplier, and define the landscape

$$\mathcal{E}_{r,p}(\mathbf{s};\lambda) = -\frac{1}{p!}\sum_{i_1,i_2,\dots i_p} M_{i_1 i_2 \cdots i_p} s_{i_1} s_{i_2} \cdots s_{i_p} + \frac{\lambda}{2}\left(\sum_i s_i^2 - N\right), \quad \mathbf{s} \in \mathbb{R}^N. \tag{85}$$

Stationary points of this modified function are pairs $(\mathbf{s}^*, \lambda^*)$ satisfying (for symmetric $\mathbf{M}$):

$$\frac{\partial \mathcal{E}_{r,p}(\mathbf{s},\lambda)}{\partial s_i} = -\sum_{i_2 \le i_3 \cdots \le i_p} M_{i i_2 \cdots i_p} s_{i_2} \cdots s_{i_p} + \lambda s_i \bigg|_{\mathbf{s}^*,\lambda^*} = 0,$$

$$\frac{\partial \mathcal{E}_{r,p}(\mathbf{s},\lambda)}{\partial \lambda} = \sum_i s_i^2 - N \bigg|_{\mathbf{s}^*,\lambda^*} = 0. \tag{86}$$

Once more, multiplying the first equation by $s_i$, summing over $i$ and using the second equation, we obtain:

$$\lambda^* = -\frac{\nabla \mathcal{E}_{r,p}(\mathbf{s}^*) \cdot \mathbf{s}^*}{N} = -p\frac{\mathcal{E}_{r,p}(\mathbf{s}^*)}{N} = -p\epsilon_N(\mathbf{s}^*), \tag{87}$$

which fixes the value of the Lagrange multiplier $\lambda$ as being proportional to the energy density of the configuration $\mathbf{s}^*$. However, at variance with the case $p = 2$, the first of the equations (86) is not linear, and the number of its solutions is not fixed by the dimensionality $N$.

Similarly to (64), we define:

$$\mathcal{N}_N(\epsilon) = \{\text{number stationary points } \mathbf{s}^* \text{ such that } \epsilon_N(\mathbf{s}^*) = \epsilon\} = \max_{q \in [0,1]} \mathcal{N}_N(\epsilon, q), \tag{88}$$

where

$$\mathcal{N}_N(\epsilon, q) = \{\text{number stationary points } \mathbf{s}^* \text{ such that } \epsilon_N(\mathbf{s}^*) = \epsilon \text{ and } q_N(\mathbf{s}^*, \mathbf{v}) = q\}. \tag{89}$$

In the case of quadratic landscape $p = 2$, the RMT results imply that:

(i) Almost all stationary points are saddles and lie at the equator, where $q = 0$. Therefore, $\mathcal{N}_N(\epsilon) = \max_{q \in [0,1]} \mathcal{N}_N(\epsilon, q) = \mathcal{N}_N(\epsilon, q = 0)$;

(ii) $\mathcal{N}_N(\epsilon)$ is a random variable of $O(N)$ when $N \gg 1$: the scaled variable $\mathcal{N}_N(\epsilon)/N$ remains of $O(1)$ when $N \to \infty$;

(iii) $\mathcal{N}_N(\epsilon)/N$ is self-averaging, its distribution converges to its average when $N \to \infty$.

These facts cease to be granted when $p > 2$. One finds instead that:

(i) $\mathcal{N}_N(\epsilon, q)$ and $\mathcal{N}_N(\epsilon)$ are random variables of $O(e^N)$ when $N \gg 1$, meaning that one can set $\mathcal{N}_N(\epsilon, q) = e^{N\Sigma_N(\epsilon, q)}$ where $\Sigma_N(\epsilon, q)$ is a random variable that remains of $O(1)$ when $N \to \infty$;

(ii) $\mathcal{N}_N(\epsilon, q)$ and $\mathcal{N}_N(\epsilon)$ *are not* self-averaging in general, while $\Sigma_N(\epsilon, q)$ is:

$$\lim_{N \to \infty} \Sigma_N(\epsilon, q) = \lim_{N \to \infty} \mathbb{E}[\Sigma_N(\epsilon, q)] = \Sigma_\infty(\epsilon, q). \tag{90}$$

The asymptotic value of this self-averaging random variable,

$$\Sigma_\infty(\epsilon, q) = \lim_{N \to \infty} \frac{\log \mathcal{N}_N(\epsilon, q)}{N} = \lim_{N \to \infty} \mathbb{E}\left[\frac{\log \mathcal{N}_N(\epsilon, q)}{N}\right], \tag{91}$$

is called the *complexity of the energy landscape*. This function, which plays the role of an entropy for stationary points, is the focus of the following subsections.

### 3.1.3 Averages *vs* typical, annealed *vs* quenched, and replicas

The quantity (91) controls the scaling of the *typical* value (i.e., the most probable value) of the random variable $\mathcal{N}_N(\epsilon, q)$ when $N$ is large: this means that with probability converging to one when $N \to \infty$, it holds $\mathcal{N}_N(\epsilon, q) \sim e^{N\Sigma_\infty(\epsilon, q)}$. For variables scaling exponentially with $N$, the most probable value in general differs from the average value, $\mathbb{E}[\mathcal{N}_N(\epsilon, q)] \nsim e^{N\Sigma_\infty(\epsilon, q)}$. In other words, to characterize the typical behavior of the system one has to compute the average of the logarithm of the random variable, instead of the average of the random variable itself. In the jargon of disordered systems, (91) is defined as a *quenched* complexity, in contrast to the *annealed* complexity defined as

$$\Sigma_A(\epsilon, q) = \lim_{N \to \infty} \frac{\log \mathbb{E}[\mathcal{N}_N(\epsilon, q)]}{N}. \tag{92}$$

The annealed complexity is simpler to determine, and it is often computed as an approximation to the quenched one. Getting the latter involves determining the average of the logarithm of the random variable, which is more involved due to the non-linearity of the logarithm. To address this difficulty, one can exploit limiting formulas such as:

$$\mathbb{E}[\log \mathcal{N}_N] = \lim_{n \to 0} \frac{\mathbb{E}[\mathcal{N}_N^n] - 1}{n} \implies \Sigma_\infty = \lim_{N \to \infty} \lim_{n \to 0} \frac{\mathbb{E}[\mathcal{N}_N^n] - 1}{Nn}. \tag{93}$$

This formula is one instance of the so called *replica trick*, extensively used in the theory of disordered systems to compute self-averaging quantities such as the free-energy describing equilibrium properties. Using (93), the calculation of the quenched complexity is performed in two steps: (1) computing arbitrarily-high moments $n$ of the random variable $\mathcal{N}_N$ with $n \in \mathbb{N}$ treated as a parameter, (2) performing an analytic continuation of the resulting expressions from $n \in \mathbb{N}$ to $n \in \mathbb{R}$, in order to take the limit $n \to 0$.

The annealed complexity (92), in contrast, requires to determine only the first moment of the random variable, $n = 1$. Because of the concavity of the logarithm, the annealed complexity is always an upper bound to the quenched one,

$$\Sigma_A(\epsilon, q) \ge \Sigma_\infty(\epsilon, q) \implies \mathbb{E}[\mathcal{N}_N] \ge \mathcal{N}_N^{\text{typ}}. \tag{94}$$

This inequality reflects the fact that for quantities that are not self-averaging such as $\mathcal{N}_N$, the average value is not contributed by typical realizations of the random landscape, but rather by *rare realizations* that are associated to an atypically large number of stationary points. These rare realizations do not contribute to the typical value, but dominate the average, as we illustrate in Box [B5] with an example. Notice that a small difference between the quenched and annealed complexities may correspond to a large difference between the typical and average value of $\mathcal{N}_N$ when $N$ is large, due to the exponential amplification.

---

**[B5] When the average is atypical: an example.** Assume that $X_N$ is a random variable scaling as $X_N \sim e^{N Y_N}$, meaning that $Y_N = N^{-1} \log X_N$ remains of $O(1)$ when $N \to \infty$.

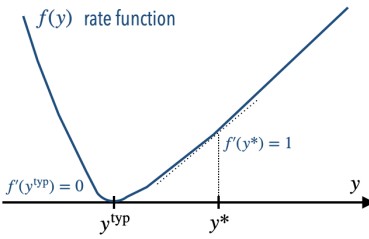

Figure 8: Sketch of the rate function of a large deviation probability: The minimum corresponds to the typical value of the variable.

---

We assume that when $N \gg 1$, the distribution of $Y_N$ takes a large deviation form:

$$P_{Y_N}(y)dy \sim e^{-Nf(y)+o(N)}dy\,, \tag{95}$$

where $f(y)$ is the rate function. The typical value of $Y_N$ is the value $y^{\text{typ}}$ where the minimum of the rate function $f(y)$ is attained; at this point, $f'(y^{\text{typ}}) = 0 = f(y^{\text{typ}})$, where the second equality follows from the fact that $P_{Y_N}(y)$ does not decay exponentially with $N$ when computed at the typical value, but is it of $O(1)$ by definition. The typical value of $X_N$ is thus $X_N^{\text{typ}} \sim e^{Ny^{\text{typ}}}$. On the other hand, the average value is given by:

$$\mathbb{E}[X_N] = \int dy\, e^{Ny} P_{Y_N}(y) = \int dy\, e^{N[y-f(y)]+o(N)} = e^{N[y^*-f(y^*)]+o(N)}\,, \tag{96}$$

where we have computed the integral with the Laplace method, choosing $y^*$ to be such that $\frac{d}{dy}[y-f(y)]|_{y^*} = 0$, meaning that $f'(y^*) = 1$. It is clear that $y^{\text{typ}} \neq y^*$, since $f'(y^*) = 1$ while $f'(y^{\text{typ}}) = 0$. Moreover, the value $y^*$ is *rare*, since $f(y^*) > 0$ and thus the probability to observe $Y_N = y^*$ is exponentially small according to (95). From (96) it appears that the average value of the exponentially-scaling quantity $X_N$ is *dominated by rare realizations* of the random variable, as values of $y$ that are $O(1)$ different from $y^{\text{typ}}$ contribute significantly to the average when $N \gg 1$.

## 3.2 How: (replicated) Kac-Rice formalism

### 3.2.1 A counting formula, with randomness

The so called *Kac-Rice formula* (KR), named after the works [69, 70], is a formula for the average number of solutions of equations with random coefficients. To introduce it, we consider first an example: let $f(x)$ be a non-monotonic function defined on an interval $[a, b]$. Assume that we want to count the number $\mathcal{N}(y)$ of points $x$ such that $f(x) = y$ for some given value $y$. We define a measure on these points, $\nu_y(x) = \delta(x - f^{-1}(y))$, and get the number as the integral of this measure on the interval $[a, b]$:

$$\mathcal{N}(y) = \int_a^b d\nu_y(x) = \int_a^b dx\, \delta(x - f^{-1}(y)) = \int_a^b dx\, |f'(x)|\, \delta(f(x) - y)\,, \tag{97}$$

where in the last equality we performed a change of variable on the delta distribution,

$$\delta(f(x) - y) = \frac{\delta(x - f^{-1}(y))}{|f'(x)|}\,,$$

which generates the Jacobian $|f'(x)|$. This formula can be easily generalized to higher dimensions: given a function $\mathbf{f}(\mathbf{x})$ defined for $\mathbf{x} \in \mathcal{I} \subset \mathbb{R}^d$ and given a value $\mathbf{y} \in \mathbb{R}^d$, the formula generalizes to

$$\mathcal{N}(\mathbf{y}) = \int_{\mathcal{I}} d\mathbf{x} \prod_{i=1}^d \delta(f_i(\mathbf{x}) - y_i) \left| \det\left( \frac{\partial f_i(\mathbf{x})}{\partial x_j} \right)_{ij} \right|\,, \tag{98}$$

where now the Jacobian term involves the calculation of the determinant of a matrix of derivatives.

Let us apply this formula to our counting problem: we are interested in the number of solutions of the non-linear equation $\nabla_\perp \mathcal{E}_{r,p}(\mathbf{s}) = 0$, which corresponds to $\mathbf{f}(\mathbf{x}) \to \nabla_\perp \mathcal{E}_{r,p}(\mathbf{s})$ and $\mathbf{y} \to 0$. In addition, we are interested in those configurations that satisfy the double

constraint $\epsilon_N(\mathbf{s}) = \epsilon$ and $q_N(\mathbf{s}, \mathbf{v}) = q$. We set $\mathcal{E}_{r,p} \to \mathcal{E}$ to simplify the notation. The analog of (98) becomes:

$$\mathcal{N}_N(\epsilon, q) = \int_{\mathcal{S}_N(\sqrt{N})} d\mathbf{s}\, \delta(\mathbf{s} \cdot \mathbf{v} - Nq)\, \delta(\mathcal{E}(\mathbf{s}) - N\epsilon)\, \delta(\nabla_\perp \mathcal{E}(\mathbf{s}))\, \left| \det \nabla_\perp^2 \mathcal{E}(\mathbf{s}) \right|. \qquad (99)$$

Taking the average of this expression, we get the Kac-Rice formula for the mean number of stationary points:

$$\mathbb{E}[\mathcal{N}_N(\epsilon, q)] = \int_{\mathcal{S}_N(\sqrt{N})} d\mathbf{s}\, \delta(\mathbf{s} \cdot \mathbf{v} - Nq)\, \mathbb{E}\left[\left| \det \nabla_\perp^2 \mathcal{E}(\mathbf{s}) \right| \right]_{\substack{\nabla_\perp \mathcal{E} = 0 \\ \mathcal{E} = N\epsilon}} \mathbb{P}_{\nabla_\perp \mathcal{E}, \mathcal{E}}(\mathbf{0}, N\epsilon). \qquad (100)$$

In this formula, $\mathbb{P}_{\nabla_\perp \mathcal{E}, \mathcal{E}}(\mathbf{0}, N\epsilon)$ denotes the joint probability distribution of the $(N-1)$-dimensional gradient vector $\nabla_\perp \mathcal{E}(\mathbf{s})$ and of the energy field $\mathcal{E}(\mathbf{s})$ at the point $\mathbf{s}$, evaluated at $\mathbf{0}$ for the gradient and $N\epsilon$ for the energy. This is the probability that the configuration $\mathbf{s}$ is a stationary point of energy density $\epsilon$. The term $\mathbb{E}\left[\left| \det \nabla_\perp^2 \mathcal{E}(\mathbf{s}) \right| \right]_{\substack{\nabla_\perp \mathcal{E} = 0 \\ \mathcal{E} = N\epsilon}}$ denotes instead the expectation value of the Hessian at $\mathbf{s}$, *conditional* to the fact that $\mathbf{s}$ is a stationary point of energy density $\epsilon$. To compute this term, one has to characterize the statistics of the curvature of the landscape at stationary points of a given energy density. Both terms in the integrand depend on $\mathbf{s}$, even though this is not explicit in our notation.

Variations of this formula have appeared in the earliest works on spin glasses starting from [71–73]. In these seminal works, the determinant was treated without considering its absolute value —an approximation that was later improved upon in [74,75], where strong connections to RMT were highlighted. Complexity calculations are currently an active area of research in both physics and mathematics [76–79], see also [7] for a review of recent developments.

### 3.2.2 The annealed complexity: A three-steps calculation

The annealed complexity (92) can be obtained computing the leading order term in $N$ of the Kac-Rice formula (100). In this section, we do not provide a full detailed calculation but rather outline the main steps needed to reach the final results, given in Eq. (116). Specifically, our aim is to highlight the three key ingredients that make this calculation feasible: (1) Gaussianity, (2) isotropy, and (3) large dimensionality combined with RMT. Let us discuss how these ingredients enter in the calculation.

**(1) Gaussianity.** The functions $\mathcal{E}(\mathbf{s})$, $\frac{\partial \mathcal{E}(\mathbf{s})}{\partial s_i}$ and $\frac{\partial^2 \mathcal{E}(\mathbf{s})}{\partial s_i \partial s_j}$ are Gaussian: to characterize their distribution, only the averages and covariances are needed. The latter can be computed explicitly (see Box [B6] for an example). Computing such averages and covariances, one can establish the following two Facts:

F1. The vector $\nabla_\perp \mathcal{E}(\mathbf{s})$ is uncorrelated (and thus independent) from the scalar and matrix $\mathcal{E}(\mathbf{s})$, $\nabla_\perp^2 \mathcal{E}(\mathbf{s})$ evaluated at the same configuration $\mathbf{s}$. This has some nice consequences: (i) the joint distribution $\mathbb{P}_{\nabla_\perp \mathcal{E}, \mathcal{E}}$ factorizes,

$$\mathbb{P}_{\nabla_\perp \mathcal{E}, \mathcal{E}}(\mathbf{0}, N\epsilon) = \mathbb{P}_{\nabla_\perp \mathcal{E}}(\mathbf{0})\, \mathbb{P}_{\mathcal{E}}(N\epsilon), \qquad (101)$$

and each of the two distributions is a Gaussian, which can be easily computed as outlined in Box [B6]. (ii) The distribution of the Hessian needs not to be conditioned to the gradient, being independent from it, but only to the energy density:

$$\mathbb{E}\left[\left| \det \nabla_\perp^2 \mathcal{E}(\mathbf{s}) \right| \right]_{\substack{\nabla_\perp \mathcal{E} = 0 \\ \mathcal{E} = N\epsilon}} \equiv \mathbb{E}\left[\left| \det \nabla_\perp^2 \mathcal{E}(\mathbf{s}) \right| \right]_{\mathcal{E} = N\epsilon}. \qquad (102)$$

This equality implies that for this model, the statistics of the Hessian at a stationary point are the same as at any arbitrary point of the same energy density.

F2. When $N$ is large, the $(N-1) \times (N-1)$ matrix $\nabla_\perp^2 \mathcal{E}(\mathbf{s})$ conditioned to $\mathcal{E}(\mathbf{s}) = N\epsilon$ has exactly the same statistics as random matrices of the form:

$$\tilde{\mathbf{M}}(\mathbf{s}) := \mathbf{M}(\mathbf{s}) - p\epsilon_N(\mathbf{s})\mathbf{I} := \tilde{\mathbf{J}} - r_{\text{eff}}[q_N(\mathbf{s}, \mathbf{v})]\mathbf{w}_\perp \mathbf{w}_\perp^T - p\epsilon_N(\mathbf{s})\mathbf{I}, \qquad (103)$$

where (i) $\tilde{\mathbf{J}}$ is a GOE($\tilde{\sigma}^2$) matrix with

$$\mathbb{E}[\tilde{J}_{ij}] = 0, \qquad \mathbb{E}[\tilde{J}_{ij}\tilde{J}_{kl}] = \frac{\tilde{\sigma}^2}{N}\delta_{ik}\delta_{jl}(1+\delta_{ij}) \begin{array}{c} i \le j, \\ k \le l, \end{array} \qquad \tilde{\sigma}^2 = \frac{p(p-1)\sigma^2}{p!}; \quad (104)$$

(ii) the diagonal shift proportional to $p\epsilon_N(\mathbf{s})$ arises from the spherical constraint, see (29); (iii) the vector $\mathbf{w}_\perp$ has unit norm, $\|\mathbf{w}_\perp\| = 1$, and it is the normalized projection of the signal vector $\mathbf{v}$ onto the tangent plane $\tau[\mathbf{s}]$; (iv) the function

$$r_{\text{eff}}[q] = \frac{r}{p!}p(p-1)q^{p-2}(1-q^2), \qquad (105)$$

is evaluated at $q_N(\mathbf{s}, \mathbf{v}) = \mathbf{s} \cdot \mathbf{v}/N$. Therefore, the statistics of the Hessian matrices at a stationary point of given energy density is that of a *shifted, rank-one perturbed GOE matrix*: these are precisely the type of matrices that we have discussed extensively in Sec. 2.2.2, in connection to quadratic landscapes. All that knowledge from RMT therefore turns out to be crucial to characterize the curvature of higher-order random landscapes in the vicinity of its stationary points, consistently with the quadratic approximation of the landscape around its stationary points.

---

**[B6] Computing correlations: an example.** Consider the unconstrained gradient components

$$\frac{\partial \mathcal{E}(\mathbf{s})}{\partial s_i} = -\frac{1}{p!}\sum_{k=1}^{p}\sum_{i_1,i_2,\dots,i_p}\delta_{i_k,i}J_{i_1 i_2 \cdots i_p}s_{i_1}\cdots s\!\!\!/_{i_k}\cdots s_{i_p} - \frac{rN}{p!}p\left(\frac{\mathbf{s}\cdot\mathbf{v}}{N}\right)^{p-1}\frac{v_i}{N},$$

where $s\!\!\!/_{i_k}$ indicates that the component $s_{i_k}$ is not included in the product. The average of these components coincides with the second term in this expression. Let us compute the covariance between different components at two different configurations $\mathbf{s}, \mathbf{s}'$:

$$\text{Cov}\left(\frac{\partial \mathcal{E}(\mathbf{s})}{\partial s_i}, \frac{\partial \mathcal{E}(\mathbf{s}')}{\partial s_j'}\right) = \left(\frac{1}{p!}\right)^2 \sum_{k_1=1}^{p}\sum_{k_2=1}^{p}\sum_{i_1,i_2,\dots,i_p}\sum_{j_1,j_2,\dots,j_p}\delta_{i_{k_1},i}\delta_{j_{k_2},j}$$
$$\times \mathbb{E}\left[J_{i_1 i_2 \cdots i_p}J_{j_1 j_2 \cdots j_p}\right]s_{i_1}\cdots s\!\!\!/_{i_{k_1}}\cdots s_{i_p}s_{j_1}'\cdots s\!\!\!/_{j_{k_2}}'\cdots s_{j_p}',$$

where $\text{Cov}(x, y) = \mathbb{E}[xy] - \mathbb{E}[x]\mathbb{E}[y]$. The correlations $\mathbb{E}\left[J_{i_1 i_2 \cdots i_p}J_{j_1 j_2 \cdots j_p}\right]$ are non-vanishing only if the two entries of the tensor are related by symmetry, i.e., the unordered index set $(i_1 i_2 \cdots i_p)$ must coincide with the unordered set $(j_1 j_2 \cdots j_p)$. For given $(i_1 i_2 \cdots i_p)$, the sum over $j_1, \dots, j_p$ is restricted to $p!/\prod_{k=1}^{N}c_k(i_1, \dots, i_p)!$ possibilities. Using (82), one sees that the terms $\prod_{k=1}^{N}c_k(i_1, \dots, i_p)!$ cancel, and

$$\text{Cov}\left(\frac{\partial\mathcal{E}}{\partial s_i}, \frac{\partial\mathcal{E}}{\partial s_j'}\right) = \frac{\sigma^2}{N^{p-1}}\frac{1}{p!}\sum_{k_1=1}^{p}\sum_{k_2=1}^{p}\sum_{i_1,i_2,\dots,i_p}\delta_{i_{k_1},i}\delta_{i_{k_2},j} \times s_{i_1}\cdots s\!\!\!/_{i_{k_1}}\cdots s_{i_p}s_{i_1}'\cdots s\!\!\!/_{i_{k_2}}'\cdots s_{i_p}'.$$

We now have to distinguish the case $k_1 = k_2$ ($p$ possibilities) and $k_1 \ne k_2$ ($p(p-1)$ possi-

bilities), to get

$$\text{Cov}\left(\frac{\partial \mathcal{E}}{\partial s_i}, \frac{\partial \mathcal{E}}{\partial s'_j}\right) = \frac{\sigma^2}{p!}\left[p\delta_{ij}\left(\frac{\mathbf{s}\cdot\mathbf{s}'}{N}\right)^{p-1} + p(p-1)\frac{s'_i s_j}{N}\left(\frac{\mathbf{s}\cdot\mathbf{s}'}{N}\right)^{p-2}\right].$$

Now, $\nabla_\perp \mathcal{E}(\mathbf{s})$ is the projection of $\nabla \mathcal{E}(\mathbf{s})$ on the space orthogonal to $\mathbf{s}$, i.e., on the tangent plane $\tau[\mathbf{s}]$. It is convenient to choose an orthonormal basis $\hat{\mathbf{e}}_\alpha(\mathbf{s})$ with $\alpha = 1,\ldots,N-1$ of $\tau[\mathbf{s}]$ as in Box [B3], and to choose

$$\mathbf{e}_{N-1}(\mathbf{s}) = \frac{1}{\sqrt{N(1-q_N^2(\mathbf{s},\mathbf{v}))}}\left(\mathbf{v}-q_N(\mathbf{s},\mathbf{v})\mathbf{s}\right), \qquad \mathbf{e}_\alpha(\mathbf{s})\perp\mathbf{s},\mathbf{v}, \qquad \alpha \leq N-2,$$

where $q_N = \mathbf{v}\cdot\mathbf{s}/N$, as the only basis vector in $\tau[\mathbf{s}]$ that is not orthogonal to $\mathbf{v}$. Setting $\mathbf{s}' \to \mathbf{s}$ and $(\nabla_\perp \mathcal{E})_\alpha := \nabla_\perp \mathcal{E}(\mathbf{s})\cdot\hat{\mathbf{e}}_\alpha(\mathbf{s})$, we find

$$\mathbb{E}[(\nabla_\perp \mathcal{E})_\alpha] = -\frac{rN}{p!}p\left(\frac{\mathbf{s}\cdot\mathbf{v}}{N}\right)^{p-1}\left(\frac{\mathbf{v}\cdot\hat{\mathbf{e}}_\alpha(\mathbf{s})}{N}\right) = -\frac{r\sqrt{N}}{p!}p[q_N(\mathbf{s},\mathbf{v})]^{p-1}\sqrt{1-q_N^2(\mathbf{s},\mathbf{v})}\,\delta_{\alpha,N-1},$$

and

$$\text{Cov}\left((\nabla_\perp \mathcal{E})_\alpha,(\nabla_\perp \mathcal{E})_\beta\right) = \frac{\sigma^2}{p!}\left[p\,\hat{\mathbf{e}}_\alpha(\mathbf{s})\cdot\hat{\mathbf{e}}_\beta(\mathbf{s}) + p(p-1)\frac{\hat{\mathbf{e}}_\alpha(\mathbf{s})\cdot\mathbf{s}\;\mathbf{s}\cdot\hat{\mathbf{e}}_\beta(\mathbf{s})}{N}\right] = \frac{\sigma^2 p}{p!}\delta_{\alpha\beta},$$

where we used the fact that $\hat{\mathbf{e}}_\alpha(\mathbf{s})\perp\mathbf{s}$. This shows that the distribution of the components of $\nabla_\perp \mathcal{E}(\mathbf{s})$ depends on $\mathbf{s}$ only through the overlap $q_N = \mathbf{v}\cdot\mathbf{s}/N$, via the averages. From these formulas, one can get the distribution (106).

**(2) Isotropy.** As we have stressed several times, on our denoising problem there is only one special direction on the hypersphere, that of the signal $\mathbf{v}$: in all other directions, the problem is isotropic. A consequence of this is that all averages and covariances of $\mathcal{E}(\mathbf{s})$, $\frac{\partial \mathcal{E}(\mathbf{s})}{\partial s_i}$ and $\frac{\partial^2 \mathcal{E}(\mathbf{s})}{\partial s_i \partial s_j}$ depend on the point $\mathbf{s}$ only through the overlap $q_N(\mathbf{s},\mathbf{v}) = \mathbf{s}\cdot\mathbf{v}/N$, see Box [B6]. Therefore, the dependence on $\mathbf{s}$ of the distributions (101) is parametrized in terms of the overlap, and for all configurations $\mathbf{s}$ such that $q_N(\mathbf{s},\mathbf{v}) = q$, we find:

$$
\begin{aligned}
\mathbb{P}_{\nabla_\perp \mathcal{E}}(\mathbf{0}) \to P_N^{(1)}(q) &= \left(\frac{2\pi\sigma^2}{(p-1)!}\right)^{-\frac{N-1}{2}} e^{-\frac{N}{2(p-1)!}\left(\frac{r}{\sigma}\right)^2 q^{2p-2}(1-q^2)}, \\
\mathbb{P}_{\mathcal{E}}(N\epsilon) \to P_N^{(2)}(\epsilon,q) &= \sqrt{\frac{p!}{2\pi N\sigma^2}}e^{-\frac{Np!}{2\sigma^2}\left(\epsilon+\frac{rq^p}{p!}\right)^2}.
\end{aligned}
\tag{106}
$$

Similarly, the expected value of the determinant (102) is only a function of the parameters $\epsilon$ and $q$, and we can use the notation:

$$\mathcal{D}_N(\epsilon,q) := \mathbb{E}\left[\left|\det\nabla_\perp^2 \mathcal{E}(\mathbf{s})\right|\right]_{\mathcal{E}=N\epsilon}. \tag{107}$$

Plugging everything into (100) we obtain

$$
\begin{aligned}
\mathbb{E}[\mathcal{N}_N(\epsilon,q)] &= \int_{\mathcal{S}_N(\sqrt{N})} d\mathbf{s}\,\delta(\mathbf{s}\cdot\mathbf{v}-Nq)\,\mathcal{D}_N(\epsilon,q)\,P_N^{(1)}(q)\,P_N^{(2)}(\epsilon,q) \\
&= S_N(q)\,\mathcal{D}_N(\epsilon,q)\,P_N^{(1)}(q)\,P_N^{(2)}(\epsilon,q),
\end{aligned}
\tag{108}
$$

where $S_N(q)$ is the surface of the $(N-2)$-dimensional hypersphere corresponding to the subspace of $S_N$ at overlap $q$ with the signal,

$$S_N(q) = \int_{S_N(\sqrt{N})} d\mathbf{s}\, \delta(\mathbf{s}\cdot\mathbf{v} - Nq) = \frac{2\pi^{\frac{N-1}{2}}}{\Gamma\left(\frac{N-1}{2}\right)}(1-q^2)^{\frac{N-2}{2}} \sim e^{\frac{N}{2}\log[2\pi e(1-q^2)]+o(N)}. \quad (109)$$

**(3) Large dimensionality and random matrix theory.** The statistical equivalence between the Hessians and the matrices of the form (103) implies that for large $N$

$$\mathcal{D}_N(\epsilon, q) = \mathbb{E}\left[\left|\det\left(\tilde{\mathbf{J}} - p\epsilon\mathbf{I} - r_{\text{eff}}[q]\,\mathbf{w}_\perp \mathbf{w}_\perp^T\right)\right|\right], \quad (110)$$

where the average is over random matrices $\tilde{\mathbf{J}}$ extracted from a GOE($\tilde{\sigma}^2$) with variance (104). We now exploit the RMT results of Section 2.2.2 to compute this determinant. Let us denote with $\lambda^1 \leq \cdots \leq \lambda^{N-1} = \{\lambda^\alpha\}_{\alpha=1}^{N-1}$ the eigenvalues of $\mathbf{M} = \mathbf{J} - r_{\text{eff}}[q]\,\mathbf{w}_\perp \mathbf{w}_\perp^T$. Then

$$\mathcal{D}_N(\epsilon, q) = \mathbb{E}\left[\prod_{\alpha=1}^{N-1}|\lambda^\alpha - p\epsilon|\right] = \mathbb{E}\left[e^{\sum_{\alpha=1}^{N-1}\log|\lambda^\alpha - p\epsilon|}\right] = \mathbb{E}\left[e^{(N-1)\int d\nu_{N-1}(\lambda)\log|\lambda - p\epsilon|}\right], \quad (111)$$

where we have introduced the eigenvalue distribution (32) of the matrix $\mathbf{M}$. The calculation is completed recalling two facts from Section 2.2.2:

1. In the large-$N$ limit, the leading-order contribution to the eigenvalue distribution is given by the continuous part, i.e., by the eigenvalue density $\rho_N(\lambda)$: isolated eigenvalues that may be generated by the rank-one perturbation, indeed, give corrections that are subleading in $1/N$. Therefore,

$$\mathcal{D}_N(\epsilon, q) = \mathbb{E}\left[e^{N\int \rho_N(\lambda)\log|\lambda - p\epsilon|+o(N)}\right] = \int \mathcal{D}\rho(\lambda)\mathcal{P}_N[\rho]\, e^{N\int d\lambda\, \rho(\lambda)\log|\lambda - p\epsilon|+o(N)}, \quad (112)$$

   where in the last equality we are using the fact that, given that the integrand depends only on the eigenvalue density, the average can be computed as an average over the probability distribution of the eigenvalue density $\rho_N(\lambda)$. For GOE matrices, the latter takes the large deviation form (56). The explicit form of the large deviation function $S[\rho]$ is not needed to proceed with the calculation: the only relevant ingredient is that the probability decays with speed $N^2$, i.e., much faster than the determinant term in (112), which behaves only exponentially in $N$. When computing (112) via the Laplace method for large $N$, the term proportional to $N^2$ dominates and must be optimized. This obviously selects the typical density $\rho_\infty(\lambda)$ as the optimizer.

2. The density $\rho_N(\lambda)$ is self-averaging, and its limiting value $\rho_\infty(\lambda)$ does not depend on the rank-one perturbation and it is given by the semicircle law $\rho_{\text{sc},\tilde{\sigma}}(\lambda)$. Therefore, to leading order in $N$:

$$\mathcal{D}_N(\epsilon, q) = e^{N\int d\lambda\, \rho_\infty(\lambda)\log|\lambda - p\epsilon|+o(N)} = e^{N\int d\lambda\, \rho_{\text{sc},\tilde{\sigma}}(\lambda)\log|\lambda - p\epsilon|+o(N)}. \quad (113)$$

The calculation is concluded by computing explicitly the integral at the exponent:

$$\begin{aligned} \int \rho_{\text{sc},\tilde{\sigma}}(\lambda)\log|\lambda - p\epsilon| &= \int d\lambda\, \frac{p!}{2\pi p(p-1)\sigma^2}\sqrt{\frac{4p(p-1)\sigma^2}{p!} - \lambda^2}\log|\lambda - p\epsilon| \\ &= \int d\mu\, \frac{\sqrt{2-\mu^2}}{\pi}\log\left|\sqrt{\frac{2p(p-1)\sigma^2}{p!}}\mu - p\epsilon\right| \\ &= \log\sqrt{\frac{2p(p-1)\sigma^2}{p!}} + I\left(\frac{p\epsilon}{\sqrt{\frac{2p(p-1)\sigma^2}{p!}}}\right), \end{aligned} \quad (114)$$

where $I(y)$ is an even function, that for $y \leq 0$ equals to

$$
I(y) = \int d\mu \frac{\sqrt{2-\mu^2}}{\pi} \log|\mu-y| = \begin{cases} \frac{y^2-1}{2} + \frac{y\sqrt{y^2-2}}{2} + \log\left(\frac{-y+\sqrt{y^2-2}}{2}\right), & y \leq -\sqrt{2}, \\ \frac{y^2}{2} - \frac{1+\log 2}{2}, & -\sqrt{2} < y \leq 0. \end{cases} \tag{115}
$$

We have now determined all the terms appearing in (108), to leading (exponential) order in $N$. This allows us to get the annealed complexity:

$$
\Sigma_A(\epsilon, q) = \lim_{N \to \infty} \frac{1}{N} \log\left[ S_N(q) \, \mathcal{D}_N(\epsilon, q) \, P_N^{(1)}(q) P_N^{(2)}(\epsilon, q) \right] \tag{116}
$$

$$
= \frac{1}{2} \log\left[ 2e(p-1)(1-q^2) \right] - \frac{r^2 q^{2p-2}(1-q^2)}{2(p-1)!\sigma^2} - \frac{p!\left(\epsilon + \frac{r}{p!}q^p\right)^2}{2\sigma^2} + I\left( \frac{p\epsilon}{\sqrt{\frac{2p(p-1)\sigma^2}{p!}}} \right).
$$

This expression coincides with that obtained in [80] by taking the annealed limit of the quenched complexity,[5] and it is consistent with the results of [81], where the annealed complexity of this model has been derived rigorously. From (116), one can check that the complexity vanishes for $p \to 2$, consistently with the results of Sec. 2 (see Box [B7]).

---

[B7] **The limit of quadratic landscapes: vanishing complexity.** For all energy densities $\epsilon$, the annealed complexity (116) is maximal at $q = 0$, that is, at the *equator*. We define $\Sigma_A(\epsilon) = \Sigma_A(\epsilon, q = 0)$. We now want to show the consistency of (116) with the results of Section 2 on the case of quadratic landscapes. We can then easily check from (116) that

$$
\Sigma_A(\epsilon) \xrightarrow{p \to 2} \frac{1}{2} \log(2e) - \frac{\epsilon^2}{\sigma^2} + I\left( \sqrt{\frac{2}{\sigma^2}} \epsilon \right) = 0, \tag{117}
$$

where we have used the fact that $\epsilon > -\sigma$ to choose the correct branch for $I$. This result is consistent with the fact that for $p = 2$, there are not exponentially many stationary points. Notice that one can use the Kac-Rice formula to recover (65): try it!

---

**The stability and the threshold.**  Let us now go back to our strategy discussed in Sec. 2.1.3. Within the annealed approximation, (116) gives the distribution of stationary points in energy density and geometry. What about the linear stability? As we have already remarked, the Hessian at a stationary point with parameters $\epsilon, q$ is a rank-one perturbed, shifted GOE (103). Such matrices have an eigenvalue density given by a shifted semicircle law, see Fig. 9; when $r_{\text{eff}}$ is large enough, in addition to the continuous density there is also an isolated eigenvalue, which is generated by the rank-one perturbation and reads:

$$
\lambda^{\text{iso}}(\epsilon, q) = -\frac{p(p-1)\sigma^2}{p! \, r_{\text{eff}}(q)} - r_{\text{eff}}(q) - p\epsilon. \tag{118}
$$

Because of the signs in (103), the isolated eigenvalue, when it exists, is the *smallest* eigenvalue of the Hessian. For the stationary point to be a *local minimum*, all the eigenvalues of the Hessian must be positive. Two conditions have to be met: (i) the semicircle law must be entirely supported on the positive semi-axis. This is guaranteed whenever $-p\epsilon > 2\sigma\sqrt{p(p-1)/p!}$:

---

[5]Ref. [80] discusses a generalization of the model considered in these notes. From [80], in order to get the model discussed in these notes one has to choose $k \to p$ and to redefine $r \to r/(p-1)!$. Moreover, [80] corresponds to the choice $\sigma^2 = p!/2$.

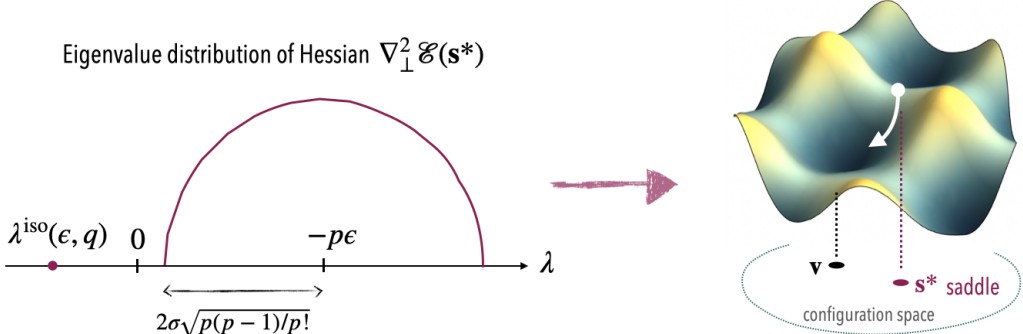

Figure 9: Distribution of eigenvalues of the Hessian $\nabla_\perp^2 \mathcal{E}(\mathbf{s}^*)$ at a stationary point $\mathbf{s}^*$ of energy density $\epsilon$ and overlap $q$ with the signal, for values of $r$ and $q$ for which an isolated eigenvalue exists. The stationary point is a saddle of index $\kappa = 1$, since all the eigenvalues are positive except the outlier. The BBP scenario implies that the direction of negative curvature of the landscape has an $O(1)$ overlap with the direction connecting $\mathbf{s}^*$ to the signal $\mathbf{v}$ in configuration space: the saddle is "geometrically connected" to the signal.

therefore, the positivity of the continuous part of the spectrum is guaranteed whenever the energy density of the stationary point satisfies:

$$\epsilon < \epsilon_{\text{th}} := -2\sigma \sqrt{\frac{1}{p!} \left( \frac{p-1}{p} \right)}. \tag{119}$$

The energy density $\epsilon_{\text{th}}$ is called the *threshold energy*. It marks a transition between saddles of extensive index $\kappa = O(N)$ for $\epsilon > \epsilon_{\text{th}}$, to minima (or saddles of index $\kappa = 1$ if the isolated eigenvalue exists and it is negative) for $\epsilon < \epsilon_{\text{th}}$. (ii) In addition, when an isolated eigenvalue $\lambda^{\text{iso}}(\epsilon, q)$ exists, it must also satisfy $\lambda^{\text{iso}}(\epsilon, q) > 0$. One can check that there are values of $\epsilon, q$ for which both these conditions are satisfied, and $\Sigma_A(\epsilon, q) > 0$: therefore, the average number of metastable states (local minima) is exponentially large in $N$.

### 3.2.3 The quenched complexity: A roadmap

The annealed complexity $\Sigma_A(\epsilon, q)$ gives some indications on the distribution of stationary points in the energy landscape, but such indications may be biased by rare events. To characterize typical properties of the landscape, one needs to compute the quenched complexity, making use of the replica trick (93) combined with the Kac-Rice formula (100). This type of calculation has been carried over in [80] precisely for the inference problem that we are discussing, and we refer to it as the *replicated Kac-Rice formalism*. Notice that this formalism can be employed also to settings where there is no random energy landscape, to count the quenched complexity of solutions of random dynamical equations of notion (or equilibria of the dynamics) [82–85]: this is relevant for complex systems arising in biological or ecological contexts, where typically the interactions between the degrees of freedom are *non-reciprocal*, rendering the dynamics non-conservative. Since the calculation of the quenched complexity is quite involved, in this Section we give just a few indications on what changes with respect to the calculation of the annealed complexity, and refer to [80] for details.

The starting point is a generalization of the Kac-Rice formula (100) to compute higher moments $n$ of $\mathcal{N}_N$. This involves introducing $n$ configurations $\mathbf{s}^a$ with $a = 1, \dots, n$ representing

as many stationary points, such that:

$$\mathbb{E}[\mathcal{N}^n] = \int_{\mathcal{S}_N^{\otimes n}} \prod_{a=1}^n d\mathbf{s}^a \, \delta(\mathbf{s}^a \cdot \mathbf{v} - Nq) \, \mathbb{E}^{(n)} \left[ \prod_{b=1}^n \left| \det \nabla_\perp^2 \mathcal{E}(\mathbf{s}^b) \right| \right]_{\substack{\nabla_\perp \mathcal{E}^a = 0 \\ \mathcal{E}^a = N\epsilon}} \mathbb{P}^{(n)}_{\nabla_\perp \mathcal{E}^a, \mathcal{E}^a}(\mathbf{0}, N\epsilon), \quad (120)$$

where now the expectation value of the product of determinant is conditioned to all the gradients and energies, $\nabla_\perp \mathcal{E}^a := \nabla_\perp \mathcal{E}(\mathbf{s}^a) = 0$ and $\mathcal{E}^a := \mathcal{E}(\mathbf{s}^a) = N\epsilon$ for all $a = 1, \ldots, n$, and the distribution is a *joint* probability distribution of the gradient and the energy landscape evaluated at all $n$ configurations. To get the quenched complexity, one has to estimate the leading-order term in large $N$ of this expression. What makes the calculation tricky is that the fields $\mathcal{E}(\mathbf{s}^a)$, $\nabla_\perp \mathcal{E}(\mathbf{s}^a)$ and $\nabla_\perp^2 \mathcal{E}(\mathbf{s}^a)$ are correlated with each others for different $a$. Let us mention some consequences of these correlations.

- *Correlated fields: No decoupling.* The gradient $\nabla_\perp \mathcal{E}(\mathbf{s}^a)$ for fixed $\mathbf{s}^a$ is independent of $\mathcal{E}(\mathbf{s}^a)$ and $\nabla_\perp^2 \mathcal{E}(\mathbf{s}^a)$, but not of $\mathcal{E}(\mathbf{s}^b)$ and $\nabla_\perp^2 \mathcal{E}(\mathbf{s}^b)$ for $b \neq a$. As a consequence, (i) one needs to compute joint distributions of these fields evaluated at all replicas, (ii) the Hessians are correlated, and thus computing the conditional expectation of their determinants becomes a problem of coupled random matrices. What helps for (i) is the Gaussianity. The crucial ingredient to deal with (ii) is the fact that one is interested in the large-$N$ limit of the expectation, and in this limit correlations turn out to be negligible (the joint expectation value of the product of determinants factorizes into a product of expectation values [80, 86]).

- *Breaking (slightly more) isotropy.* When more configurations are introduced, the distributions of the fields evaluated at the points does not depend only on $q_N(\mathbf{s}^a, \mathbf{v})$ but also on the mutual overlaps $Q_N^{ab} := q_N(\mathbf{s}^a, \mathbf{s}^b)$. These overlaps are *order parameters* of the complexity calculation, and the quenched complexity has to be evaluated by solving an optimization problem over these parameters. This illustrates that in the quenched problem, there is no longer a single special direction (the signal), but $n+1$ special directions (the signal, and the configuration $\mathbf{s}^a$). Despite this, the problem still shows a significant dimensionality reduction, typical of mean-field problems: while in principle the expression (120) depends on $Nn$ distinct variables $s_i^a$, in fact when $N$ is large it can be parametrized in terms of only $n(n-1)/2 + n$ quantities, that are the overlaps.

- *Symmetry breaking?* The quenched calculation has several order parameters, the overlaps $Q^{ab}$, that are fixed by solving an optimization problem (arising from a saddle point calculation). What is the good space in which to seek for the optimizer? Should one look for solutions in which the $Q^{ab}$ are symmetric with respect to permutations of the indices $a$ labeling replicas, i.e., for replica symmetric solutions? Or is this symmetry broken at the optimizer? This questions naturally leads to the issue of *Replica Symmetry Breaking* (RSB) [5]. For the implementation of interesting patterns of RSB within the complexity calculation, see [87, 88].

- *A Hessian with higher-rank perturbations.* In the quenched problem, the conditional distribution of the Hessian at one point $\mathbf{s}^a$ still follows a perturbed GOE distribution; however, the finite-rank perturbations have a more complicated structure, that arises due to the conditioning of the Hessian $\nabla_\perp^2 \mathcal{E}(\mathbf{s}^a)$ to $\nabla_\perp \mathcal{E}(\mathbf{s}^b) = 0$ for $b \neq a$. In other words, the quenched calculation encodes correlations, and it accounts for the fact that the curvature of the landscape at one stationary point $\mathbf{s}^a$ is affected by the presence of other stationary points $\mathbf{s}^b$ in its vicinity. The finite-rank perturbations to the GOE in the quenched case are both additive and multiplicative. The calculation of the explicit form of the isolated eigenvalue, when it exists, is more involved [80], but it is remains doable because the perturbation is still of finite rank, not scaling with $N$.

## 3.3 What: Ground state, metastability, dynamics

We now exploit the information we have gathered on the distribution of stationary points to address the questions of Sec. 2.1.3. Even though in these notes we computed only the annealed complexity, in the following we discuss the picture that emerges from the calculation of the quenched complexity [80]. We focus on $q \geq 0$, i.e., on the region of configurations space that corresponds to positive overlap with the signal.

### 3.3.1 Q1. A discontinuous recovery transition

As for the quadratic case, recovery with maximum likelihood becomes possible when $\mathbf{s}_{GS}$ acquires a positive asymptotic overlap with $\mathbf{v}$, i.e., when (16) holds true. For $p \geq 3$, this again occurs at a critical value of the signal-to-noise ratio, which we denote with $(r/\sigma)_{1st}$. This notation refers to the fact that, at variance with the quadratic case, for higher-order landscapes this transition is *discontinuous*, see Fig. 10 (*Left*): for values of $r/\sigma$ below the critical threshold, the ground state[6] is at the equator, orthogonal to $\mathbf{v}$, and the recovery is impossible. At $(r/\sigma)_{1st}$, the ground state jumps in the northern hemisphere, acquiring a positive overlap with the signal. This is therefore akin to a first order phase transition, in which the order parameter $q_\infty(\mathbf{s}_{GS}, \mathbf{v})$ jumps from zero to a finite value. The value of $(r/\sigma)_{1st}$ can be determined from $\Sigma_\infty(\epsilon, q)$: for each $q$, the quenched complexity vanishes at a value of $\epsilon_{min}(q)$ that corresponds to the minimal energy density of stationary points at that $q$; the energy density of the ground states is the minimum of this curve, and $q_\infty(\mathbf{s}_{GS}, \mathbf{v})$ the minimizer. For $p = 3$, the numerical value of the recovery threshold is $(r/\sigma)_{1st} \approx 2.56$. The comments we made for the case $p = 2$ extend also to the tensorial case: (i) the order parameter is like a magnetization in the direction of a generalized magnetic field given by $\mathbf{v}$. The recovery transition can be obtained with an equilibrium calculation, computing the free energy of the system as done in [89]; it corresponds to a "ferromagnetic" transition occurring at $\beta \to \infty$. However, at variance with the $p = 2$ case, in the low-$r$ phase the system is not a "ferromagnet in disguise", but it is a true *spin-glass*: the Boltzmann measure is partitioned into a number of states larger than two, and they are not related by symmetry. (ii) Also in the tensor case, the recovery threshold achieved by maximum likelihood is *optimal*—that is, it coincides with the *detection threshold*—if a spherical prior is assumed on the signal $\mathbf{v}$ [90, 91]. As for the matrix case, this is not expected to be generic under more structured assumptions on the statistics of the signal.

### 3.3.2 Q2. A landscape made of local minima

We now discuss the distribution of the stationary points at energy density higher than the ground state, as it emerges from the calculation of the quenched complexity $\Sigma_\infty(\epsilon, q)$. First, one finds that similarly to the annealed complexity, the quenched complexity is positive in a whole range of parameters $\epsilon, q$: in *typical* realizations of the landscape there are exponentially many stationary points. However, for general values of $\epsilon$ and $q$, quenched and annealed complexity do not coincide. Let us now comment on the distribution of the exponentially-many stationary points counted by the quenched complexity.

- *Most stationary points are at the equator: they are uninformative of* $\mathbf{v}$. The complexity $\Sigma_\infty(\epsilon, q)$ is maximal at $q = 0$, i.e., the exponential majority of the stationary points are at the equator, orthogonal to the signal (not informative). At the equator the quenched complexity coincides with the annealed one:

$$\Sigma_\infty(\epsilon) = \max_q \Sigma_\infty(\epsilon, q) = \Sigma_\infty(\epsilon, q = 0) = \Sigma_A(\epsilon, q = 0). \tag{121}$$

---

[6]Henceforth we talk about a single ground state. However, recall that for $p$ even the energy landscape is an even function, and thus it admits two degenerate ground states, as we have seen for $p = 2$.

In this region of configuration space, the distribution of stationary points and their properties (such as the curvature of the landscape around them) do not depend on $r$: the landscape at the equator is exactly the one that one would find for the purely random model at $r = 0$. On the contrary, when $q > 0$ the properties of the stationary points depend on the strength of the signal $r$, as we argue below.

- *There are exponentially many local minima: the majority are marginally stable.* The characterization of the statistics of the Hessian at stationary points has revealed that the latter are minima whenever (119) is satisfied, and the isolated eigenvalue (if it exists) is positive. The analysis of $\Sigma_\infty(\epsilon, q)$ shows that the number of stationary points with energy density smaller than the threshold one $\epsilon_{\text{th}}$ is exponentially large in $N$. Most of these stationary points (in particular, all those at the equator $q = 0$) have an Hessian with no isolated eigenvalue. Therefore, the energy landscape has exponentially many local minima that can trap the dynamics. Among them, the most numerous (i.e., those associated with the highest complexity) are such that $q = 0$ and $\epsilon = \epsilon_{\text{th}}$. These stationary points are *marginally stable*: the eigenvalue distribution of their Hessian is a semicircle that touches zero with the lower edge of its support, meaning that the stationary points are at the *boundary of stability*. Marginality plays a crucial role in the physics of glassy systems [92, 93], as we shall see also below when discussing the dynamics of this model.

- *Local minima in the northern hemisphere undergo a stability transition with $r$.* Stationary points with $\epsilon < \epsilon_{\text{th}}$ are such that the Hessian has all eigenvalues positive, except possibly the isolated eigenvalue. Let us now discuss the behavior of the latter. When $q = 0$, $r_{\text{eff}}[q]$ in (105) is equal to zero: therefore, stationary points at the equator have no isolated eigenvalues. For $q > 0$ instead, when $r$ becomes large enough, an isolated eigenvalue is generated, and it becomes negative at a critical value of $r$ that depends on $\epsilon, q$. This marks a stability transition for the stationary point, that go from being minima to being saddles with index $\kappa = 1$, i.e., with one single negative eigenvalue of the Hessian as in Fig. 9 (*Left*). From the discussion of the BBP transition in Sec. 2.2.2, we also conclude that when the isolated eigenvalue exists, the corresponding eigenvector is localized in the direction of the signal $\mathbf{v}$. Therefore, the direction of negative curvature of the saddle is aligned with the direction connecting the saddle and the signal in configuration space, see Fig. 9 (*Right*). The instability is *towards* the signal, and we say that the saddle is *geometrically connected* to $\mathbf{v}$. When $r$ is large enough, the landscape at large $q$ has no trapping local minima, since all metastable states have been destabilized by the signal with this mechanism.

- *The landscape becomes topologically trivial at extensively large signal-to-noise ratio.* All the above analysis is performed assuming that $r = O(N^0)$ and taking the limit $N \to \infty$. In this setting, the local minima at the equator are never destabilized by the isolated eigenvalue, and the landscape remains rugged at all values of $r$. One may however consider much stronger signal strengths, $r \to r(N) \sim N^\alpha$, and ask: are there values of $\alpha$ for which the signal is so strong, that it destabilizes also the local minima at the equator? This can be figured out with a simple scaling argument. For finite $N$, the overlap between a random configuration $\mathbf{s}$ and the fixed signal vector $\mathbf{v}$ typically scales as $q_N(\mathbf{s}, \mathbf{v}) \sim N^{-1/2}$, see Box [B2]. Therefore, for configurations $\mathbf{s}$ extracted uniformly on $\mathcal{S}_N(\sqrt{N})$ it holds:

$$r_{\text{eff}}(q_N) = r p(p-1) \left(\frac{1}{\sqrt{N}}\right)^{p-2} \left(1 - \left(\frac{1}{\sqrt{N}}\right)^2\right) \sim r N^{-\frac{p-2}{2}}. \qquad (122)$$

In order for such random configurations to be destabilized by the signal through a BBP transition, one needs $r_{\text{eff}} = O(1)$, which is attained scaling $r(N) \sim N^{\frac{p-2}{2}}$. Therefore, for

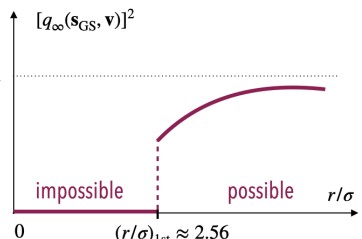 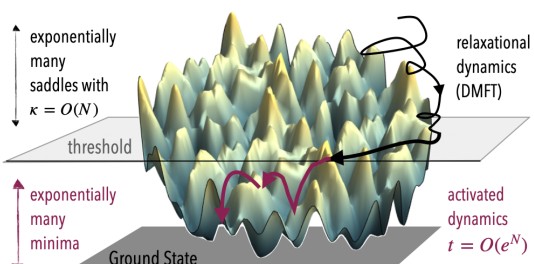

Figure 10: *Left.* Squared overlap of the ground state with the signal as a function of the signal-to-noise ratio, for a higher-order landscape with $p = 3$. The recovery transition is discontinuous. *Right.* Sketch of the behavior of the dynamics initialized randomly, for $p \geq 3$: mean-field, "short-time" dynamics describes the descent from $\epsilon = 0$ to $\epsilon = \epsilon_{\text{th}}$, in the region of the landscape dominated by saddles of extensive index. The "long-time" dynamics beyond mean-field describes the exploration of the bottom of the energy landscape, dominated by local minima, via activated processes.

$\alpha > \alpha_c = (p - 2)/2$, the signal term is strong enough to destroy all metastable local minima in the landscape. In the terminology of [94], the signal leads to a *topological trivialization* of the landscape.

### 3.3.3 Q3. A "hard" landscape optimization

The energy landscape associated to the low-rank tensor estimation problem for $r = O(N^0)$ is *rugged*, with an exponentially large number of metastable states, the majority of which are at the equator, uninformative of the signal. One expects therefore the optimization of the landscape to be hard, such that $\tau_{\text{eq}} \sim e^N$, when the system is initialized in a configuration $\mathbf{s}_0$ extracted with uniform distribution on the hypersphere. In fact, this is the case. As we know, with overwhelmingly high probability (converging to one when $N \to \infty$) the random initial condition lies at the equator, orthogonal to the signal $\mathbf{v}$, where the landscape has the same statistical properties as in the $r = 0$ case. In analogy to the case $p = 2$, we now discuss properties of the Langevin dynamics at $\beta \to \infty$ starting from such random initial condition, distinguishing between short and at large timescales.

■ **Short timescales: Dynamical Mean Field Theory (DMFT).** As for the case of quadratic landscapes, the dynamics in the mean-field limit is essentially the same that one has for the $r = 0$ model, i.e., for the pure spherical $p$-spin model. The DMFT equations for this model have been studied quite extensively in the literature, in particular in [95–97]. Let us summarize some features.

- *Far from the ground state: aiging at the threshold.* The most striking difference with respect to the $p = 2$ case is that the excess energy (68) does not decay to zero (at $\beta \to \infty$), but instead converges to a finite value:

$$\lim_{t \to \infty} \lim_{N \to \infty} \Delta \epsilon_N(t) = \lim_{t \to \infty} \lim_{N \to \infty} (\epsilon_N(t) - \epsilon_{\text{GS}}) = \epsilon_{\text{th}} - \epsilon_{\text{GS}} > 0. \tag{123}$$

This means that over the timescales described by DMFT, the system never reaches the energy density of the ground state, but it visits regions of the landscape at extensively higher energy, approaching asymptotically the marginally stable minima at $\epsilon_{\text{th}}$. This

phenomenon, that was understood in the early works [95], has been also recently rigorously proven [98]. As for the $p = 2$ case, the DMFT equations describe a relaxational dynamics that is out-of-equilibrium, characterized by phenomena such as aging.

- *Landscape interpretation.* Again, this out-of-equilibrium aging dynamics, as well as the convergence to the threshold energy density, have a landscape interpretation: as we saw in Sec. 3.2.2, $\epsilon_{\text{th}}$ is the energy density of the most numerous metastable states; it is also the energy density where a transition occurs in the stability of the stationary points at $q = 0$: for $\epsilon > \epsilon_{\text{th}}$, stationary points are saddles with extensive index $\kappa = O(N)$, while for $\epsilon \leq \epsilon_{\text{th}}$ they are local minima (at $q = 0$, there is no isolated eigenvalue that can turn these minima into saddles). Therefore, the DMFT equations describe how the system starting from random $\mathbf{s}_0$ descends in the region of the energy landscape dominated by extensive-index saddles, approaching asymptotically the energy density where local minima start to appear, see Fig. 10 (*Right*). While for $p = 2$ this energy density coincides with that of the ground state, for $p \geq 3$ this coincides with $\epsilon_{\text{th}} > \epsilon_{\text{GS}}$.

- *A paradigmatic solution.* For the pure spherical $p$-spin model, we also know *how* the system approaches asymptotically the threshold energy with its out-of-equilibrium dynamics. This comes from the fact that an *analytic solution* of the DMFT equations in the large time limit has been found in [95]. The form found in [95] solves the equations up to a reparametrization of time [99]. Although some aspects of this solution may not generalize straightforwardly to other glassy energy landscapes [100], this result has profoundly shaped our understanding of relaxational out-of-equilibrium dynamics in high-dimensional systems, serving as a framework for developing key concepts such as timescales separation, weak ergodicity breaking, aging, effective temperatures, the violation of the fluctuation-dissipation theorem, and quasi-equilibrium dynamics [16, 57].

■ **Large timescales: Dynamics at finite $N$.** While for $p = 2$ the dynamics at large timescales (diverging with $N$) is quite well understood thanks to RMT, for $p \geq 3$ characterizing this regime of the dynamics remains an open problem. Consider $\beta \gg 1$, i.e., weak noise. Because of the presence of metastability, the large time dynamics is expected to be markedly different with respect to the quadratic case, and to be characterized by a sharp *separation of timescales*: the system spends long time trapped into one metastable state (local minimum), performing fluctuations within the basin; these large windows of time are interspersed by fast transitions or *jumps* from the local minimum to another one. These jumps are *activated events*: to transition from one minimum to another, the system has to cross the high energy barriers $\Delta\mathcal{E} = O(N)$ separating them. These escape processes are allowed by the noise in the equation (18), which counteracts the action of the gradient and allows the system to climb up in the landscape (instead of descending). When the noise is weak, these *escape processes* are *rare*: for stochastic dynamics, the typical timescale associated to these jumps follows an *Arrhenius scaling*, i.e., it is exponentially large in the energy barrier, $\tau_{\text{typ}} \sim e^{\beta \Delta\mathcal{E}} \sim e^{\beta N \Delta\epsilon}$. Therefore, to study activated processes, one has to study the dynamics at timescales growing exponentially with $N$. This is a challenge, that is also connected to large deviation theory, as we discuss in Sec. 4.

■ **Cranking-up the signal: The "easy" phase.** When $r = O(N^0)$, the dynamics from random initial conditions is insensitive to the signal for a huge range of timescales; in fact, it is stuck in the high-entropy region that is orthogonal to the signal, the equator: there, the landscape behaves as if there was no signal. As we argued in Sec. 3.3.2, when the signal-to-noise ratio scales with $N$ as $r(N) \sim N^\alpha$ with $\alpha > \alpha_c = (p-2)/2$, the energy landscape is completely modified by the signal: metastability is destroyed, and the landscape is topologically trivial.

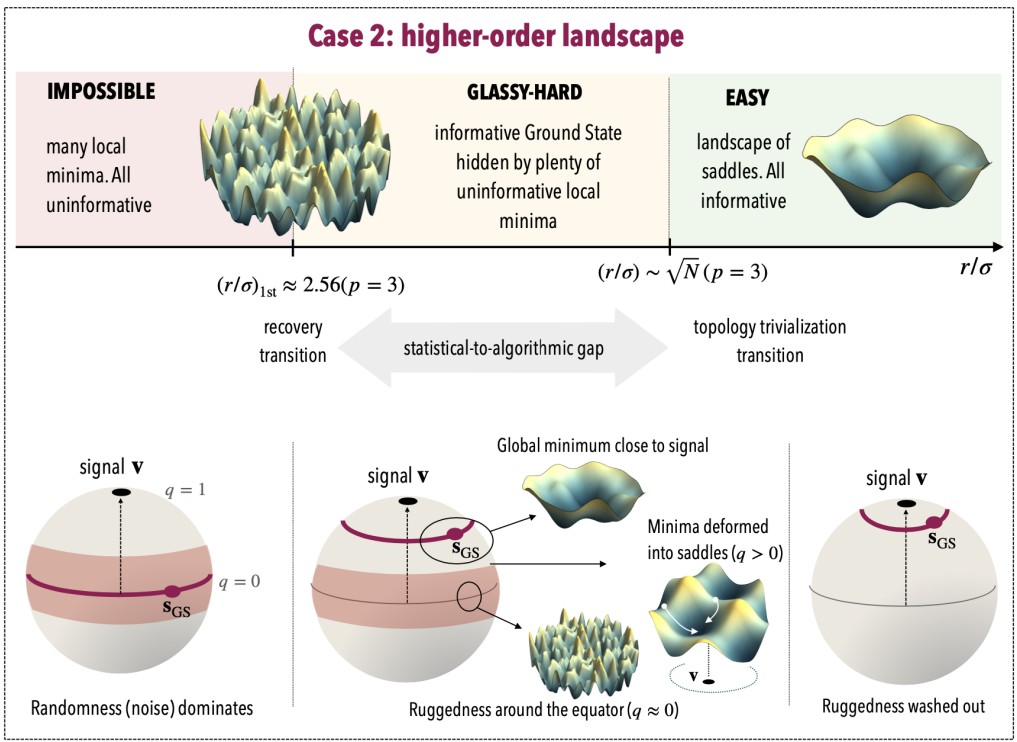

Figure 11: Summary of the phenomenology emerging from the landscape analysis in case 2.

In this regime, gradient descent dynamics converges to $\mathbf{s}_{GS}$ in times that are of $O(N^0)$ [101], showing clearly the tight link between landscape's structure and optimization dynamics.

The spiked-tensor problem is therefore a prototypical example of hard inference problem exhibiting a *statistical- to-algorithmic gap*: for a window of values of the signal-to-noise ratio (a window with width scaling with $N$), the inference problem is not impossible – it can be solved by optimizing the landscape, but it is algorithmically hard – finding the solution requires timescales exponentially large in the dimensionality of the signal. As in the matrix case, we have focused here on a specific optimization algorithm, Langevin dynamics (or gradient descent). Other algorithms can be considered, which may outperform Langevin. In fact, for the spiked tensor problem, it is known that there exist algorithms that recover the signal at values of the signal-to-noise ratio that scale with $N$ with a smaller power than the $\alpha_c = (p-2)/2$ required by Langevin [9, 64]. With this, we conclude our analysis of the low-rank tensor estimation problem. A summary of the phenomenology emerging from the landscape analysis in case 2 is given in Figure 11.

# 4 Escaping metastability in high dimension: The landscape program meeting large deviation theory

We conclude these notes by discussing how the landscape program summarized in Fig. 1 connects with Large Deviation Theory (LDT). Within the landscape program, one of the most natural applications of LDT is the identification and characterization of families of metastable states that are atypical, but may play a crucial role in shaping the system's behavior. In recent years, this has shown to be relevant in problems of computer science and machine learning [102]. Strong connections to LDT emerge also when trying to understand optimization

dynamics in rugged landscapes at the largest timescales: these connections are the focus of the remainder of this Section. As we mentioned in Sec. 3.3.3, these dynamics (for $p \geq 2$) are *dominated by activated processes*, that in the high-dimensional setting are *rare events* associated to an exponentially small (in $N$) probability of occurrence. These events, which describe transitions from metastable state to metastable state, despite being rare are the driving mechanism for the dynamical exploration of the rugged landscape. Understanding them would allow us to really characterize high-dimensional dynamics beyond the relaxational regime, for which we have an established and versatile theory (DMFT). Theoretically, it is also a playground to understand glassy dynamics beyond the mean-field approximation.

## 4.1    Rare dynamical events: Activated processes

Activated processes are jumps from local minimum to local minimum of the landscape, see Fig. 10 (*Right*), which involve climbing up in the energy landscape and crossing an *energy barrier*. In particular, a single activated process can be pictured as a combination of a fluctuation path (a transition from a minimum to an index-1 saddle) and a relaxation path (a transition from the saddle down to another minimum), as shown in Fig. 12 (*Right*). These transitions are allowed by the noise in the dynamics (18), much like in the well-known Kramers escape problem [103,104]. The main challenge in the high-dimensional context lies in the huge entropy (or, in the language of these notes, the huge complexity) associated with the problem: there is a proliferation of metastable states that can be accessed, as well as a multitude of saddles connecting these states. In a way, one is dealing with a complicated version of the escape problem in a double-well potential, in which the number of wells (and barriers between them) is exponentially large in $N$. This raises several questions:

(i) which paths are selected by the dynamics? Are the lowest energy barriers separating minima the relevant ones, or entropic effects prevail?

(ii) What sequence of activated jumps is necessary to fully decorrelate the system, i.e., to reach a state at zero overlap with the initial configuration? What is the effective energy barrier for these transitions?

(iii) How do these processes occur? For instance, what is the shape of the correlation function along these activated trajectories?

Activated dynamics in high dimension has been studied through phenomenological models such as the *trap model* [105,106], which describes dynamics as a random walk between "traps" (the metastable states) with randomly assigned energies. Transitions between the traps occur at rates depending solely on the energy of the departing trap, based on the assumption that escaping requires reaching a fixed *threshold* energy level (often set to zero for simplicity, and identifiable with the threshold energy (119) in the $p$-spin model). Energy barriers are the difference between this threshold and the trap's energy, and the transition rate is an exponential in this barrier. Once the threshold level is reached, any other trap is accessible (the trap network is fully-connected). Trap-like dynamics is renewal: at the threshold the system looses memory of the trap it was coming from, and the dynamical process starts again independently of the past history. This model is exactly solvable [107,108], and it shows paradigmatic features of glassy dynamics such as aging and the emergence of effective temperatures.[7] The trap model provides the correct asymptotic description of Metropolis dynamics in the Random Energy Model [109], where the energy landscape is devoid of correlations (corresponding to the $p \to \infty$ limit of the $p$-spin model) [110]. It is not completely clear to what extent the activated

---

[7]Notice that here aging is rooted in extreme value statistics, a different mechanism with respect to the landscape-based geometrical interpretation we discussed for relaxational dynamics in Secs. 2.3 and 3.3.

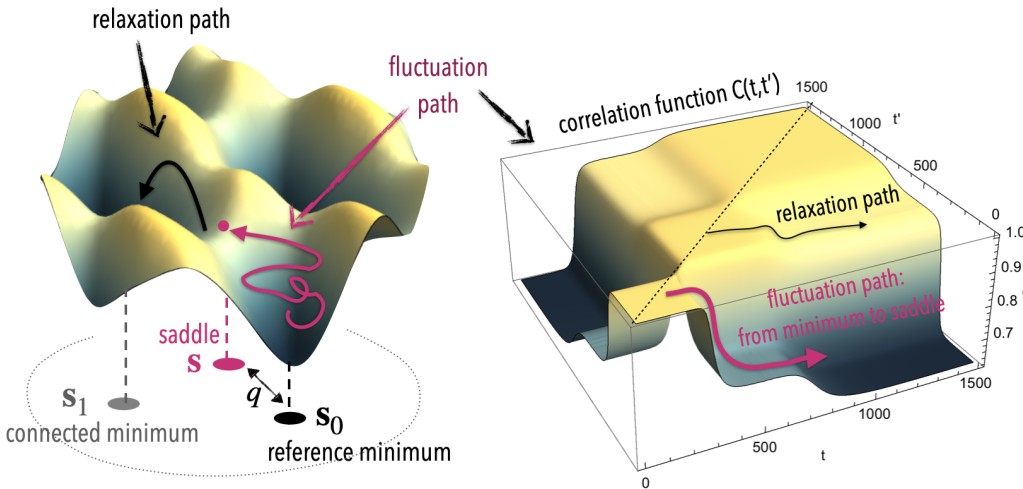

Figure 12: *Left.* Two local minima $\mathbf{s}_0$ and $\mathbf{s}_1$ connected by a saddle $\mathbf{s}$ of index 1. The activated path between them can be decomposed into a fluctuation path and a relaxation path. *Right.* Correlation function along an approximate instantonic trajectory. Figure taken from [118].

regime of Langevin dynamics in the $p$-spin model falls into the trap-like framework [111,112]. The mapping is not straightforward: minima of the landscape have non-trivial connectivity properties in configuration space, and the height of the barrier to be crossed to escape from a minimum may be correlated non-trivially to the energy of the minimum. With this motivation, extensions of the trap model to sparse networks [113] and to different transition rates between the traps [114,115] have been investigated.

Let us now briefly review some attempts to gain more information on these processes, which rely on the direct investigation of the landscape and of the dynamics of the model (83) with $r = 0$, the pure spherical $p$-spin model. Notice that even for $r > 0$, the landscape at the equator is exactly described by the $r = 0$ model. Since random initial conditions of the dynamics typically lie at the equator, it is natural to expect that Langevin dynamics is trapped by the metastable states at the equator for a huge range of timescales. Therefore, studying activated dynamics for the $r = 0$ model should be of relevance to understand optimization dynamics at finite $N$ for the spiked tensor model as well, at least for a large window of times.

## 4.2 Landscape's local geometry, and large deviations

A key question to understand activated dynamics is the connectivity of minima and saddles within configuration space: given a particular minimum $\mathbf{s}_0$ of energy density $\epsilon_0 < \epsilon_{\text{th}}$, (i) how many index-1 saddles are *geometrically connected* to this minimum, see Fig. 12 (*Left*)? (ii) What are the distances (overlaps) between the minimum and these saddles? (iii) How are the energy densities of the saddles distributed, i.e., what is the *distribution of energy barriers* surrounding the minimum? These questions define what we refer to as the *local geometry* of the energy landscape. For the spherical $p$-spin model, they have been addressed in [116,117], through the calculation of the complexity of saddles that are geometrically connected to reference minima $\mathbf{s}_0$ at arbitrary density $\epsilon_0$ below the threshold.

**Large deviations for the landscape curvature: A perturbed GOE ensemble.** The connection to LDT is due to the entropic nature of the problem. Below the threshold energy, $\epsilon < \epsilon_{\text{th}}$, the energy landscape is overwhelmingly dominated by minima, see Fig. 10 (*Right*). At those

values of energy, the local minima are exponentially more numerous than the saddles of finite index $\kappa = O(N^0)$: if a stationary point is selected at random, it is almost certainly a minimum. However, below the threshold energy the saddles with finite rank have a positive complexity, even though smaller than that of local minima [72]. The number of saddles of energy density $\epsilon$ is suppressed with respect to that of minima by an exponential factor in $N$, that is precisely the probability that a stationary point of energy density $\epsilon < \epsilon_{\text{th}}$ has one (or more) negative eigenvalue(s) of the Hessian. This probability is a large deviation probability related to the smallest eigenvalue(s) of a GOE matrix, the Hessian. Indeed, the statistics of the Hessian at a stationary point is given by (103). When $r = 0$ and $\epsilon < \epsilon_{\text{th}}$, there is no isolated eigenvalue and the eigenvalue density is a shifted semicircle law supported on the positive semi-axis, so *typically* the stationary points are minima. The probability to get a saddle is the likelihood that the smallest eigenvalue of the Hessian deviates from the boundary of the semicircle and becomes an outlier, taking a value smaller than zero. This large deviation probability is well known for GOE matrices [31], as recalled in Sec. 2.2.2.

The problem we are dealing with is however more complicated, since we are interested in computing the distribution of saddles *close to a given minimum in configuration space*. We assume that $\epsilon_0$ is the energy density of the minimum $\mathbf{s}_0$, and that we want to track saddles $\mathbf{s}$ of energy $\epsilon$ that are at overlap $q$ with the minimum, see Fig. 12 (*Left*). This proximity constraint is implemented as a conditioning in the counting of the saddles, that modifies the statistics of the Hessian at $\mathbf{s}$: the latter turns out to be a shifted GOE with both an additive (as in the spiked problem) and a multiplicative rank-1 perturbation [116]. Its statistics is equivalent to that of the $(N-1) \times (N-1)$ matrices:

$$\tilde{\mathbf{H}}(\mathbf{s}) := \mathbf{H}(\mathbf{s}) - p\epsilon_N(\mathbf{s})\mathbf{I} = \mathbf{F}[q_N]\tilde{\mathbf{J}}\,\mathbf{F}[q_N] + \left(\mu[\epsilon_N(\mathbf{s}), q_N|\epsilon_N(\mathbf{s}_0)] + \frac{\zeta[q_N]\xi}{\sqrt{N}}\right)\mathbf{e}\,\mathbf{e}^T - p\epsilon_N(\mathbf{s})\mathbf{I}, \quad (124)$$

where (i) $\tilde{\mathbf{J}}$ is a GOE with moments (104), (ii) $\mathbf{F}[q_N]$ is a rank-1 multiplicative perturbation:

$$\mathbf{F}[q] \equiv \mathbf{I} - \left(1 - \frac{\Delta[q]}{\tilde{\sigma}}\right)\mathbf{e}\,\mathbf{e}^T, \qquad \tilde{\sigma} = \sqrt{\frac{p(p-1)}{p!}}\sigma, \quad (125)$$

evaluated at $q = q_N(\mathbf{s}, \mathbf{s}_0)$, (iii) $\xi$ is a Gaussian variable with zero average and unit variance that is independent of $\tilde{\mathbf{J}}$, (iv) $\Delta(q), \mu[\epsilon, q|\epsilon_0]$ and $\zeta[q]$ are functions given explicitly in [116], (v) the direction of the finite rank perturbation is

$$\mathbf{e} = \frac{1}{\sqrt{N(1-q^2)}}(\mathbf{s}_0 - q\mathbf{s}), \quad (126)$$

namely it is the projection on the tangent plane $\tau[\mathbf{s}]$ of the vector connecting the reference minimum $\mathbf{s}_0$ to the stationary point $\mathbf{s}$. The perturbations to the GOE arise when enforcing that $\mathbf{s}$ has to be close to $\mathbf{s}_0$ in configuration space, and disappear when lifting this constraint: indeed, $\zeta[q], \mu[\epsilon, q|\epsilon_0] \xrightarrow{q \to 0} 0$ and $\Delta[q] \xrightarrow{q \to 0} \tilde{\sigma}$, meaning that when $q \to 0$ (the typical value of the overlap between stationary points in the landscape) the statistics of the Hessian reduces to the standard GOE. We define $\tilde{\sigma}' = \sqrt{\tilde{\sigma}^2 - \Delta^2[q]}$, which vanishes when $q \to 0$.

The spectral properties of the perturbed GOE ensemble (124) have been analyzed in the works [116, 117, 119]. In [116, 117] we have shown that the BBP transition for this ensemble occurs when

$$\mu[\epsilon, q|\epsilon_0] < -\tilde{\sigma}\left(1 + \left(\frac{\tilde{\sigma}'}{\tilde{\sigma}}\right)^2\right), \quad (127)$$

and the typical value of the isolated eigenvalue of $\mathbf{H}(\mathbf{s})$ reads

$$\lambda_\infty^{\text{iso}} = g_{\text{sc},\tilde{\sigma}}^{-1}\left(g_{\text{sc},\tilde{\sigma}'}(\mu)\right) = \frac{1}{g_{\text{sc},\tilde{\sigma}'}(\mu)} + \tilde{\sigma}^2 g_{\text{sc},\tilde{\sigma}'}(\mu), \quad (128)$$

where $g_{sc,\nu}(z)$ is the Stieltjes transform (41) of a GOE($\nu^2$). Comparing (128) with (47), one notices the same structure with $r \to 1/g_{sc,\tilde{\sigma}'}(\mu)$. The additional Stieltjes transform is the effect of the multiplicative rank-1 perturbation (125): when $q \to 0$ and $\Delta[q] \to \tilde{\sigma}$, the multiplicative part of the perturbation vanishes and indeed one finds

$$\lim_{\tilde{\sigma}' \to 0} g_{sc,\tilde{\sigma}'}(\mu) = \frac{1}{\mu}, \tag{129}$$

recovering the case of a purely additive perturbation (47) of strength $\mu$. The joint large deviations of the smallest eigenvalue and eigenvector of the matrices (124) have been determined in [117], generalizing the results of [39] for the purely additive case. The emergence of different regimes of the large deviation function is again interpretable in terms of a BBP transition of the second-smallest eigenvalue. Tracking the eigenvector projection in the direction of $\mathbf{e}$ is important for our problem, as it allows to understand whether the saddle is geometrically connected to the reference minimum, or not.

**On the statistics of barriers.** The combination of these approaches enables one the study of the complexity of rank-1 saddles surrounding a given local minimum, both in regions where they are typical [116] and in those where they are atypical [117] with respect to local minima. In this way, one has access to the *statistics of the energy barriers* for local activated jumps out of a minimum, as a function of the energy of the minimum itself. We remark that these barriers separate local minima that are *close* to each others in configuration space. The effective energy barrier separating distant minima can be much higher, as the two minima may be connected by a sequence of local jumps. A proxy of it can be obtained studying the energy profile along predefined paths interpolating between distant minima in configuration space [120]. We also remark that, given the reference minimum $\mathbf{s}_0$ and one among the exponentially many saddles that are connected to it, one can determine the statistical properties of the local minima $\mathbf{s}_1$ connected to $\mathbf{s}_0$ via the saddle, see Fig. 12: this is done with a dynamical approach, studying the solutions of some modified Dynamical Mean Field Theory (DMFT) equations where one enforces the saddle as an initial condition [118]. Finally, we remark that local correlations between stationary points in the landscapes potentially affecting activated dynamics can be characterized further, by studying the complexity of *triplets* of them [121].

## 4.3 Dynamical instantons, and large deviations

The Dynamical Mean Field Theory (DMFT) equations characterize the typical behavior of correlation functions, energy density etc. along dynamical trajectories. These equations emerge from a large-$N$ limit, as a result of a saddle-point calculation. Formally, the DMFT equations are derived from a *dynamical partition function* $\mathcal{Z}_{\mathbf{s}_0}^{\mathrm{dyn}}$ associated to the Langevin equation (18): this partition function sums up the probabilities of all possible trajectories originating from a given initial condition, $\mathbf{s}_0$, as in path integrals. In the limit of large $N$, the average of the dynamical partition function over both the noise of the dynamics and the realizations of the random landscape takes the form [122]:

$$\mathcal{Z}_{\mathbf{s}_0}^{\mathrm{dyn}} = \int_{\mathbf{s}(t=0)=\mathbf{s}_0} \mathcal{D}\mathbf{s}_t \, e^{-\mathcal{A}_N[\mathbf{s}_t]} \xrightarrow{\mathbb{E}[\cdot]} \int \mathcal{D}\mathbf{Q} \, e^{-N\,\mathcal{A}[\mathbf{Q}(t,t')]+o(N)}, \tag{130}$$

where $\mathbf{Q}(t,t')$ denotes the collections of *dynamical order parameters* of the theory (among which the correlation function, the energy density etc). The functional $\mathcal{A}[\mathbf{Q}(t,t')]$ can be interpreted as a large deviation functional. Typical processes are those that dominate the functional integral (130), and thus are obtained taking the variation

$$\frac{\delta\mathcal{A}}{\delta\mathbf{Q}} = 0 \implies \text{DMFT equations.} \tag{131}$$

The DMFT equations are therefore the equations for the saddle point of a *dynamical action*, describing its minimizer. Clearly, when evaluated at the solutions of the DMFT equations the action vanishes, consistently with the fact that one is describing typical properties occurring with a probability $\mathbb{P} \xrightarrow{N\to\infty} 1$. As we know, however, activated events are not typical, and in fact they are not captured by this mean-field (saddle point) approximation.

**Large deviations for the dynamical trajectories: Constrained DMFT.** Activated events occur on timescales that scale with $N$ and cannot be captured by simply taking the large-$N$ limit at finite times, that leads to (130). Instead, one should study the Langevin equation (18) on timescales that diverge exponentially with $N$, which is a challenge (see [123] for an example). However, the large-$N$ formalism can still be used to investigate how an activated event occurs when it is enforced to happen within a finite time. To this end, one could condition the dynamics not only on the initial state but also on the final state, such as two local minima separated by an extensive distance in configuration space and by an extensive energy barrier. This leads to an expression similar to (130), but with a modified action that accounts for the conditioning on the final state:

$$\mathcal{Z}^{\mathrm{dyn}}_{\mathbf{s}_0,\mathbf{s}_f} = \int_{\mathbf{s}(t=0)=\mathbf{s}_0}^{\mathbf{s}(t_f)=\mathbf{s}_f} \mathcal{D}\mathbf{s}_t \, e^{-\mathcal{A}_N[\mathbf{s}_t]} \quad \xrightarrow{\mathbb{E}[\cdot]} \quad \int \mathcal{D}\mathbf{Q} \, e^{-N\mathcal{A}_{\mathrm{const}}[\mathbf{Q}(t,t')]+o(N)}. \tag{132}$$

By taking the variation of the *constrained dynamical action* $\mathcal{A}_{\mathrm{const}}$, one should get different dynamical equations describing *dynamical instantons* [124–126]. These equations differ from standard DMFT and are expected to be non-causal. They describe the most probable way in which a rare event occurs. The action evaluated at these instantonic solutions is positive, consistent with their interpretation as rare events associated to an exponentially small probability. The program would be to derive such equations and classify all possible instantonic solutions. A significant advancement in this direction has been recently achieved in Ref. [127].

**On the shape of high-dimensional instantons.** In [127] the calculation of (132) has been performed for the pure spherical $p$-spin model, choosing as initial and final state two equilibrium configurations at given inverse temperature $\beta$, which are typically at zero overlap with each others. The numerical integration of the resulting equations reveals that the system climbs significantly in energy along the dynamical path, reaching levels far higher than those associated with the energies of finite-rank saddles. The resulting trajectories are very different with respect to the approximate instantons constructed in [118], by constraining the trajectories to pass through the nearest saddles connecting nearby local minima in configuration space, see Fig. 12; it might be that the processes described by the large deviations of the dynamical action are to be interpreted as the effective path emerging from a sequence of local jumps in the landscape. Speculations aside, the analysis and classification of instantonic solutions under different constraints remains an open challenge in this field.

## 5 Conclusion

In these notes, we have discussed an high-dimensional inference problem in two different variations: low-rank matrix, and tensor estimation. Hopefully, this example has illustrated how the study of high-dimensional random landscapes (through the tools and concepts developed within the statistical physics of disordered and glassy systems) provides valuable insights into the optimization dynamics in a variety of contexts where high-dimensionality plays a crucial role, which go beyond the physics of glasses. Large deviation theory, the topic of this Les

Houches school, is among the tools relevant in this context. In particular, developing a theory, beyond mean-field, of high-dimensional dynamics in the presence of metastability remains a largely open problem with fundamental connections to large deviation theory, many of which are still waiting to be explored.

## Acknowledgments

I would like to thank the organizers of the school *Theory of large deviations and applications* (Abhishek Dhar, Joachim Krug, Satya N. Majumdar, Alberto Rosso and Grégory Schehr) for inviting me to give a course, which I was very unfortunately unable to deliver since I got Covid right after reaching the Les Houches School of Physics. I thank all the organizers for allowing me to contribute anyway to the school through these lecture notes. I thank in particular Grégory Schehr, the staff of the School of Physics and some participants and lecturers for nicely dealing with my stay in Les Houches under those circumstances. I also thank the referees of these notes for their suggestions, which helped making them more precise and complete.

**Funding information** I acknowledge funding by the French government under the France 2030 program (PhOM - Graduate School of Physics) with reference ANR-11-IDEX-0003.

## A  Spiked Gaussian matrices: Three exercises

The goal of the first two exercises is to characterize the spectrum of $N \times N$ matrices $\mathbf{M} = \mathbf{J} + \mathbf{R}$, where $\mathbf{J}$ is a GOE matrix with $\mathbb{E}[J_{ij}] = 0$ and $\mathbb{E}[J_{ij}^2] = \frac{\sigma^2}{N}(1 + \delta_{ij})$, while $\mathbf{R} = r\mathbf{w}\mathbf{w}^T$ is a rank-one perturbation, with $||\mathbf{w}||^2 = 1$. We denote with $\lambda^\alpha$, $\alpha = 1, \dots, N$, the eigenvalues of $\mathbf{M}$, and with $\mathbf{u}^\alpha$ the corresponding eigenvectors. The resolvent of $\mathbf{M}$ is

$$\mathbf{G_M}(z) = \frac{1}{z\mathbf{I} - \mathbf{M}} = \sum_{\alpha=1}^{N} \frac{\mathbf{u}^\alpha [\mathbf{u}^\alpha]^T}{z - \lambda^\alpha},$$

and it encodes information both on the eigenvalue density and on the outliers, as the exercises illustrate. References for these two exercises are [19, 128].

The third exercise revisits results of the work [14]. The low-rank matrix estimation problem is formulated in terms of the ground state of the energy landscape:

$$\mathcal{E}_r[\mathbf{s}] = -\frac{1}{2}\sum_{ij} s_i(J_{ij} + rv_i v_j)s_j, \qquad ||\mathbf{s}||^2 = N = ||\mathbf{v}||^2, \qquad \mathbf{J} \sim \mathrm{GOE}(\sigma^2).$$

The behavior of the ground state can be characterized by studying the thermodynamics of the system in the limit $\beta \to \infty$, through the partition function:

$$\mathcal{Z}_\beta = \int_{S_N(\sqrt{N})} d\mathbf{s}\, e^{-\beta \mathcal{E}_r[\mathbf{s}]}, \qquad S_N(\sqrt{N}) = \left\{\mathbf{s} : ||\mathbf{s}||^2 = N\right\}.$$

As a function of temperature, this model exhibits a transition at a critical inverse temperature $\beta_c(r)$, which can be interpreted as a *condensation transition*, like in Bose Einstein condensation.

## A.1 Exercise 1: Replica calculation of the Stieltjes transform

The goal of this exercise is to derive the self-consistent equations for the Stieltjes transform of $\mathbf{M}$, Eq. (40). The starting point of the calculation is the following Gaussian identity:

$$\left(\frac{1}{z\mathbf{1}-\mathbf{M}}\right)_{ij} = \frac{1}{\mathcal{Z}[\mathbf{M},z]}\int\prod_{i=1}^{N}\frac{d\psi_i}{\sqrt{2\pi}}\psi_i\psi_j e^{-\frac{1}{2}\sum_{k,l=1}^{N}\psi_k(z\mathbf{I}-\mathbf{M})_{kl}\psi_l}, \qquad (A.1)$$

with the normalization

$$\mathcal{Z}[\mathbf{M},z] = \int\prod_{i=1}^{N}\frac{d\psi_i}{\sqrt{2\pi}}e^{-\frac{1}{2}\sum_{k,l=1}^{N}\psi_k(z\mathbf{I}-\mathbf{M})_{kl}\psi_l}. \qquad (A.2)$$

The quantities $\psi_i$ are auxiliary variables. We wish to take the average of this expression with respect to the matrix $\mathbf{M}$. However, averaging the partition function in the denominator makes the calculation potentially difficult; to proceed, we make use of a variation of the replica trick, to write

$$\mathcal{Z}^{-1} = \lim_{n\to 0}\mathcal{Z}^{n-1},$$

and follow the standard steps of replica calculations.

(i) *From randomness to coupled replicas.* Using the replica trick, justify why

$$(z\mathbf{I}-\mathbf{M})_{ij}^{-1} = \lim_{n\to 0}I_{ij}^{(n)},$$

where

$$I_{ij}^{(n)} = \int\prod_{a=1}^{n}\prod_{k=1}^{N}\frac{d\psi_k^a}{\sqrt{2\pi}}\psi_i^1\psi_j^1 e^{-\frac{1}{2}\sum_{a=1}^{n}\sum_{l,m=1}^{N}\psi_l^a(z\mathbf{I}-\mathbf{J}-r\mathbf{w}\mathbf{w}^T)_{lm}\psi_m^a}.$$

Show that the average of this expression with respect to $\mathbf{J}$ gives

$$\mathbb{E}[I_{ij}^{(n)}] = \int\prod_{a=1}^{n}\prod_{k=1}^{N}\frac{d\psi_k^a}{\sqrt{2\pi}}\psi_i^1\psi_j^1 e^{-\frac{1}{2}\sum_{a=1}^{n}\sum_{kl=1}^{N}\psi_k^a(z\delta_{kl}-rw_kw_l)\psi_l^a}e^{\frac{\sigma^2}{4N}\sum_{a,b}\left(\sum_{k=1}^{N}\psi_k^a\psi_k^b\right)^2}.$$

Notice: we have ended up with an expression without randomness, in which the replicated variables $\psi^a$ are coupled with each others.

(ii) *Order parameters and Hubbard–Stratonovich.* We would like now to perform the integral over the auxiliary variables $\psi_i^a$; however, this integral contains quartic terms in the exponent. In order to turn such an integral into a Gaussian one, we perform a Hubbard-Stratonovich transformation. We introduce the "order parameters"

$$\Psi_{ab}[\psi] = \frac{1}{N}\sum_{i=1}^{N}\psi_i^a\psi_i^b, \quad a\leq b,$$

and write the integral as

$$\int\prod_{a=1}^{n}\prod_{i=1}^{N}\frac{d\psi_i^a}{\sqrt{2\pi}}\cdots \to N^{\frac{n(n+1)}{2}}\int\prod_{a\leq b}d\Psi_{ab}\int\prod_{a=1}^{n}\prod_{i=1}^{N}\frac{d\psi_i^a}{\sqrt{2\pi}}\prod_{a\leq b}\delta\left(N\Psi_{ab}-\sum_{i=1}^{N}\psi_i^a\psi_i^b\right)\cdots$$

Show that using the integral representation of the delta distributions

$$\delta\left(N\Psi_{ab}-\sum_{i=1}^{N}\psi_i^a\psi_i^b\right) = \int\frac{d\lambda_{ab}}{2\pi}e^{i\lambda_{ab}\left(N\Psi_{ab}-\sum_{i=1}^{N}\psi_i^a\psi_i^b\right)},$$

and introducing the $n \times n$ matrix $\Lambda$ with components $\Lambda_{ab} = 2\lambda_{aa}\delta_{ab} + \lambda_{ab}(1-\delta_{ab})$ and the $N \times N$ matrix $\mathbf{A}$ with components $A_{ij} = z\delta_{ij} + rw_i w_j$, the average can be cast in the following form:

$$\mathbb{E}[I_{ij}^{(n)}] = N^{\frac{n(n+1)}{2}} \int \prod_{a \leq b} d\Psi_{ab} d\lambda_{ab} e^{\frac{N\sigma^2}{4}\mathrm{Tr}_n[\Psi^2] + \frac{N}{2}\mathrm{Tr}_n[i\Lambda\Psi]} f_N[\Psi, \mathbf{w}], \qquad (A.3)$$

with

$$f_N[\Psi, \mathbf{w}] = \int \prod_{a=1}^{n} \prod_{k=1}^{N} \frac{d\psi_k^a}{\sqrt{2\pi}} \psi_i^1 \psi_j^1 e^{-\frac{1}{2}\sum_{a,b}\sum_{l,m}\psi_l^a[\mathbf{I}_N \otimes i\Lambda + \mathbf{A} \otimes 1_n]_{lm}^{ab}\psi_m^b}.$$

(iii) *Gaussian integration and dimensionality reduction.* Performing the Gaussian integral over the auxiliary variables, show that

$$\mathbb{E}[I_{ij}^{(n)}] = \delta_{ij} \int \prod_{a \leq b} d\Psi_{ab} d\lambda_{ab} e^{\frac{N}{2}\mathcal{A}_N[\Psi, i\Lambda]} \left[ (\mathbf{A} \otimes 1_n + \mathbf{I}_N \otimes i\Lambda)^{-1} \right]_{ij}^{11},$$

$$\mathcal{A}_N[\Psi, i\Lambda] = \frac{\sigma^2}{2}\mathrm{Tr}_n[\Psi^2] + \mathrm{Tr}_n[i\Lambda\Psi] - \frac{1}{N}\mathrm{Tr}_{nN}[\log(\mathbf{A} \otimes 1_n + \mathbf{I}_N \otimes i\Lambda)].$$

*Hint.* Use that $\int \prod_{i=1}^{d} \frac{dx_i}{\sqrt{2\pi}} x_l x_m e^{-\frac{1}{2}\mathbf{x}\cdot\mathbf{Kx}} = [\mathbf{K}^{-1}]_{lm} |\det \mathbf{K}|^{-\frac{1}{2}}$ and that $\log|\det \mathbf{K}| = \mathrm{Tr}\log \mathbf{K}$.

Notice: We started from an integral over $Nn$ variables $\psi_i^a$, and we ended up with an integral over the variables $\Psi_{ab}$ and $\lambda_{ab}$, whose number scales as $n^2$. This is a dimensionality reduction which is typical of mean-field problems: the properties of the system for large $N$ are encoded into few global order parameters, such as the overlaps $\Psi_{ab}$.

(iv) *Large-N and saddle point.* The integral can now be computed with a saddle point approximation: show that the saddle point equations for the matrices $\Psi$ and $i\Lambda$ read

$$i\Lambda = -\sigma^2 \Psi, \qquad \Psi = \frac{1}{N}\mathrm{Tr}_{nN}\left[ \frac{1}{\mathbf{A} \otimes 1_n + \mathbf{I}_N \otimes i\Lambda} \right].$$

Show that, plugging the first into the second and assuming that the matrices $\Lambda, \Psi$ are diagonal and replica symmetric, i.e. $\Psi_{ab} = \delta_{ab}\mathfrak{g}$ and $\lambda_{ab} = \delta_{ab}\ell$, one reduces to a single equation for $\mathfrak{g}$ which reads

$$\mathfrak{g} = \frac{1}{N}\mathrm{Tr}_N\left[ \frac{1}{(z - \sigma^2\mathfrak{g})\mathbf{I}_N - r\mathbf{w}\mathbf{w}^T} \right].$$

Using that

$$\mathbb{E}\left[ (z\mathbf{I} - \mathbf{M})^{-1} \right] = \lim_{n \to 0} \mathbb{E}\left[ I_{ij}^{(n)} \right] = \left[ (\mathbf{A} \otimes 1_n - \sigma^2\mathfrak{g}\,\mathbf{I}_N \otimes 1_n)^{-1} \right]_{ij}^{11},$$

justify why $\mathfrak{g}$ is the Stieltjes transform of the matrix $\mathbf{M}$. Show that expanding $\mathfrak{g} = \mathfrak{g}_\infty + \mathfrak{g}_1/N + \ldots$, the leading order term satisfies Eq. 40,

$$\mathfrak{g}_\infty^{-1} = z - \sigma^2\mathfrak{g}_\infty.$$

## A.2 Exercise 2: The isolated eigenvalue and eigenvector projection

The goal of this exercise is to derive the expressions for the isolated eigenvalue and for the eigenvector projection, Eqs. (45) and (46). We assume $r \geq 0$.

(i) Show that if $\mathbf{A}$ is a matrix and $\mathbf{v}, \mathbf{u}$ are vectors, then

$$(\mathbf{A} + \mathbf{u}\mathbf{v}^T)^{-1} = \mathbf{A}^{-1} - \frac{\mathbf{A}^{-1}\mathbf{u}\mathbf{v}^T\mathbf{A}^{-1}}{1 + \mathbf{v} \cdot \mathbf{A}^{-1}\mathbf{u}}.$$

Use this formula (Shermann-Morrison formula) to get an expression for the resolvent operator $\mathbf{G_M}(z)$.

(ii) The isolated eigenvalue, when it exists, is a pole of the resolvent operator $\mathbf{G_M}(z)$, which is real and such that $\lambda^{\text{iso}} > 2\sigma$. Using that $\lambda^{\text{iso}}$ does not belong to the spectrum of the unperturbed matrix $\mathbf{J}$, show that it solves the equation

$$r\mathbf{w} \cdot \mathbf{G_J}(\lambda^{\text{iso}})\mathbf{w} = 1.$$

(iii) Using that $\mathbf{J}$ and $\mathbf{w}$ are independent (free) and that typically $\mathbf{w}$ is *delocalized* in the eigenbasis of $\mathbf{J}$, show that

$$\mathbf{w} \cdot \mathbf{G_J}(\lambda^{\text{iso}})\mathbf{w} \overset{N \to \infty}{\longrightarrow} \mathfrak{g}_{\text{sc},\sigma}(\lambda^{\text{iso}}),$$

where $\mathfrak{g}_{\text{sc},\sigma}(\lambda)$ is the Stieltjes transform of the GOE($\sigma^2$) matrix $\mathbf{J}$.

(iv) Using the self-consistent equation satisfied by $\mathfrak{g}_{\text{sc},\sigma}(\lambda)$, derive the expression of the inverse function $\mathfrak{g}_{\text{sc},\sigma}^{-1}$ and determine its domain; use it to show that

$$\lambda^{\text{iso}} = \frac{\sigma^2}{r} + r, \qquad r \geq \sigma.$$

(v) The eigenvectors projections $\xi^\alpha = (\mathbf{w} \cdot \mathbf{u}^\alpha)^2$ can be obtained from the resolvent as residues of the poles:

$$\xi^\alpha = \lim_{\lambda \to \lambda^\alpha} (\lambda - \lambda^\alpha)\mathbf{w} \cdot \mathbf{G_M}(\lambda) \cdot \mathbf{w}.$$

Use this to show that

$$\xi^{\text{iso}} = -\frac{1}{r^2 \mathfrak{g}'_{\text{sc},\sigma}(\lambda^{\text{iso}})} = 1 - \frac{\sigma^2}{r^2}.$$

*Hint.* Use that if $\lim_{\lambda \to \lambda_0} f(\lambda) = 0 = \lim_{\lambda \to \lambda_0} g(\lambda)$, then $\lim_{\lambda \to \lambda_0} \frac{f(\lambda)}{g(\lambda)} = \lim_{\lambda \to \lambda_0} \frac{f'(\lambda)}{g'(\lambda)}$.

## A.3 Exercise 3: Equilibrium phase diagram

The goal of this exercise is to derive the equilibrium phase diagram and discuss the condensation transition in the $p = 2$ spherical model with interaction couplings given by matrices of the form $\mathbf{M} = \mathbf{J} + \mathbf{R}$.

(i) Call $\lambda^\alpha$ ($\lambda^1 \leq \lambda^2 \leq \cdots \lambda^N$) the eigenvalues of $\mathbf{M} = \mathbf{J} + \mathbf{R}$, and $\mathbf{u}^\alpha$ the corresponding eigenvectors. Call $s_\alpha = \mathbf{s} \cdot \mathbf{u}^\alpha$. Show that the partition function can be written as

$$\mathcal{Z}_\beta = \int d\lambda \int \prod_{\alpha=1}^N ds_\alpha e^{\frac{\beta}{2}\left[\sum_\alpha \lambda^\alpha s_\alpha^2 - \lambda\left(\sum_\alpha s_\alpha^2 - N\right)\right]}.$$

(ii) Show that the thermal expectation value of the mode occupations is

$$\langle s_\gamma^2 \rangle_\beta = \frac{1}{\mathcal{Z}_\beta} \int d\lambda \int \prod_{\alpha=1}^N ds_\alpha s_\gamma^2 e^{-\frac{\beta}{2}\left[-\sum_\alpha \lambda^\alpha s_\alpha^2 + \lambda\left(\sum_\alpha s_\alpha^2 - N\right)\right]} = \frac{1}{\beta(\lambda^* - \lambda^\gamma)},$$

where $\lambda^* > \lambda^\gamma$ for all $\gamma$ is fixed by the equation

$$\sum_{\gamma=1}^N \langle s_\gamma^2 \rangle_\beta = N = \sum_{\gamma=1}^N \frac{1}{\beta(\lambda^* - \lambda^\gamma)}.$$

(iii) The matrix $\mathbf{M}$ is a spiked GOE. Take $r < r_c = \sigma$. Justify why for large $N$ the equation for $\lambda^*$ becomes:

$$\beta = \mathfrak{g}_{\mathrm{sc},\sigma}(\lambda^*), \qquad \lambda^* > 2\sigma,$$

where $\mathfrak{g}_{\mathrm{sc},\sigma}(\lambda^*)$ is the Stieltjes transform of the GOE($\sigma^2$); show that there is a critical temperature $\beta_c = \sigma^{-1}$ and compute the solution $\lambda^*$ for $\beta < \beta_c$. Show that at $\beta_c$, $\lambda^*$ attains its maximal possible value. Show that at low temperature $\beta > \beta_c$ the equation can be solved assuming *condensation* of the fluctuations in the lowest-energy mode:

$$\frac{1}{N}\langle s_N^2 \rangle_\beta = 1 - \frac{1}{\beta\sigma}.$$

This condensation transition corresponds also to a transition between a paramagnet at high temperature, and a "ferromagnet in disguise" at low temperature.

(iv) Consider now $r > r_c = \sigma$, when the maximal eigenvalue is $\lambda^N = \lambda^{\mathrm{iso}} = \frac{\sigma^2}{r} + r$; justify why now the critical temperature is $\beta_c = 1/r$, and a solution of the equation for $\lambda^*$ (with $\lambda^* > \lambda^\gamma$) exists for $\beta < \beta_c$. Show that for $\beta > \beta_c$ it must hold

$$\frac{1}{N}\langle s_N^2 \rangle_\beta = \frac{1}{N}\langle s_{\mathrm{iso}}^2 \rangle_\beta = 1 - \frac{1}{\beta r}.$$

In this regime, the condensation transition coincides with a transition between a paramagnet at high temperature, and a ferromagnet at low temperature.

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
