# Peer review of "High-dimensional random landscapes: from typical to large deviations"

_SciPost Physics Lecture Notes, doi:SciPost Phys. Lect. Notes 102 (2025)_

## Round 1 · Referee Report · Bertrand Lacroix-A-Chez-Toine (Referee 1) · 2025-3-20

Report

In these lecture notes, the author considers the subject of high-dimensional random landscapes and illustrates the general theory on inference problems. This tool can be applied to understand a number of complex/disordered systems arising in subjects ranging from economy, computer science to statistical physics and bayesian inference.

These notes describe this topic in a very pedagogic and well-written way. They refer to very recent and top-level literature in this subject without getting too technical. It also provides illustrations and hands-on exercises for students to get familiar with the content and some technical aspects.

I highly recommend these lecture notes for publication in SciPost Physics Lecture Notes once the small list of changes below are implemented.

Requested changes

Find below a list of typos/comments:

1 - p6: "This properties allows us to draw" should be "This property allows us to draw"

2 - p12: "The Riemannian gradient on the hypersphere is and (N − 1)-dimensional vector" should be "The Riemannian gradient on the hypersphere is an (N − 1)-dimensional vector"

3 - p17: "by computing its the large-N expansion" should be "by computing its large-N expansion"

4 - p28: "An “hard" inference problem: noisy tensors" should be "A “hard" inference problem: noisy tensors"

5 - p30: In the section "In the case of quadratic landscape p = 2, the RMT results imply that:", it seems that point (ii) and (iii) are incompatible: If the variable N (ε)/N "converges to its average", its limiting distribution is a delta function which seems incompatible with the fact that it has a "well-defined limit when N →∞"

This point should be clarified in the revised version

6 - p37: "One can check that that there are values" should be "One can check that there are values"

7 - p40: "These stationary points are marginally stabile" should be "These stationary points are marginally stable"

8 - p40: "stationary points at the equator have no isolate eigenvalues" should be "stationary points at the equator have no isolated eigenvalues"

9 - p42: A limit t\to\infty should appear in Eq. (120) (after the limit N\to \infty is taken)

10 - p42: "An paradigmatic solution" should be "A paradigmatic solution"

Recommendation

Publish (surpasses expectations and criteria for this Journal; among top 10%)

  • validity: top
  • significance: high
  • originality: high
  • clarity: high
  • formatting: excellent
  • grammar: excellent

Author:  Valentina Ros  on 2025-08-09  [id 5716]

(in reply to Report 1 by Bertrand Lacroix-A-Chez-Toine on 2025-03-20)

I would like to thank the Referee for carefully reading these lecture notes and for pointing out the typos and/or imprecisions. In the revised version of the notes, I have implemented the suggested changes. In particular, regarding point 5: thank you for noting that the previous formulation was confusing. To clarify, I have removed the statement that the distribution has a “well-defined limit when N→∞” and have instead written that “the scaled variable N(ε)/N remains of order O(1) when N→∞.” I have implemented this correction in all other instances where the same phrasing appeared, including in Box [B5].

---

## Round 1 · Referee Report · Anonymous (Referee 2) · 2025-6-4

Report

This manuscript deals with the properties of high-dimensional random functions, in particular the number and organization of their stationary points of various indices, and their consequences, in particular for dynamical evolution exploring them. It is well-written with a strong effort to make the presentation as pedagogical as possible, it is therefore complying with the standards of Scipost Physics Lecture Notes and I recommend its publication. I have a series of small remarks and suggestions that could be considered in a revision prior to publication.

  • the two examples of random landscapes treated in Sections 2 and 3 are "planted spin-glasses", in the sense that the distribution of the coupling constants are not simply independent random variables as in usual spin-glasses, but includes a low-rank perturbation to this independent noise background. The motivation for the study of this type of random functions comes from inference problems, as clearly explained in the manuscript, but this connection might be slightly more exploited (in the current version the emphasis is a bit more on the landscape than on its inference interpretation). For instance, the only estimator considered here is the Maximum Likelihood one, corresponding to priors on the signal uniform on the sphere, and with the objective to maximize the probability of the estimated signal in the posterior measure. A few more sentences could be added to discuss the effect of considering more general priors (sparse ones for instance) and of using the Minimal Square Error as a measure of the accuracy of the estimator, which is optimized in a Bayesian perspective by the posterior average. In this more general setting the purely spectral perspective taken in these notes is not necessarily the optimal one, algorithms of the Approximate Message Passing type being able to recover partly the signal even in the absence of outlier eigenvalues in the spectrum of the matrix. The discussion on page 22 is a bit misleading in this sense, as it seems to suggests that the spectral transition is the relevant one for a wide class of priors, whereas several cases escape from this situation. In the same line of thought, the discussion of the dynamics, for instance in Section 2.3.3 for the matrix case, concentrates on the behavior of the energy of the physical model. From the point of view of the inference problem what really matters is the overlap between the current configuration of the dynamics and the planted signal, not the energy per se, maybe this observable and its possible connection with the energy should be discussed a bit more.

  • in the introduction it could be useful to specify whether one should think of the variables as discrete or continuous: when speaking of the "landscape topology" in terms of stationary points one is led to envision continuous ones, but "the total number of configurations accessible to the system" seems to refer to a discrete setting.

  • page 2, "that can be agents in an economy, neurons in biological or artificial neural networks, species in ecosystems, particles or spins in materials and so on", it would sound natural to give here references to review papers on these various applications. Same comment applies after "the emergence of glassy phenomenology in a wide range of fields, including machine learning, quantum optimal control, ecology, and inference" on page 3.

  • page 5, figure 2, why are there two images of "signal + noise" in the leftmost panel? Is "adapted from the web" a sufficiently precise reference?

  • the name "matrix denoising" for the problem handled in section 2 might be slightly confusing, as it is usually associated to extensive-rank signals, maybe "low-rank matrix factorization" is a more common terminology in the inference context.

  • page 6, equation (2), it might be specified that $i \le j$ and $k \le l$, otherwise the covariance contains additional symmetric terms. Same comment for equation (101) page 33.

  • page 6, after equation (4), the justification that the transformation $J \to J_R$ has unit Jacobian is slightly more complicated than just $\det O = 1$ (and incidentally this determinant can also be -1 for an orthogonal matrix). Indeed the transformation $J \to J_R$ acts on a number of variables of order $N^2$, its Jacobian is not simply $O$, but in some sense the tensor product of $O$ with itself.

  • page 6, "one finds that the distribution of the eigenvectors averaged over the GOE ensemble is equivalent to that of random vectors on the sphere", maybe one could be a bit more precise here: a single eigenvector is indeed uniformly distributed on the sphere, but the joint law of the eigenvectors does not factorize as independent random vectors, since they obey an orthogonality condition. Mentioning the Haar measure of the orthogonal group might be a better formulation of this statement.

  • page 6, "we have access to several instances of the noisy matrix $M$", I am puzzled by the "several", it seems that the inference problem considered here consists in trying to infer the signal v from a single instance of $M$, not several ones.

  • page 10, equation (18), one could specify that the noise $\eta$ has zero average.

  • page 10, the articulation between the main text and the box [B3] could be improved, since the box uses equations and concepts (Lagrange multipliers) not yet encountered in the main text at the point where the box is first mentioned. In that box, after equation (28), one could explain that the last component in equation (28) vanishes only if one assumes that s is on the sphere.

  • page 11, after equation (21), one could mention that the eigenvectors $u$ are normalized.

  • page 14, "refer to the cited references for such generalizations", at this point the only reference given on RMT is [10], maybe the other ones should be given here.

  • page 14, in the middle panel it would be more enlightening to have a density with a vertical slope at the edge, since the semi-circle (and most of the other densities encountered in RMT) have root singularities at their edges.

  • page 15, equation (37) the notation $\mathbb{C}^-$ is not defined, its definition could be given a few lines later when "lower-half complex plane" is mentioned.

  • page 19, between equation (52) and (53), "Gaussian random variable", specify its mean and variance. Between equations (54) and (55), maybe recall the definition of $\omega$ which was given before (49). At the end of this paragraph one could mention the paper

S F Edwards and R C Jones, The eigenvalue spectrum of a large symmetric random matrix, Journal of Physics A: Mathematical and General 9, 1595 (1976)

that predates [18] and where the outlier transition for rank one perturbed GOE matrix was found.

  • page 20, in figure 4 the notation $g_N$ for the gaps between eigenvalues should be defined (the definition only appears afterwards in equation (66)), and a warning against the possible confusion with the Stieltjes transform could be given.

  • page 20, in equation (56) the Heaviside function $\theta$ should be defined. Right afterwards, is it really "straightforward" to obtain this from the $r=0$ case? Maybe a few additional explanations could be given, starting from the $r=0$ formula which might not be obvious to all readers.

  • page 20, the sentence before (57) is a bit confusing, the $q^\alpha$ do not describe fully the eigenvectors, only their projections in one direction. Also when $r=0$ $w$ is not really defined, it should be explained that the $q^\alpha$ can then be chosen as the projections onto a fixed but arbitrary vector.

  • page 22, in the description of the phase transition, when $r<r_c(\sigma)$ is the phase a spin-glass or a paramagnet? Maybe a few words to justify the "spin-glass" qualification would be helpful.

  • page 25, the formulation "For the model we are looking at, the solution to these equations has been studied in detail in [32]. One finds that in the mean-field limit, the dynamics does not really depend on $r$" is somehow confusing since [32] did not consider the rank-one perturbation of the GOE. Also it could be useful to recall at this point that the summary of properties stated on pages 25 and 26 correspond to the zero-temperature limit of the dynamics.

  • page 27, could there be a reference for the justification of the result of equation (75), or some explanations (in particular on the exponent $3/2$ for the algebraic prefactor) if it can be easily obtained from the results presented before in the manuscript?

  • page 28, does the scaling function of the critical regime defined in equation (76) depend on $\omega$?

  • page 29 and later, in the p>2 case, some more details should be given on the symmetry properties of the tensors, for instance the variance in equation (79) is probably written assuming that all indices are distinct, by analogy with the GOE matrix case the variance should depend on the number of distinct indices among the $p$ ones. Of course these distinctions do not modify the leading order behavior in the large $N$ limit, but if lower order terms are neglected it would be more pedagogical to mention it. A similar comment apply to the third equation of box [B6] page 34, instead of multiplying by $p!$ one should sum over all permutations between the indices $i$ and $j$, unless you take as understood that $i_1 < i_2 < ... < i_p$ everywhere, and then you should put the factorial in (79).

  • page 31, equation (91), Jensen's inequality does not necessarily imply that the annealed average of ${\cal N}$ is much larger than the typical value, if the quenched and annealed complexity coincide they can be of the same order.

  • page 34, first equation of box [B6], the slashed notation could be explained.

  • page 35, equation (106), shouldn't the exponent of $(1-q^2)$ be $(N-4)/2$ instead of $(N-2)/2$?

  • page 36, the justification of (110) by the self-averaging of $\rho_N$ should be improved: in (109) $\rho_N$ appears as $\exp(N \rho_N)$, hence if the LDP of $\rho_N$ was with speed $N$ one would not have (110). Fortunately the LDP for $\rho_N$ has speed $N^2$, which allows to replace it by its average in (109) and thus justify (110).

  • page 36, equation (112), the function $I(y)$ as defined by the first integral seems to be an even value of $y$, if I'm not mistaken. However the final expression does not seem to have this property, or maybe I'm missing some simplification?

  • page 42, paragraph "An paradigmatic solution", maybe say that what is found in [79] is a solution up to a reparametrization of time, not a fully analytical solution.

  • typos :

  • page 12, box [B3], "is and $N-1$" -> "is an $N-1$"

  • page 13, box [B3], after equation 29, the "Riemannian Hessian" should be $\nabla^2_\perp E_r(s)$ and not $\nabla^2 E_r(s,\lambda)$
  • page 15, "finite $N$ fluctuation are" -> "fluctuations"
  • page 16, "obtained ad" -> "as"
  • page 16, equation (41), isn't it $1/z^2$ instead of $1/z$ inside the square root?
  • page 17, "computing its the large-$N$"
  • page 18, "a transitions"
  • page 23, equation (63), $s_i$ and not $\bf s$ in the left hand side
  • page 26, "taking in limit"
  • page 28, "behaves a power law"
  • page 29, I imagine that "As for the spherical case" should read "As for the matrix case"
  • page 29, a global - sign is missing in the first line of equation (83)
  • page 31, equation (90), there should be a $\mathbb{E}$ in the left hand side of the first equality
  • page 38, caption of figure 7, "The BB implies"
  • page 40, "no isolate eigenvalues" -> "isolated"
  • page 42, equation (120), there is $lim_{t \to \infty}$ missing in the first two terms
  • page 48, "initial and finite" -> "final"
  • page 51, in the hint on Gaussian integrals, $({\bf K}^{-1})_{lm}$ instead of $\hat{K}$, and the exponent of the determinant is $-1/2$.

Recommendation

Publish (easily meets expectations and criteria for this Journal; among top 50%)

  • validity: high
  • significance: high
  • originality: good
  • clarity: high
  • formatting: good
  • grammar: good

Author:  Valentina Ros  on 2025-08-09  [id 5717]

(in reply to Report 2 on 2025-06-04)

I would like to thank the Referee for their very careful reading of these lecture notes and for all the comments and suggestions. Below, I provide an account of the modifications made to the notes, following the Referee’s comments. The changes are listed in the same order as the corresponding comments in the report, and the numbering refers to that order. The numbering of the equations and of the references refer to the updated version of the notes.

1) I thank the referee for this relevant suggestion. I have now included several comments to emphasize that the concepts and methods presented in the notes (with the aim of introducing tools for high-dimensional landscapes) represent a specific and not exhaustive approach to the inference problem. These clarifications, with the relative references, have been included at the following points throughout the notes:

Sec. 1.2, on the scope of the lecture notes: In these notes, we consider the denoising problems as illustrative examples to motivate our discussion of the landscape program. To this end, we focus on a specific inference framework, maximum likelihood, since it naturally maps the denoising problem onto a landscape optimization problem. We caution, however, that this is not the only possible approach to address inference tasks, and in some cases, it may not be the optimal one, as we briefly mention in Secs. 2.3.1 and 3.3.1. For a broader overview of how these denoising problems are addressed within the framework of statistical inference, we refer the interested reader to the review articles [4,9].

Section 2.3.1, on the recovery transition — matrix case: The threshold $(r/\sigma)_c=1$ is associated to one specific procedure to estimate the signal, that is maximum likelihood. One may wonder whether this threshold is optimal, or whether different estimators allow to recover information on the signal for values of $(r/\sigma)<1$. This is a question to address within the Bayesian formalism, and the answer depends on the assumptions made on the statistical distribution of the signal (i.e., in the language of Box [B1], on the prior). For the low-rank matrix approximation problem that we are studying, one can show that the threshold given by maximum likelihood is optimal, for instance, when the signal vector v is taken with a spherical prior, meaning that v is extracted randomly with uniform measure in $\mathcal{S}_N(\sqrt N)$; in this case, $(r/\sigma)=1$ coincides with the detection threshold, below which no estimator is able to distinguish between the spiked matrix and a GOE matrix. On the other hand, maximum likelihood is generally not optimal when the signal prior encodes additional structure, for example sparsity [43].

Sec. 2.3.3, on gradient descent —matrix case: We remark that this corresponds to a specific choice of the optimization algorithm, just as maximum likelihood is a specific choice of the estimator of the signal. Different algorithms may be considered, which can perform better than Langevin at signal estimation. The readers interested in algorithms commonly used in the context of statistical inference and in their application to the low-rank matrix estimation problem are referred to [62, 63]. (The two added references are on AMP)

Section 3.3.1, on the recovery transition — tensor case: Also in the tensor case, the recovery threshold achieved by maximum likelihood is optimal—that is, it coincides with the detection threshold—if a spherical prior is assumed on the signal v [90,91]. As for the matrix case, this is not expected to be generic under more structured assumptions on the statistics of the signal.

Sec. 3.3.3, on gradient descent —tensor case: As in the matrix case, we have focused here on a specific optimization algorithm, Langevin dynamics (or gradient descent). Other algorithms can be considered, which may outperform Langevin. In fact, for the spiked tensor problem, it is known that there exist algorithms that recover the signal at values of the signal-to-noise ratio that scale with N with a smaller power than the $\alpha_c = (p−2)/2$ required by Langevin [9, 64].

2) Indeed, the notes focus on continuous variables. On p. 3, I now write: For ease of discussion, we henceforth assume that the si are continuous variables, and we refer to $\mathcal{E}(s)$ as an “energy landscape" that the system aims at minimizing through its dynamics. Subsequently, on the same page, I have rephrased the sentence as: their number grows exponentially fast with N; that is, it exhibits the same scaling with N as the volume of the configuration space accessible to the system.

3) In the revised version, I have added the general references ([1]–[4]) shortly after the first sentence, when mentioning different examples of landscapes (energy, fitness, loss, and cost functions). In addition, I have modified the last sentence of the introduction to cite volume [8], which collects review papers discussing the emergence of glassy-related phenomena in various contexts. The concluding sentence of the introduction now reads: Recently, the characterization of high-dimensional random landscapes has regained prominence, also driven by the emergence of ruggedness and glassy phenomenology in a wide range of fields, including machine learning, quantum control, theoretical biology, and inference (see [8] for a collection of review papers).

4-6) Following the suggestions, in Figure 2 I have removed one of the two images of the noisy picture and added in the caption a citation to the source of the image. I have replaced 'matrix denoising' with 'low-rank matrix estimation' throughout the notes, and have made the same replacement for the tensor case. Moreover, I have specified the ordering of the indices in the expression for the variance in both places, as suggested.

7) I thank the referee for this remark. I have now introduced the Jacobian of the transformation (just below Eq. (4)) and added a comment to clarify that it is this Jacobian whose determinant has absolute value equal to unity, which in turn follows from the orthogonality of the matrix O. This comment is provided in footnote 1.

8) In the main text, I have now clarified that I refer to a single eigenstate. Moreover, I have added footnote 2 with the following comment: Notice that the joint distribution of the complete set of eigenvectors must include the constraint that they are orthogonal to one another, and this orthogonality condition couples the eigenvectors. The orthogonal matrix made by the eigenvectors can however be viewed as being drawn at random from the set of all orthogonal matrices: every orthonormal basis is equally likely, with no preferred direction. This is phrased mathematically by saying that the eigenvectors are distributed according to the Haar measure on the orthogonal group O(N) of $N \times N$ orthogonal matrices.

9-12) To avoid this confusing formulation, I have removed the word “several” and, just below, added the following comment: Tools of statistical physics allow us to answer this question for typical instances of the noisy matrix, i.e., to characterize what happens with high probability with respect to the noise. I have implemented the other suggested changes.

13) The relevant references are [18] and [20–25], which discuss some of the results in this subsection in more general contexts than spiked GOEs. Since these references pertain to different aspects (e.g., the Wigner law or the typical value of the outliers), I have decide to cite each of them in the corresponding points in the text where the different results are discussed for the spiked GOE case, and I have added at the beginning of the subsection the sentence: readers interested in such generalizations are referred to the specific references cited when each result is presented.

14-16) Thank you for the suggestions, that I have implemented. In particular, I have added the very relevant reference that was erroneously missing, and included the following comment: For rank-1 GOE perturbed matrices, the transition of the typical value was first determined in the seminal work [19]. The transition in the scaling of the fluctuations was instead characterized in [27] for Wishart matrices, and it is now referred to generically as the BBP transition. Ref [19] is now also cited on p. 17, when discussing the typical value of the isolated eigenvalue.

17) Following the suggestion, in Fig. 4 I have added the definition of $g_N$ in the caption. Moreover, I have modified the symbol denoting the Stieltjes transform to avoid any confusion. Thank you for pointing this out.

18) The definition of $\theta(x)$ is now given in Eq. (58). Moreover, I have expanded that section of the notes to include a discussion on how Eq. (57) is derived. The procedure for $r>0$ is very similar to that for $r=0$, as it is now commented at the end of page 21. I hope these comments gives some useful insight on how Eq. (57) arises.

19) Following this suggestion, I have modified the text around Eq. (60) and rephrased the comment as follows: The components of the vector w in the basis $u^\alpha$ are distributed like the components of a vector extracted randomly from the hypersphere. This mirrors the statement that the eigenvectors of GOE matrices have the statistics of an orthonormal basis sampled uniformly, as discussed around Eq. (4) and in Box [B2].

20) Indeed, as the referee suggests this part of the notes lacked some precision regarding the nature of the low-temperature phase. I have added a discussion that hopefully clarifies that this phase is not a "usual"spin glass phase, but rather a “ferromagnet in disguise.” The new paragraph appears at the end of page 23 (not reproduced here because it is a bit lengthy).

21) Following the suggestions, I have modified the formulation of the sentence as follows: For the model we are looking at, not surprisingly one finds that in the mean-field limit the dynamics does not depend on r: exactly as for the eigenvalue distribution, the rank-one perturbation gives sub-leading contributions that are negligible when $N \to \infty$. The solution to the DMFT equations for $r = 0$ has been studied in detail in [45]. We briefly summarize some of the results obtained studying the equations in the noiseless limit $\beta \to \infty$. The footnote 3 specifies: Due to the fact that r does not affect the equations in the mean-field limit, the phenomenology of [45] holds true also for r >0, if one focuses on the short timescale regime. (The current reference [45] was the previous reference [32]).

22-23) Regarding Eq. (78), I have added a citation to Ref. [61], where the corresponding calculation is performed, and included a comment that the prefactor scaling as $t^{−3/2}$ actually arises from the contribution of the eigenvalues in the continuous part of the eigenvalue distribution. Concerning the scaling function in the critical regime, the numerical analysis in Ref. [61] indeed shows that the function depends only on the scaling variable $\omega$ and appears to follow a power law with an exponent linear in $\omega$. I have added these comments around Eq. (79).

24) Indeed, as the referee points out the equations written in the previous version of the notes were correct only up to subleading terms in N, but this was not made explicit. Following the referee’s comment, I have modified the variance in Eq. (82) writing the multiplicity factors that arise due to repetitions in the indices, thereby uniformizing the choice of variance with the matrix case. Moreover, I have added footnote 4 to clarify how these factors arise when constructing a symmetric tensor by taking linear combinations of the entries of an asymmetric tensor. I have also slightly modified the discussion in Box [B6] accordingly, since the covariances were used there.

25-26) I have modified Eq. (94) and added the following comment: Notice that a small difference between the quenched and annealed complexities may correspond to a large difference between the typical and average value of N_N when N is large, due to the exponential amplification. Also, the slashed notation is now explained.

27) Here, I am using the fact that the surface of a sphere of dimension $n-1$ and radius $r$ is given by $\frac{2 r^{n-1} \pi^{n/2}}{\Gamma(n/2)}$. Eq. (109) corresponds to this formula, for $n-1\to N-2$ and $r \to \sqrt{1-q^2}$.

28) Indeed, the argument around Eq. (110) could be made more precise. I have hopefully made it more precise by adding the following sentence: For GOE matrices, the latter takes the Large Deviation form (56). The explicit form of the Large Deviation function $\mathcal{S}[\rho]$ is not needed to proceed with the calculation: the only relevant ingredient is that the probability decays with speed $N^2$, i.e., much faster than the determinant term in (112), which behaves only exponentially in N. When computing (112) via the Laplace method for large N, the term proportional to $N^2$ dominates and must be optimized. This obviously selects the typical density $\rho_\infty(\lambda)$ as the optimizer. Correspondingly, in Sec. 2.2.2, when accounting for Large Deviation results for the GOE, I have added a comment on the Large Deviation probability of the GOE eigenvalue density around the newly added Eq. (56).

29) Indeed, the referee is correct, I(y) is an even function depending on |y|. I have specified that in the text we are giving the explicit expression for I(y) only for y<0.

30) Following the referee’s suggestion, I have added the sentence: This comes from the fact that an analytic solution of the DMFT equations in the large time limit has been found in [95]. The form found in [95] solves the equations up to a reparametrization of time [99]. The newly-added Ref. [99] discuss the issue of time reparametrization.

31) Thank you very much for spotting these typos, which I have corrected in the revised version of the notes.

---

## Round 2 · Author Response

I would like to thank both Referees for their careful reading of these lecture notes and for all their comments and suggestions.
In the v2 of the notes, their suggestions have been accounted for. Below, I give a list of the main changes that I have implemented.
Additionally, I am uploading a version of the v2 of the notes in which the main changes with respect to the v1 are highlighted in blue.

---

## Round 2 · List of Changes

Main changes related to Referee Report 1:

  • Corrected typos mentioned in points 1–4 and 6–10 of the referee's report:
  • Removed the statement that the distribution has a “well-defined limit when $N \to \infty$”, replaced with “the scaled variable $\mathcal{N}(\epsilon)/N$ remains of order O(1) when $N \to \infty$”. Correction implemented in all other instances where the same phrasing appeared, including in Box [B5].

Main changes related to Referee Report 2:

  • Included several comments to emphasize that the concepts and methods presented in the notes (with the aim of introducing tools for high-dimensional landscapes) represent a specific and not exhaustive approach to the inference problem. These comments are included in Sec. 1.2 when discussing the scope of the notes, in Sec. 2.3.1 when discussing optimality of maximum likelihood in the low-rank matrix estimation problem, in Sec. 2.3.3 when discussing gradient descent dynamics, and in Sections 3.3.1 and 3.3.3 when discussing the corresponding issues for low-rank tensor estimation. Clarified that the notes focus on continuous variables;
  • Added Refs. [1]–[4] and [8], that provide examples of applications of the problem of high-dimensional landscapes optimization in several contexts;
  • Modified Fig. 2 and added a reference to the source of the image in the caption;
  • Replaced 'matrix denoising' with 'low-rank matrix estimation' throughout the notes, and similarly for the tensor case;
  • Modified the discussion on the Jacobian of the change of variable around Eq. (4);
  • Modified the discussion on the statistics of eigenvectors of GOE matrices, added a mention to the Haar measure (footnote 2);
  • Clarified that the denoising problem is addressed for typical instances of the noise;
  • Implemented the changes suggested in points 10-19 of the report;
  • Added a discussion on how the joint distribution of eigenvalues and eigenvectors’ projections is obtained, Eq. (57);
  • Added comments to clarify the nature of the low-temperature phase of the p=2 spherical p-spin model;
  • Clarified that the DMFT analysis is independent of the parameter r, and thus the analysis presented in Ref. [45] and discussed in the notes is valid also for r>0, within the timescales accessible with DMFT;
  • Improved the discussion around Eq. (79), on the dependence of the scaling function in the critical regime;
  • Corrected the expression for the variance of the symmetric random tensor, Eq. (82). Added footnote 4 to clarify the origin of the combinatorial terms;
  • Implemented the changes suggested in points 25 and 26 of the referee's report;
  • Improved the justification of Eq. (112);
  • Specified that the expression for I(y) given in Eq.(115) holds true for y<0;
  • Added a comment on reparametrization invariance regarding Ref. [95];
  • Corrected typos identified by the referee (and some more);
  • Added references.

---

## Editorial Decision

published